



# High-resolution global map (100 m) of soil organic carbon reveals critical ecosystems for carbon storage

Cynthia M. Crézé[1,2], Sassan S. Saatchi[1,2,3], Nicholas Kwon[2], Yan Yang[2] and Shuang Li[2]

[1]Institute of the Environment and Sustainability, University of California, Los Angeles, Los Angeles, United States
5  [2]CTrees, Pasadena, United States
[3]NASA Jet Propulsion Laboratory, California Institute of Technology, Pasadena, United States

*Correspondence to*: Cynthia M. Crézé (ccreze@ucla.edu)

**Abstract.**

10      Uncertainty in soil organic carbon (SOC) stocks and fluxes resulting from land disturbance and recovery processes remains a significant challenge for closing the global carbon budget, accurately quantifying the land carbon sink, and assessing restoration carbon credits in nature-based climate solutions. To address this, we develop a spatially resolved SOC estimate at a 1-hectare resolution globally, aligning with the scale of land-use disturbances, to significantly improve carbon accounting accuracy and reduce uncertainty across multiple use cases. We compile and harmonize a global SOC inventory, incorporating 84,880 (30 cm depth) and 44,304 (100 cm depth) measurements. Additionally, we identify high-resolution remote sensing and in situ spatial covariates to map SOC using advanced, biome-specific machine learning algorithms.

We measure global SOC stocks of 1,049 Pg C at 30 cm and 2,822 Pg C at 100 cm. Our results reveal a 31% increase in SOC at 30 cm and a 45% increase at 100 cm compared to the average of prior estimates. Our model indicates that peatlands including peat-in-soil mosaics store 146 Pg C at 30 cm depth and 344 Pg C at 100 cm depth, accounting for 14% and 12% of global SOC stocks respectively. Mangrove ecosystems have some of the highest soil carbon densities among global biomes, and hold 1.3 Pg C at 30 cm depth and 4.4 Pg C at 100 cm depth, despite covering a relatively small global area. We find that biome-level SOC estimates strongly depend on biome area and its changes over time. Our analysis indicates that annual wildfire dynamics and shifts in agricultural land can influence SOC by 132 Pg C and 140 Pg C at 30 cm, and by 345 Pg C and 368 Pg C at 100 cm, representing approximately 13% of the global stocks. The SOC maps from this study, including pixel-level 95% confidence intervals to quantify model uncertainty, are hosted on a Zenodo repository. These data will be made publicly available upon publication to support large-scale carbon accounting and integration by the scientific and policy communities. The repository is accessible through the reviewer link (Creze et al., 2025): https://zenodo.org/records/15391412?preview=1&token=eyJhbGciOiJIUzUxMiJ9.eyJpZCI6ImNjZjk2YTM5LTQyM2ItNDZiMC1iY2RlLTg0ZTA1ZjU1MDZjNSIsImRhdGEiOnt9LCJyYW5kb20iOiIxMjc1YmI0OTZhOTNiMmQyNTIxYjYyNzRiM2ZlZjBmMyJ9.M5VUSwR4GkeoKV1Kno1v3b3qLUAzErns1Zh6u0om2HhVDrnxcjKJS3WCOVAoJlSyxt-5Kbc809apXwYmAnMqyQ



## 1 Introduction

Soil organic carbon (SOC) is a vital component of the carbon cycle, playing a central role in global biogeochemical
cycling. As plant materials decompose, they constitute SOC, forming a dynamic and essential carbon reservoir (Lal et al.,
2004; Schlesinger and Bernhardt, 2013). Soil carbon accumulation is often described as having a multiplier effect, wherein
the storage of carbon not only sequesters atmospheric carbon but also supports ecosystems' capacity for future carbon
capture through sustained plant productivity. Ecosystems maintain a steady-state soil carbon concentration through a balance
of rapid (e.g., microbial decomposition) and gradual (e.g., humidification) carbon cycling processes. This ecosystem stability
is crucial, as it helps regulate carbon fluxes and contributes to the broader regulation of terrestrial and atmospheric carbon
pools.

Despite the pivotal role of SOC in the global carbon cycle and as a major carbon pool for climate mitigation
policies, its total magnitude, geographical distribution, and its dynamic changes remain uncertain. In recent years, several
global maps have been developed to represent the distribution of SOC across ecosystems. Key SOC datasets include the
World Inventory of Soil Emission Potentials (WISE30sec v.3), the Harmonized World Soil Database (HWSD v1.21), the
Global Soil Dataset for Earth System Models (GSDE), the Global Soil Organic Carbon map (GSOCmap v1.5), the
Sanderman soil carbon debt map (2017) and SoilGrids250m. These are commonly used in global studies (Tifafi et al., 2018;
Dai et al., 2019; Lin et al., 2024). Collectively, these maps suggest that soil carbon stocks represent between 588-1,176 Pg C
at 30 cm depth and 1,130-2,769 Pg C at 100 cm depth. These differences in estimates are substantial, exceeding atmospheric
carbon stocks (~850 Pg C). On a global scale, current SOC stock estimates diverge significantly, highlighting the challenges
in achieving consistent global SOC measurements.

Variations in global datasets have been attributed to differences in data sources, mapping approach and statistical
methods (Lin et al., 2024). Scharlemann et al. (2014) observed that studies using the same underlying dataset can yield
varying SOC stock estimates, due to differing statistical models. For instance, Tifafi et al. (2018) report a SOC stock of
3,400 Pg C using SoilGrids250m at 100 cm depth, while Wang et al. (2022) report 3,011 Pg C with the same dataset.
Similarly, using WISE30sec, Batjes (2016) reports 1,408 Pg C, whereas Wang et al. (2022) reports 1,827 Pg C, representing
a 30% difference. Such variability can occur due to differences in calculation methods, as noted by Köchy et al. (2015). Of
global models, some perform better in certain regions than others, with varying accuracy across ecosystems. Lin et al. (2024)
found that WISE30sec performs well in the northern circumpolar permafrost regions and GSDE has superior accuracy in
China. All prior maps consistently indicate large SOC stocks in higher latitudes (Tifafi et al., 2018), and there is relatively
high alignment in grasslands, croplands and shrublands/savannas (Lin et al., 2024). However, significant knowledge gaps
remain across tropical regions, particularly around peatland extent and condition in Central Africa, Amazonia, and SE Asia
(UNEP, 2022). Carbon stocks in tropical areas are not well characterized due to paucity of measurements. Recent datasets
provide critical insights into carbon stocks of the Amazonian Basin and the Congo Basin (Draper et al., 2014; UNEP, 2022)
and highlight the essential role of peatlands in global carbon storage (Crezee et al., 2022; Hastie et al., 2022). Addressing



these knowledge gaps and enhancing the accuracy of global SOC estimates is crucial for understanding the global carbon cycle, and to develop effective carbon management strategies.

Quantifying soil organic carbon is challenging due to its fine-scale heterogeneity, which is shaped by a combination of edaphic conditions, soil properties, vegetation cover, climate, and both natural and human-driven land use change. One

example of such land use change is the widespread conversion of forests to cropland and grassland, much of which is associated with small-scale agriculture (Tyukavina et al., 2018; Branthomme et al., 2023). Smallholder farming tends to operate at very fine spatial scales - in Indonesia, farms of 5 ha or less account for 88% of the agricultural area, while in Sub-Saharan Africa, farms under 2 ha comprise roughly 80% (Lowder et al., 2016). Given the highly localized nature of land use activities, earlier SOC maps with coarse spatial resolution may fail to detect significant variation in SOC stocks across

landscapes. Specifically, prior maps ranged in resolution from 250 m (SoilGrids) to >1 km (GSOCmap, Sanderman, WISE30sec, GSDE, and HWSD). While these products offer a broad view of soil carbon stocks, they lack the granularity needed for detailed local analysis. Satellites collecting a wide range of measurements across optical and microwave spectra at fine resolutions (10-30 m) provide unprecedented information on vegetation, edaphic conditions, and land use disturbances, serving as key covariates for mapping SOC distributions at fine scales (Dargie et al., 2017; Rudiyanto et al.,

2018; Melton et al., 2022).

Here, we present new global maps of soil organic carbon at 100 m spatial resolution. These maps were produced by integrating multiple satellite observations, with a homogenized global inventory of soil carbon stock measurements in an advanced machine learning algorithm. To enhance prediction accuracy and prevent model overfitting, we developed specialized algorithms that separately model critical ecosystems, notably peatlands and mangroves. These predictions were

aggregated to produce depth-specific SOC maps, representing carbon stocks of topsoil (30 cm depth) and subsoil (100 cm depth) pools globally. We quantified map uncertainty at the pixel level through bootstrapping and error propagation, providing confidence levels to ensure reliable interpretation. Our analysis of these new SOC maps includes the assessment of carbon distribution across biomes, and a comparison with prior global SOC datasets to advance our current understanding of global SOC patterns. Further, we examine the impact of major land-use activities and wildfire on SOC stocks. Through this

integrated approach, we deliver an updated, high-resolution global SOC dataset that supports improved land management, carbon accounting, and global carbon monitoring.

## 2 Methods

### 2.1 Ground truth data

We collected publicly available datasets for two depths: 30 cm and 100 cm. These are widely recognized

international standards for SOC stock assessments (Smith et al., 2020; Fowler et al., 2023). Unlike continuous soil depth functions (Malone et al., 2009), which have shown significant variability in results (Mishra et al., 2009; Kempen et al., 2011; Taghizadeh-Mehrhardi et al., 2016), our method focused on distinct depth layers. This approach allows for more accurate



SOC assessments by developing independent models for each depth, using depth-specific ground truth samples and covariates (e.g., pH, CEC). At a depth of 30 cm, we included 84,880 ground truth data points, of which 3,372 were specific
to peatlands and 1,577 to mangrove forests, both used to train unique models. At a depth of 100 cm, we included 44,304 ground truth data points, including 1,936 for peatlands and 1,453 for mangrove forests. The distribution of these data points is provided in Figures S1 and S2.

### 2.1.1. Global and national datasets

Our study drew from a range of global and national datasets, with the International Soil Carbon Network v.3.
database (ISCN) providing the largest global coverage, followed by the World Soil Information Service (WoSIS) snapshot from December 2023 and the European LUCAS 2018 campaign, provided by the European Soil Data Centre (ESDAC) (Fernandez-Ugalde et al., 2022; Calisto et al., 2023; Batjes et al., 2024). We integrated samples from the Global International Soil Reference and Information Centre (ISRIC)-World Inventory of Soil Emission Potentials (WISE30sec v.3.) (Batjes, 2016). However, we selected only samples with available bulk density measurements, which are essential for calculating soil
organic carbon stocks.

Our study also integrated data from national databases. In North America, we included the Rapid Carbon Assessment (RaCA) dataset for the United States (Soil Survey Staff RaCA, 2013). In this dataset, bulk density values were derived through a combination of direct measurements and modelling. For Mexico, SOC stock values were obtained from Guevara et al. (2020) and included data from the Instituto Nacional de Estadística y Geografía (INEGI) as well as
Krasilnikov et al. (2013). The study provided SOC stock values using bulk density estimates derived from soil type maps, texture, organic matter content, and soil structure, rather than from in situ measurements. We included ground truth data from the Canadian National Pedon Database (NPDB), selecting entries with available bulk density data (Agriculture and Agri-Food Canada, 2021). In South America, we included the recent CLSoilMaps dataset, which had already been integrated into the WoSIS dataset (Dinamarca et al., 2023). We also included the Ecuador Harmonized Soil Database v0.2 (HESD). In
this dataset, UTM coordinates were converted to decimal degrees. The HESD improves spatial representativeness nationally, compared to the WoSIS database (Batjes et al., 2020; Armas et al., 2022). However, Armas et al. (2022) note that certain areas in Ecuador remain underrepresented, particularly within the Amazon region. Additionally, we included a recent national dataset for Guatemala (Vásquez-Toxcón et al., 2023). In this dataset, we excluded data points without bulk density measurements, which accounted for 23% of the dataset. In Brazil, we included version 1 of MapBiomas's Training Field Soil
Data Collection for soil carbon mapping, which integrates multiple datasets published prior to 2021. Carbon stocks were already calculated and provided within this dataset. Finally, we incorporated a tropical montane dataset from China, which provided both in situ bulk density and final SOC stock measurements (de Blécourt et al., 2017). The full list of global and national datasets is provided in Table S1.



### 2.1.2. Biome and Ecosystem-specific datasets

Ecosystems such as peatlands, permafrost regions, forested wetlands, and mangroves are known to store large quantities of carbon but are often unevenly represented in global maps. To improve the representation of major soil carbon pools globally, we incorporated biome- and ecosystem-specific ground datasets that capture key carbon-rich ecosystems. These datasets were integrated into our modelling framework to improve the spatial and ecological completeness of soil carbon stock estimates across diverse biomes.

Wetland ecosystems including peatlands, mangroves, and forested wetlands received particular attention due to their critical role in global carbon storage and their frequent underrepresentation in large-scale soil datasets. To improve the representation of forested wetlands, we included the 'cryptic carbon' dataset developed by Stewart et al. (2024). For mangrove and peatland-specific data, we drew extensively from regional field campaigns conducted by SWAMP, a collaborative effort between the Center for International Forestry Research (CIFOR) and the USDA Forest Service (USFS).

Much of the datasets covered tropical regions, particularly in South-East Asia and Brazil, where mangrove and peatland ecosystems are common. In South-East Asia, we added datasets from the Katingan, Bunaken, Kubu Raya, Sembilang, Teminabuan and Timika regions in Indonesia (Saragi-Sasmito et al., 2015; Murdiyarso et al., 2019a, b, c, d, e), as well as from West Papua (Sasmito et al., 2019), and from Can Gio and Can Mau in Vietnam (Vien et al., 2019a, b). In Brazil, we included a dataset specific to the Acarau Boca mangroves (Kauffman et al., 2019). We also integrated global mangrove

assessments including work by Atwood et al. (2017) and the Nature Conservancy – WHRC dataset by Sanderman et al. (2018). Further, we included recent peatland data from the Congo Basin (Crezee et al., 2022) and the Peruvian peatlands (Hastie et al., 2022) and included datasets on non-forested wetlands notably the MarSOC dataset (Maxwell et al., 2023), to improve the coverage of these ecosystems. The complete list of datasets is provided in Table S2.

       For high-latitude systems, we included data from the Northern Circumpolar Soil Carbon Dataset (NCSCDv2)

(Hugelius et al., 2013) to account for permafrost soils, which store vast amounts of carbon, as well as additional data from Sweden (Siewert, 2018). Overall, added datasets improved representation across tropical and permafrost regions, capturing carbon stocks across diverse and often underrepresented biomes.

### 2.1.3. High-confidence gridded datasets

       We integrated high-quality national and regional maps derived from local ground datasets. These gridded products

benefited from regional validation and the use of comprehensive national datasets combined with local knowledge, providing strong ground truth support for their predictions.

       In cases where country-level point data were not publicly accessible, we applied a randomized subsampling approach to national and regional maps, using 10 bins with 50 samples per bin. This allowed us to incorporate information from these products while minimizing potential biases introduced by dense spatial coverage. We processed recent maps of



Australia (Viscarra Rossel et al., 2014), the Third Pole region (Wang et al., 2021a), Tanzania (Kempen et al., 2019), and South Africa (Venter et al., 2021).

We also subsampled gridded maps derived from recent high-confidence in-field surveys targeting ecosystems with sparse direct measurements, such as peatlands and mangroves. To ensure spatial representativeness, we applied a randomized selection using 10 bins with a fixed number of samples per bin. For Peruvian and Congo Basin peatlands (Hastie

et al. 2022; Crezee et al., 2022), we selected 25 samples per bin, while for global mangrove regions (Sanderman, 2018), we used 50 samples per bin.

Overall, the subsampling approach only moderately impacted our machine learning performance metrics (RMSE, $R^2$, and MAE) by <10%. A complete list of the subsampled rasters is available in Table S3.

**2.1.4. SOC stock calculation**

For each raw data point, SOC stock was calculated in tons of carbon per hectare according to Eq. (1), following the method used by Hengl et al. (2017):

$$SOC \ (tons \ C \ / \ ha) = C \ x \ BD \ x \ D \ x \ (1 - CF \ / \ 100) \ , \qquad (1)$$

where C is the soil organic carbon concentration (%), BD is the bulk density (g/cm³), D is the soil layer depth (cm), and CF is the coarse fragment content (%).

As mentioned in a previous section, the analytical methods used to assess soil carbon (i.e., dry combustion or the Walkley-Black method) vary across soil datasets, particularly given the extended timeline over which SOC data were collected. Consistent with prior studies (Guevara et al., 2020), we opted to retain original SOC values without adjusting for potential method-based discrepancies. Further, in global datasets, the reporting of BD measurement methods was inconsistent, and sample drying methods varied across datasets. When method information was available, we selected the

most accurate protocol: oven-drying at 110°C as opposed to desorption to 33 kPa, or air drying. Coarse fragment content was also not uniformly measured across datasets. We adjusted for the fine earth fraction when possible. These differences in sampling protocols are well-known sources of error in global soil carbon estimation and remain inherent to global soil organic carbon databases (Hengl et al., 2017; Poeplau et al., 2017). Approaches to address these methodological gaps are actively being developed to improve the consistency and accuracy of SOC assessments.

Most soil sampling campaigns followed layered protocols. However, soil depth specifications varied widely amongst datasets. To generate a homogenized and comprehensive dataset, we developed a standardized approach for sample preprocessing. Our approach identifies nested layers, merges them and computes SOC sums for these nested sets. In our 30 cm depth sample processing, we discarded samples for which the bottom layer was less than 20 cm in depth or was greater than 45 cm. For our 100 cm depth sample processing, we excluded samples with a bottom layer of less than 80 cm in depth

and if necessary, truncated the layer containing the 100 cm depth. We either extrapolated or truncated SOC measurements which met these depth requirements using linear projection. For peatland-specific datasets, we did not apply the 80 cm




cutoff, as sampling methodologies in these regions typically focus on deeper peat soil profiles rather than only topsoil layers. Additionally, due to the scarcity of peatland soil data, we adjusted the filtering criteria to maximize the inclusion of available peatland data. We excluded soil samples for which soil depth was not specified. In the development of the broader

homogenized dataset, we checked for dataset redundancy by filtering samples with matching geocoordinates and matching SOC values and removed repetitive samples. Overlap primarily occurred in data points located in the United States.

## 2.2. Remote sensing datasets and Environmental covariates

Environmental covariates were selected to represent soil formation factors spatially, according to Jenny (1941): edaphic, climatic, vegetative, topographic and anthropogenic. While some ground datasets provided a range of soil and

environmental covariates at each sample site, most in-situ carbon measurements did not provide additional data. Therefore, we followed the approach of Hengl et al. (2017), using spatially continuous data from remote sensing and gridded soil data to inform SOC predictions. First, we incorporated the ISRIC SoilGrids 250m v2.0 collection accessible on Google Earth Engine, which provides global raster layers of soil property predictions at different depths (0, 5, 15, 30, 60, 100 cm) (Hengl et al., 2017). We selected standard numeric soil properties that are key factors in soil carbon cycling: Cation Exchange

Capacity (CEC), pH, soil texture fractions (clay, sand, and silt), and total nitrogen. We calculated mean pixel-based values for 30 cm and 100 cm depths by combining weighted averages for each property across corresponding depth layers. The following units were used: CEC in mmol/kg, the proportions of clay, sand, and silt in g/kg, total nitrogen in cg/kg, and pH represented as pHx10. These layers were prepared using the Google Earth Engine script editor and exported as individual rasters for 30 cm and 100 cm depths for each soil property.

We also selected environmental and biophysical layers that capture key factors influencing soil carbon dynamics. We used the Global 30m Digital Elevation Model (DEM) from the COPERNICUS/DEM/GLO30, as elevation influences soil properties, erosion, water flow, and organic matter accumulation. To account for climate-driven factors, we integrated MODIS Evapotranspiration (ET) data (kg/m²/8-day), capturing global evapotranspiration patterns (MODIS/061/MOD16A2GF), and Land Surface Temperature (LST), both day and night, from the

MODIS/061/MOD11A2 dataset, to account for temperature effects on soil carbon. For these dynamic covariates, we calculated the mean over the largest available timeframe—2000-2022 for MODIS ET and 2000-2024 for LST—ensuring long-term trends are well-represented while minimizing short-term variability. Similarly, we calculated the average number of burned days per pixel per year from 2000 to 2023 to capture long-term fire history and its impact on soil carbon dynamics. Fire frequency measured in number of burned days per pixel per year, was derived from the MODIS/061/MDC64A1 dataset

within this timeframe. We processed and integrated the global bare soil cover fraction from the Copernicus Global Land Cover Layers (CGLS-LC100). This provides critical information on the extent of bare ground, a key factor influencing carbon loss and storage. We used ALOS-2 PALSAR-2 data for synthetic aperture radar (SAR) backscatter (HH and HV bands) to account for the effects of soil moisture and structure on soil carbon. We calculated the median for the years 2019-2020 for each band, to align with the timeframe of the Global Peatland Assessment (GPA) Database 2022 v.2, used in this





study. We processed Landsat 8 data (bands red, NiR, SWIR1 and SWIR2), taking the median for 2022 and gap-filling with data from prior years. This approach captures vegetation and surface reflectance. Collectively, these datasets provide a comprehensive suite of predictors to model soil carbon stocks globally, accounting for both physical and environmental drivers. All covariates were converted to the coordinate system EPSG:4326 WGS 84 and the extent: [-180, -56.25, 180, 83.86 degrees]. The source of remote sensing and environmental covariate datasets are provided in Table S4.

**2.2.1. Categorical datasets**

To better delineate ecosystems and soil groups, we included categorical datasets for model training. This included the WWF ecoregions, which define 867 global terrestrial ecoregions, and the World Reference Base (WRB) 2006 subgroup classes from SoilGrids250m (2017-03), which distinguish 118 ecologically distinct soil classes globally. We also included recent datasets, such as the global tidal marsh distribution and the NCSCDv2 Circumpolar permafrost region extent, as
binary layers, to further improve the model's accuracy and to reflect the distinct characteristics of different ecosystems.

Mangroves and peatlands are vital blue carbon ecosystems that are highly vulnerable to land use activities. Categorical datasets representing their extent were used as training layers in the global model. We also developed ecosystem-specific models using these layers, providing a more accurate baseline for these environments. For mangroves, we used the Global Mangrove Watch v.3.0. for year 2020 (Bunting et al., 2022), which reports a mangrove forest extent of
13 Mha for year 2020. For peatlands, we used the Global Peatland Assessment Database 2022 v.2 (GPM2.0) from UNEP. This dataset categorizes peatlands into two primary classes: 'peat dominated', where peat is the dominant land cover, and 'peat-in-soil mosaic', where peat exists in smaller patches interspersed within other land cover types. We combined both classes (GPM2.0 classes 1 and 2) into a single mask to capture the full extent of peat-dominated and peat-in-soil mosaics. Including both peat-dominated and peat-in-soil mosaics classes increases the peatland extent to 858 Mha. To enhance
regional accuracy, we also integrated high-resolution peatland masks for the Congo Basin and Peruvian Amazon, ensuring recent and comprehensive data coverage in these key tropical regions.

We masked oceans, permanent water bodies, and snow cover using the Copernicus Global Land Cover Layers (CGLS-LC100 Collection 3). The source of categorical datasets is detailed in Table S5.

**2.2.2. Machine learning inference, bias-correction and map visualization**

We first developed global models at 30 cm and 100 cm soil depths, applying an 80/20 split for training and validation. This model was implemented using a Random Forest algorithm with 300 trees. We selected this model for its strong predictive capacity and interpretability. Model metrics are provided in Figure S3.

Peatland and mangrove ecosystems were modelled separately for each soil depth, due to their distinct biophysical characteristics and the importance of their soil carbon stocks. Each model was developed by extracting all ecosystem-
specific ground data points from the global dataset using peatland and mangrove extent masks defined in the categorical dataset section. Each model was trained with the same layers as for the global model. This approach allowed for targeted





evaluation of model performance within a specific domain where biophysical processes differ from the global average. The mangrove-specific model outperformed the global model at both 30 cm and 100 cm depths. In contrast, the peatland-specific model showed improved performance over the global model only at the 100 cm depth, suggesting that ecosystem-specific patterns are more pronounced in the deeper soil layer. Although the peatland model at 30 cm did not outperform the global model, it was retained to maintain consistency in the modelling framework across depths. Paired models for each depth under the same ecosystem-specific logic support a systematic structure and facilitates comparability across layers. We integrated the predictions from ecosystem-specific models into the global gridded dataset to create the final map.

To improve the alignment between model predictions and observed values, we applied a histogram-based bias correction to the global model outputs (Figure S4 and S5). This correction was then applied to the global prediction rasters to reduce systematic error and enhance consistency with ground observations. While this post-processing step resulted in a decrease in $R^2$ (0.35 to 0.12 at 30 cm and 0.38 to 0.32 at 100 cm), it significantly reduced the overall prediction bias and improved the match between the predicted and observed SOC distributions, which is particularly important for large-scale carbon accounting applications.

## 2.3. Global map comparison

To compare our results, we analysed commonly used SOC global maps, which include WISE30secv.3, HWSDv1.21, GSDE, GSOCmap v1.5, the Sanderman soil carbon debt map (2017) and SoilGrids250m. The Harmonized World Soil Database (HWSD v1.21) was developed jointly by the FAO, the International Institute for Applied Systems Analysis (IIASA), the International Soil Reference and Information Center (ISRIC)-World Soil Information, the Institute of Soil Science, Chinese Academy of Sciences (ISSCAS), and the Joint Research Center of the European Commission (JRC) in 2012. The Global Soil Dataset for Earth System Models (GSDE) builds on the HWSD framework and included a broader dataset to improve mapping accuracy. Similarly, the ISRIC-WISE30sec expands upon HWSD by including additional soil profiles and climate zone data for more detailed SOC estimates (Batjes, 2016). The Global Soil Organic Carbon map (GSOCmap v1.5) was developed by the FAO and the Intergovernmental Technical Panel on Soils (ITPS) in 2018 (FAO, 2020). Its framework was built through a participatory process with focus on global carbon analyses. The Sanderman soil carbon debt map (2017) was developed to compare current SOC levels to pre-agricultural baselines and is hosted on the Woods Hole Research Center's GitHub repository. We included both versions of SoilGrids in our analysis due to their substantial methodological differences. The latest version SoilGrids (v2.0) was released in 2020 but is only currently available at 30 cm depth (Poggio et al., 2021). These global maps are widely used SOC products, representing diverse methodological approaches to SOC estimation (Lin et al., 2024). All mapping units are in tons of carbon per hectare, with further details of the global map datasets provided in Table S6.



## 2.4. Zonal statistics

To calculate ecosystem-specific and land-use-specific carbon stocks, we applied binary and categorical masks. We used the WWF biomes classification, which delineates 14 distinct biomes to calculate carbon stocks (Olson et al., 2001). To study SOC distribution across different soil classes globally, we used the FAO GBSmap v1.0 Distribution map (FAO, 2022) and the SoilGrids250m 2.0 Bulk Density dataset (Hengl et al., 2017). To delineate agricultural land, we used LGRIP v001 (Landsat-Derived Global Rainfed and Irrigated-Cropland Product). We combined bands for 'irrigated' and 'rainfed' agriculture into a single value. This product is derived from Landsat 8 data collected between 2014 and 2017, resulting in a nominal 2015 product (Teluguntla et al., 2023). Masks for permafrost, peatlands and mangrove forests were described in previous sections and detailed in Table S5. As previously described, a fire frequency raster was used as a model training layer. From this raster, a binary fire mask was created, where any pixel with an average annual number of burned days ≥1 was assigned a value of 1 and classified as fire-prone land. Spatial masks from Schüler (2022) and Paredes-Trejo et al. (2022) were used to delineate the Cerrado and Amazon biomes, respectively. Full detail of datasets used for zonal statistics is provided in Table S7. We applied a bounding box of (-180, -56.25, 180, 83.86 degrees) and a reprojection of 100-meter resolution (0.000897, 0.000897 degrees) to align datasets with the global extent of our SOC map.

To accurately compute zonal statistics on the global raster, pixel areas were corrected for latitude-dependent distortion inherent in geographic coordinate systems such as EPSG:4326 (WGS 84). Pixel dimensions were first computed in meters based on their geographic latitude. The north-south pixel size was calculated as a constant conversion from degrees to meters, while the east-west pixel size was adjusted by the cosine of the latitude to account for the Earth's curvature. Each valid pixel within the specified bounding box (-180, -56.25, 180, 83.86 degrees) was processed individually. Pixel areas in square meters were then converted to hectares (1 ha = 10,000 m$^2$). SOC density values (measured in tonnes of carbon per hectare, t C/ha) were multiplied by the area of each pixel (in hectares), resulting in the total carbon stock (in tonnes). These pixel-level carbon stock values were then summed across all tiles to obtain the global SOC stocks in Petagrams of carbon (Pg C).

## 2.5. Uncertainty assessment

To create a pixel-based uncertainty map, we used a bootstrapping approach, which involved splitting the dataset into 70% for training and 30% for testing. We trained 20 separate Random Forest models, each built on different subsets of the training data, and subsequently generated corresponding global SOC maps. The pixel-based standard deviation was computed from the outputs of these models to quantify the uncertainty associated with each pixel's SOC value. The use of 20 models for bootstrapping was evaluated to be reasonable for providing the confidence intervals given the very high cost of bootstrapping for a larger number of models at the global scale. To derive confidence interval (CI) maps, we multiplied the pixel-based standard deviation by the Z-scores corresponding to a 95% probability level (1.96). This step provides a statistical framework to assess the reliability of the predicted SOC values across the global maps. For both the 30 cm and 100



cm depth maps, we produced two uncertainty maps: one indicating the pixel-based standard deviation and another representing the 95% confidence intervals.

## 3 Results & Discussion

### 3.1. Global SOC distribution

### 3.1.1. Global maps comparison

We measure 1,049 Pg C in topsoils (30 cm) and 2,822 Pg C in subsoils (100 cm) globally. This represents an increase of 31% of SOC at 30 cm and an increase of 45% at 100 cm compared to the average of prior estimates (Figure 1). Our global RF model achieved an $R^2$ of 0.35 at 30 cm and an $R^2$ of 0.38 at 100 cm on the training dataset. While post-processing bias correction resulted in a notable decrease in $R^2$ at 30 cm and a smaller reduction at 100 cm, it substantially reduced bias and improved the alignment between predicted and observed SOC distributions, an essential outcome for large-scale carbon accounting. The mangrove-specific models outperformed the global model at both 30 cm and 100 cm depths ($R^2$=0.42 and $R^2$=0.47, respectively). Peatland-specific models showed lower performance in topsoil layers but outperformed the global model in subsoil layers ($R^2$=0.21 and $R^2$=0.43, respectively). As these ecosystems are known to contain higher SOC densities than other biomes, the improved predictions from the ecosystem-specific models further reinforce our findings of greater global SOC stocks.

Of leading maps, HWSD, GSDE, and WISE30sec used the taxotransfer rule approach, which links soil profiles to predefined mapping units with assigned statistical values (Lin et al., 2024). GSDE and WISE30sec expanded upon HWSD by incorporating additional soil profiles, increasing the estimated carbon stock from 588 Pg C to >700 Pg C. However, these maps use discrete representation of spatial variability and therefore have limited use for global decision making (Hengl et al., 2017; Vitharana et al. 2019). More recent maps use digital soil mapping methods, applying machine learning and environmental covariates to model SOC stocks contiguously, such as the GSOCmap, the Sanderman map, and SoilGrids250m. While Sanderman and GSOCmap did not significantly deviate from global SOC estimates (684 Pg C-GSOCmap, 869 Pg C-Sanderman), the first version of SoilGrids250m produced substantially higher SOC predictions of 1,176 Pg C (Hengl et al., 2017). Several studies have since then questioned the accuracy of this estimate, identifying overestimations of more than 30% in several regions (Mulder et al., 2016; Crowther et al., 2019; Vitharana et al., 2019; Chen et al., 2022).

A revised SoilGrids250m map was produced in 2021 with global estimates of 594 Pg C, which is around half of the previous global total (Poggio et al., 2021). This places SoilGrids250 v.2 on the lower end of global estimates. The key difference between the two SoilGrids versions was the methodological approach. The original SoilGrids map used an "interpolate first, calculate later" approach, wherein continuous maps were first produced for each soil covariate before SOC stocks were calculated by multiplying these input covariates for each pixel. This method had notable limitations, as it may propagate biases from overfitting. Hengl et al. (2017) recognized the overestimation of low values in this first version, mainly attributed to fitting issues. In contrast, the new SoilGrids250m map adopts a "calculate first, interpolate later"

approach, calculating SOC stocks at specific locations and then interpolating these values to create a more continuous map. Our map aligns with this later approach but also introduces a multi-model method, wherein critical ecosystems (peatlands and mangroves) are modelled individually. Our resulting map reveals new estimates of carbon distribution across biomes

and highlights previously underrepresented carbon hotspots.

**Figure 1. Global distribution of soil organic carbon (SOC) stocks (Pg C) at (a) 30 cm and (b) 100 cm soil depth. (c) Comparison of global SOC stock estimates from this study (soc30 and soc100) with past global assessments using digital soil mapping and taxotransfer methods. Digital soil mapping approaches include "interpolate first, calculate later" (mapping input variables before calculating stocks) and "calculate first, interpolate later" (computing stocks at points, then interpolating). SoilGrids v.2 is the latest**

**version and replaces v.1; however, v.1 is included here because v.2 was only available at 30 cm depth at the time of analysis.**

| (c) | | Digital soil mapping | | | | Taxotransfer method | | |
|---|---|---|---|---|---|---|---|---|
| | | calculate-first | | interpolate-first | | | | |
| | this study | SoilGridsv2 | Sanderman | SoilGridsv1 | GSOCv1.5 | WISE30sec | GSDE | HWSDv1.21 |
| Global (Pg C) | | 2021 | 2017 | 2017 | 2018 | 2016 | 2014 | 2012 |
| **soc30** | 1,049 | 594 | 869 | 1,176 | 684 | 883 | 794 | 588 |
| **soc100** | 2,822 | - | 1,960 | 2,769 | - | 1,969 | 1,907 | 1,130 |



### 3.1.2. Biome-specific carbon stocks

**Table 1. Biome area (Mha), soil organic carbon (SOC) density (t C/ha) and total SOC stock estimates (Pg C) at 30 cm and 100 cm soil depth, labelled soc30 and soc100 respectively. Mangrove SOC estimates are calculated using both the Olson et al. (2001) forest extent, and the more recent Global Mangrove Watch (2020) extent. Peatland SOC is based on the UNEP 2022 extent. Ecosystem-specific models were applied for mangroves and peatlands; all other biomes were modelled using a global approach and informed by the Olson et al. (2001) classification. For comparison, we also include total carbon estimates from SoilGrids version 2, Sanderman, and SoilGrids version 1 products. SoilGrids v.2 is the most recent and currently supported version, replacing SoilGrids v.1. However, we include SoilGrids v.1 in this study because, at the time of analysis, SoilGrids v.2. data were only available for the 30 cm depth.**

| Biome soc30 | this study | | | SoilGridsv2 | Sanderman | SoilGridsv1 |
|---|---|---|---|---|---|---|
| *Forests* | Mha | t C/ha | Pg C | Pg C | Pg C | Pg C |
| Tropical and subtropical moist broadleaf forest | 1,965 | 87 | 171 | 106 | 135 | 192 |
| Tropical and subtropical dry broadleaf forests | 299 | 52 | 15 | 13 | 14 | 19 |
| Tropical and subtropical coniferous forests | 71 | 59 | 4.2 | 3.8 | 5.0 | 5.8 |
| Temperate broadleaf and mixed forests | 1,254 | 96 | 121 | 72 | 104 | 138 |
| Temperate coniferous forest | 401 | 130 | 52 | 24 | 38 | 50 |
| Boreal forests / taiga | 1,435 | 153 | 219 | 100 | 181 | 308 |
| Mangroves - Global Mangrove Watch (2020) | 13 | 104 | 1.3 | 0.6 | 1.0 | 1.6 |
| Mangroves - Olson classification (2001) | 30 | 98 | 3.0 | 1.5 | 2.5 | 3.9 |
| *Grasslands and shrublands* | | | | | | |
| Tropical and subtropical grasslands, savannas and shrublands | 2,019 | 39 | 79 | 67 | 78 | 91 |
| Temperate grasslands, savannas and shrublands | 996 | 83 | 83 | 49 | 70 | 82 |
| Flooded grasslands and savannas | 106 | 66 | 7.0 | 4.2 | 5.2 | 7.6 |
| Montane grasslands and shrublands | 506 | 69 | 35 | 25 | 37 | 37 |
| *Other biomes* | | | | | | |
| Tundra | 713 | 185 | 132 | 51 | 101 | 160 |
| Mediterranean forests, woodlands and schrubs | 319 | 52 | 17 | 12 | 16 | 16 |
| Deserts and xeric shrublands | 2,787 | 38 | 106 | 64 | 72 | 59 |
| Peatlands - UNEP classification (2022) | 858 | 170 | 146 | 57 | 105 | 202 |

| Biome soc100 | this study | | | SoilGridsv2 | Sanderman | SoilGridsv1 |
|---|---|---|---|---|---|---|
| *Forests* | Mha | t C/ha | Pg C | Pg C | Pg C | Pg C |
| Tropical and subtropical moist broadleaf forest | 1,965 | 301 | 591 | - | 300 | 447 |
| Tropical and subtropical dry broadleaf forests | 299 | 168 | 50 | - | 32 | 41 |
| Tropical and subtropical coniferous forests | 71 | 219 | 16 | - | 11 | 14 |
| Temperate broadleaf and mixed forests | 1,254 | 215 | 270 | - | 219 | 291 |
| Temperate coniferous forest | 401 | 257 | 103 | - | 80 | 101 |
| Boreal forests / taiga | 1,435 | 342 | 491 | - | 409 | 791 |
| Mangroves - Global Mangrove Watch (2020) | 13 | 346 | 4.4 | - | 2.8 | 5.4 |
| Mangroves - Olson classification (2001) | 30 | 328 | 9.8 | - | 6.0 | 9.5 |
| *Grasslands and shrublands* | | | | | | |
| Tropical and subtropical grasslands, savannas and shrublands | 2,019 | 127 | 256 | - | 173 | 186 |
| Temperate grasslands, savannas and shrublands | 996 | 197 | 196 | - | 153 | 181 |
| Flooded grasslands and savannas | 106 | 176 | 19 | - | 12 | 21 |
| Montane grasslands and shrublands | 506 | 158 | 80 | - | 85 | 86 |
| *Other biomes* | | | | | | |
| Tundra | 713 | 448 | 320 | - | 237 | 413 |
| Mediterranean forests, woodlands and schrubs | 319 | 138 | 44 | - | 36 | 35 |
| Deserts and xeric shrublands | 2,787 | 130 | 362 | - | 174 | 134 |
| Peatlands - UNEP classification (2022) | 858 | 401 | 344 | - | 258 | 576 |



Our results confirm that forests store more than half of the world's SOC stocks with 584 Pg C (56%) at 30 cm and 1,525 Pg C (54%) at 100 cm (Table 1). In comparison, global grasslands store 204 Pg C (19%) at 30 cm and 551 Pg C (20%) at 100 cm. Our biome estimates differ slightly from prior analyses: notably, we find that forests hold a greater share of SOC

respective to grasslands and croplands, as compared to GSOCmap topsoil estimates, which estimated forests hold 33% of global stocks while grasslands and croplands hold 24% (FAO & ITPS, 2018).

Of climate zones, boreal and tundra regions store most of the carbon with 351 Pg C (33%) at 30 cm and 811 Pg C (29%) at 100 cm depth. Temperate regions contribute 256 Pg C (24%) at 30 cm and 569 Pg C (20%) at 100 cm. Tropical and subtropical regions contribute 269 Pg C (26%) at 30 cm and 913 Pg C (32%) at 100 cm. We align with all prior maps in

finding large SOC stocks in higher latitudes (Tifafi et al., 2018; Feeney et al., 2022). We find that the boreal forest biome (Olson biome 6) holds 21% of global stocks at 30 cm, which is greater than prior estimates of 17% in SoilGridsv2 (Poggio et al., 2021). In permafrost, we estimate 319 Pg C (30%) at 30 cm and 715 Pg C (25%) at 100 cm.

Prior global maps have largely overlooked critical peatland complexes, notably in the Congo Basin, now recognized as the world's largest tropical peatland (Dargie et al., 2017; UNEP, 2022). Similarly, the Brazilian and Peruvian Amazon

peatlands are largely absent in prior maps, despite evidence of their significant extent and carbon densities (Draper et al., 2014; Hastie et al., 2022, 2024). We estimate global peatlands including peat-in-soil mosaics hold 146 Pg C at 30 cm and and 344 Pg C at 100 cm across 858 Mha, representing 14% and 12% of the global SOC stocks. We find that Olson biome 1 which contains peatlands of the Amazon, Congo Basin and Indonesia store 16% (171 Pg C) of global SOC at 30 cm and 21% (591 Pg C) at 100 cm. Our estimates exceed the most recent global estimates, with noticeably higher values in deeper

soil layers, where we find significantly greater carbon storage—especially in tropical forests and mangrove ecosystems.

We find that soil carbon is not stored linearly with depth, showing significant variation across different biomes and regions. Our data indicate that 29% of total SOC in tropical and subtropical systems is stored in topsoil. In contrast, temperate systems store 45% of total SOC in topsoil. This indicates that while temperate systems primarily store carbon at the surface, tropical and subtropical systems may have a greater capacity for carbon storage in deeper soil layers. This

highlights the need to assess carbon distribution at depth when evaluating impacts of land use.

Our map presents slightly higher carbon values in deserts and xeric shrublands (Olson biome 13) compared to previous maps. In the first version of SoilGrids, synthetic or pseudo-observations were introduced in data-scarce regions, mainly in the Sahara and the Arabian Peninsula (Hengl et al., 2017). These points were included in SoilGrids's original training dataset but were discarded in their later version due to concerns over the reliability of pseudo-data. While deserts

consist predominantly of sand, transitional zones remain challenging to represent accurately (Hengl et al., 2017). Recent studies have identified SOC accumulation in deserts, driven by biological soil crust formation, which may marginally contribute to regional carbon fluxes (Petrie et al., 2015; Wang et al., 2021b; Lu et al., 2023). This highlights the need to better understand SOC dynamics in arid and transitional regions.

We find that significant uncertainties persist globally, and these generally fall under three categories: (1) remote

areas that are difficult to access for sampling, (2) locations with complex vegetation, limiting the utility of optical remote



sensing and (3) regions with large SOC stocks, susceptible to overfitting in global models, as seen in Figure 2 and Figure S6. We notice that aside from tundra and boreal regions, there is noticeably high uncertainty in forests of Central Africa and in mainland rainforests of SE Asia. These forests belong to Olson biome 1, which remain limited in ground samples.

**Figure 2: Uncertainty analyses at 95% confidence interval at (a) 30 cm depth and (b) 100 cm depth.**


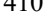



### 3.1.3. Grasslands

We estimate that grasslands store 204 Pg C (19%) of global carbon stocks at 30 cm and 551 Pg C (20%) at 100 cm.
Grasslands have gained attention recently because the processes that control carbon cycling in these systems are not well understood. Early studies indicated that substantial SOC accumulation could occur following afforestation in grasslands (Rumpel & Kögel-Knabner, 2011; Zhou et al., 2017). However, more recent research has challenged this assumption, revealing that grasses rather than trees and shrubs are often the dominant drivers of SOC in grasslands (Dohn et al., 2013; Wigley et al., 2020; Zhou et al., 2022; Coetsee et al., 2023; Zhou et al., 2023). Further, Dass et al. (2018) found that tree
cover did not consistently explain SOC levels in Californian grasslands.

Fire adds another layer of complexity to SOC dynamics in grasslands. Nearly half of tropical and subtropical grasslands and savannas (Biome 7) are vulnerable to frequent fires (49%, 982/2,018 Mha). This biome accounts for 46% of global fires, based on 2015 data (982/2,109 Mha). Fires exert considerable pressure on these ecosystems, yet the long-term impact of fire on soil carbon storage remains poorly understood. $C_4$ grasses may especially play a role in maintaining SOC
under altered fire regimes. In long-term studies, fire suppression was found to have minimal impact on subsoil SOC, where $C_4$ grass-derived carbon dominates, which is more resistant to fire (Bai et al., 2022; Zhou et al., 2022). The significant pressure of fires in grasslands may distinctly influence their carbon dynamics compared to other biomes.

### 3.1.4. Mangrove forests

Our mangrove-specific model predicts 1.3 Pg C at 30 cm and 4.4 Pg C at 100 cm based on the Global Mangrove
Watch v.3.0, year 2020, with a forest extent of 13 Mha. SOC predictions increase to 3.0 Pg C at 30 cm and 9.8 Pg C at 100 cm, using Olson's 2001 mangrove extent for Biome 14, estimated at 30 Mha. Differences in mangrove extent may be due to mangrove loss since 2001 and the resolution of the land cover classes. Our SOC values are smaller than 6.4 Pg C reported by Sanderman et al. (2018) at 100 cm and significantly higher than ~2.6 Pg C reported by Atwood et al. (2017). Sanderman et al. (2018) based their SOC estimates on a forest extent of 13.8 Mha, as reported by Giri et al. (2011). Atwood et al. (2017)
used a global mangrove extent of 8.1 Mha, as calculated by Hamilton & Casey (2016). Although Sanderman et al. (2018) estimated a similar mangrove extent to ours, their study found ~45% more carbon in mangrove forest soils globally. This may be due to either their use of a depth function to predict carbon distribution or a difference in their ground inventory samples.

Our data show substantial variation in SOC density, ranging from 66 to 140 t C/ha at 30 cm (10th and 90th
percentiles) and from 221 to 471 t C/ha at 100 cm. This highlights the considerable variability captured in our model predictions. This variation is observed among mangrove types, as indicated in Table S8. Our data indicates a global mean soil carbon density for mangrove forests of 104 t C/ha at 30 cm and 346 t C/ha at 100 cm. Our value at 100 cm is similar to





the carbon density of 361 t C/ha found by Sanderman et al. (2018). Overall, our results confirm Sanderman's findings of significantly larger SOC stocks in mangrove forests compared to the estimates previously reported by Atwood et al. (2017).

### 445   3.1.5. Permafrost

Our permafrost SOC estimates are of 319 Pg C at 30 cm and 715 Pg C at 100 cm across 1,917 Mha. This exceeds earlier estimates, including Mishra et al. (2021) at 510 Pg (100 cm), which itself marked an increase from prior estimates—472 Pg (Hugelius et al., 2014) and 170 Pg (Tarnocai et al., 2009). Permafrost is defined as land wherein soil temperatures are ≤ 0°C for ≥2 consecutive years. We used the NCSCDv2 peatland extent, which is based on the International Permafrost
Association permafrost map and was used by Tarnocai et al. (2009) (Brown et al., 1997; Hugelius et al., 2013).

Tifafi et al. (2018) noted that a large part of remaining discrepancies among global maps comes from northern latitudes. Soil C stocks remain difficult to predict as these environments feature some of the world's most variable soils, with widespread occurrence of patterned ground, i.e. symmetric landforms (Siewert et al., 2021). Because spatial SOC distribution differs substantially from phytomass C, the use of optical remote sensing may be less effective in boreal and
tundra regions than in other areas globally. Patoine et al. (2022) observed that land-cover change was a weaker driver of change in soil microbial carbon stocks in Northern latitudes, while mean annual temperature was a more critical factor (Patoine et al., 2022). Adding to this complexity, rising Arctic temperatures have led to Arctic greening with increasing vegetation biomass (Epstein et al., 2012; Siewert et al., 2021). Large-scale vegetation–climate feedback from this greening could generate substantial shifts in global soil carbon stocks (Myers-Smith et al., 2020). This highlights the challenges in
accurately mapping soil carbon in these regions, where complex environmental factors such as varying climate conditions and unique ecosystem structures contribute to uncertainties.

### 3.1.6. Peatlands

Our estimates indicate that global peatlands including peat-in-soil mosaics hold 146 Pg C at 30 cm and 344 Pg C at 100 cm across 858 Mha, representing 14% and 12% of the global SOC stocks. We find that 90% of peatland carbon at 30 cm
is located in the Northern hemisphere and 86% at 100 cm. Global average carbon density in peatlands are 170 t C/ha at 30 cm and 401 t C/ha at 100 cm. We find a wide variation in C densities across peatlands, as seen in Table S9. Our data indicate that, on average, northern peatlands are more carbon-dense than tropical peatlands. However, certain tropical peatland complexes, particularly in Central Africa and Southeast Asia, are also highly carbon-dense, especially at depth. This may reflect differential decomposition processes, impacted by climate.
To delimit peatland extent globally, we combined recent high-resolution peatland maps from UNEP (2022), Hastie et al. (2022) and Crezee et al. (2022). UNEP follows definitions established by the Convention on Wetlands (COP 8 Resolution VIII.17) (Ramsar Convention on Wetlands 2018). Peatlands are defined based on a minimum soil depth, but national thresholds vary widely from 5 to over 50 cm, affecting mapped extents. The UNEP mask prioritized studies that



used peat depth thresholds closest to 30 cm. In line with this approach, studies in the Congo Basin and Peruvian Amazon
consistently defined peat as having an organic matter content of ≥65% and a peat depth of ≥30 cm (Dargie et al., 2017).

### 3.1.7. Peatlands of the Congo Basin complex

We find that peatlands of the Congo Basin hold 5.4 Pg C at 30 cm and 18.8 Pg C at 100 cm depth, across 18 Mha (bbox: 12.128906,-5.441022,28.828125,8.233237 degrees). The extent of this ecosystem has been re-evaluated several times in the literature. Dargie et al. (2017) estimated the area at 14.6 Mha, more than five times the previous estimate by Page et al.
(2011). Crezee et al. (2022) later reported that the peatland complex covers 16.8 Mha, a 15% increase over Dargie's estimate. Our findings align with Crezee et al. (2022) and may vary slightly due to the incorporation of UNEP's (2022) raster dataset. The Congo Basin peatlands store on average 295 t C/ha at 30 cm and 1,021 t C/ha at 100 cm, and are among the most carbon-dense ecosystems on Earth (Figure 3). We note that studies have found higher average carbon densities at greater depths: 1,712 t C/ha with a maximum of 3,970 t C/ha (Crezee et al., 2022) and 2,186 t C/ha (Dargie et al., 2017). Further,
both Crezee et al. (2022) and Dargie et al. (2017) found that peat depth could reach over 5 meters. Prior studies have found that the Congo Basin is more carbon-dense than other tropical peatlands, such as those in Central Kalimantan, Borneo, and the Pastaza–Marañón Basin in the western Amazon, suggesting higher rates of decomposition in this system. Both Dargie et al. (2017) and Crezee et al. (2022) found similar carbon stocks of approximately 29 Pg C across the full peat depth. Our estimates, focused on carbon storage to 100 cm depth, align with previous findings in highlighting the Congo Basin
peatlands as a critical carbon sink.

**Figure 3. Soil Carbon Stocks (t C/ha) of the Congo Basin Peatland complex**

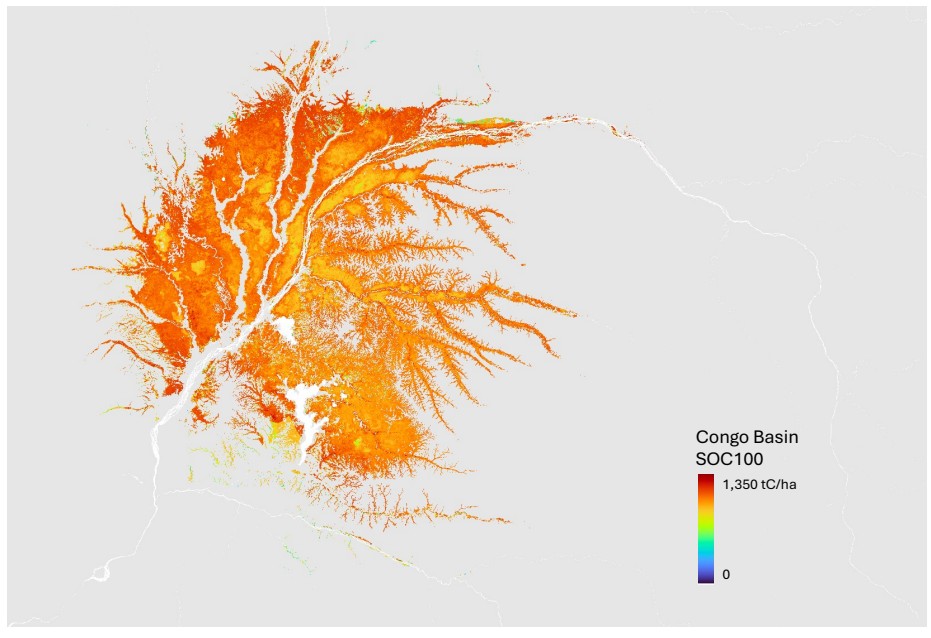



### 3.1.8. Amazon River Basin

We find that the Amazon Basin holds 40 Pg C at 30 cm and 168 Pg C at 100 cm over an extent of 593 Mha,
representing 4% and 6% of the global total, respectively (Figure 4). This contrasts with earlier estimates based on the
RADAMBRASIL project's 3,000 points collected in the 1970s. Moraes et al. (1995) area-weighed different soil types and
estimated that the Amazon Basin held 21 Pg C at a depth of 20 cm and 47 Pg C at 100 cm. Batjes and Dijkshoorn (1999)
corroborated these findings, estimating 46.5 Pg C at 100 cm for the Amazon, with the extent calculated at 500 Mha—closely
aligned with our own estimate. Nepstad et al. (1994) suggested that the Brazilian Amazon may hold up to 136 Pg C to a
depth of 8 m (Fearnside et al., 1998). However, this estimate is lower than our estimate at a depth of 1 m. These earlier
studies typically relied on average SOC and bulk density values for each soil class to estimate carbon stocks. In contrast,
Cerri et al. (1999) used the national soil map with linear regression to calculate bulk density, estimating SOC stocks to be
23.4 Pg C (30 cm) and 41 Pg C (100 cm). Gomes et al. (2019) later applied machine learning (RF), to the legacy
RADAMBRASIL soil dataset and estimated SOC stocks to be 36.1 Pg C (100 cm), which was noticeably lower than all prior
estimates. Our study suggests considerably greater carbon storage in the Amazon Basin, with estimates significantly higher
than those of earlier studies.

**Figure 4. Amazon Basin soil carbon stocks at 100 cm depth**

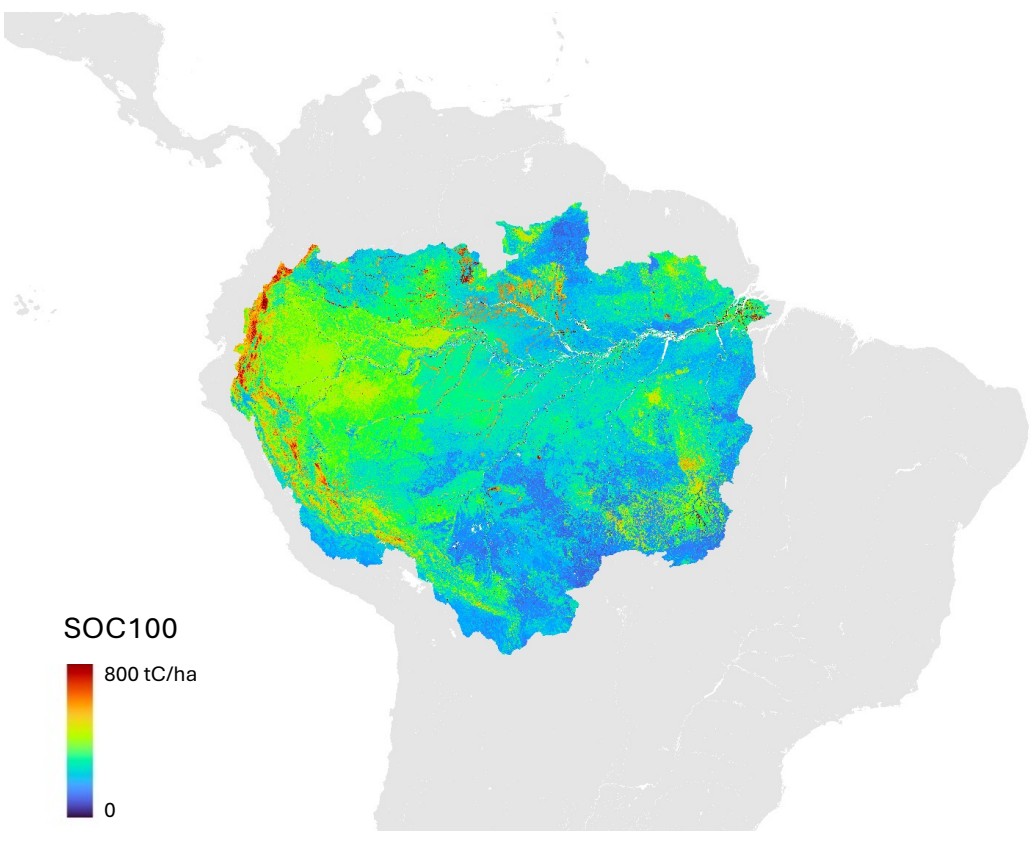





We find that Amazon peatlands hold 2.6 Pg C at 30 cm and 12.1 Pg C at 100 cm across 25 Mha. Brazil's peatlands

are interspersed across various regions whereas other peatlands, such as those in Peru's Pastaza-Marañón Foreland Basin, are more densely concentrated in the region (Hastie et al., 2022). Our estimate of the Amazon peatland extent is similar to the extent calculated by Hastie et al. (2024) of 25.1 Mha. Mapping peatlands across the Amazon Basin has largely depended on modelling approaches, primarily due to the scarcity of field data (Hastie et al., 2024). However, recent advancements have significantly enhanced the detection of peatlands (UNEP, 2022).

Our findings indicate that peatlands of the Amazon have a carbon density 1.6 times (30 cm) and 1.7 times (100 cm) greater than that of other regions in the Amazon (106 t C/ha compared to 67 t C/ha at 30 cm; 487 t C/ha compared to 283 t C/ha at 100 cm). It is known that Amazon peatlands are mostly covered by forests (66%), with pole forests representing the most carbon-dense ecosystems in Amazonia, storing up to 1,391 t C/ha in soils (Draper et al., 2014; Bourgeau-Chavez et al., 2021; Hastie et al., 2024). Overall, our measured C densities are higher than prior estimates, with Moraes et al. (1995)

estimating a mean soil C density of 103 t C/ha, and Fearnside et al. (2016) measuring ~276 t C/ha at a depth of 8 m. Compared to the Congo Basin peatlands, discussed in a previous section, the Amazon peatlands cover a 39% larger area; however, the Congo Basin holds twice as much carbon at 30 cm and 1.6 times as much at 100 cm. This is reflected in measured C densities: Amazon peatlands hold an average of 106 t C/ha at 30 cm and 487 t C/ha at 100 cm whereas the Congo Basin holds 295 t C/ha at 30 cm and 1,021 t C/ha at 100 cm. These differences reflect the possible variation in carbon

storage and density across tropical peatland systems and their important role in global carbon dynamics.

Our analysis shows higher uncertainty in the eastern Guiana and Brazilian shields, and lower uncertainty in the carbon-rich peatlands of the Peruvian and Colombian basins (Figure 5). This differs from current soil carbon maps, which commonly show high uncertainties in regions with large carbon stocks. This highlights the need for field ground truthing to improve the accuracy of soil carbon estimates and to better understand their distribution.

**Figure 5. Amazon Basin soil carbon stocks uncertainty in t C/ha (95 CI), at 100 cm depth**

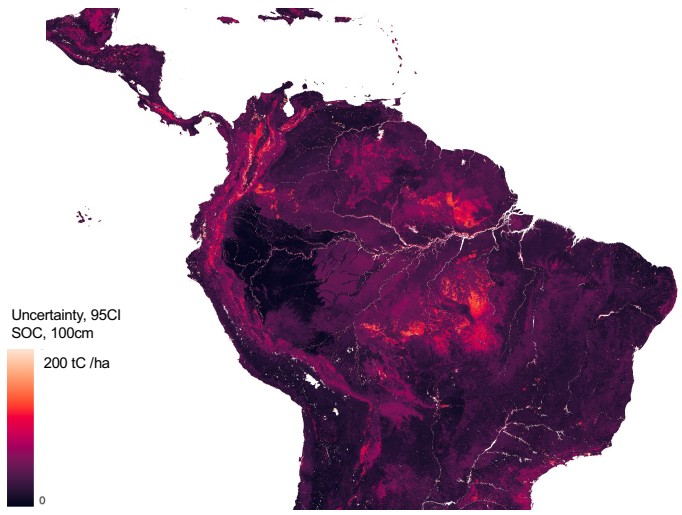



## 3.2 Land Use Impacts on Global SOC

### 3.2.1. Global fires

**Figure 6. Soil organic carbon stocks (t C/ha) of fire-prone land**

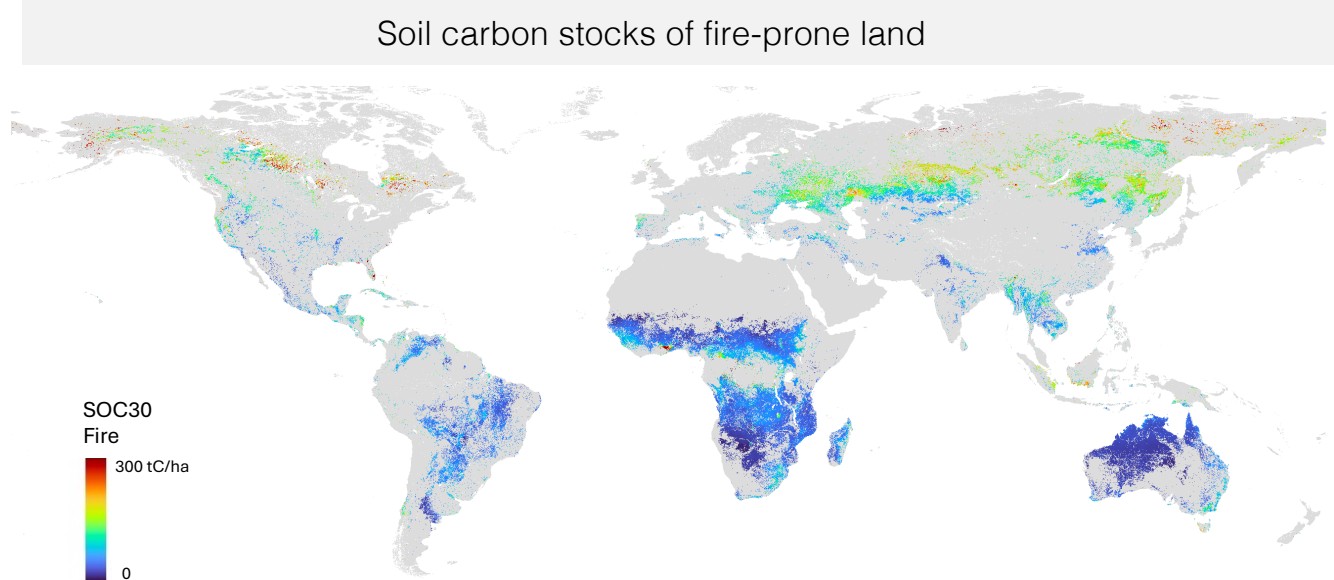


      Our results indicate that 132 Pg C at 30 cm and 345 Pg C at 100 cm across 2,109 Mha are exposed to fires annually. This represents 13% and 12% of the global SOC total. This estimate differs from prior assessments suggesting that fires affect ~70% of global topsoil carbon, based on a global topsoil carbon stock of 653 Pg C and derived from systematic reviews and regional extrapolations (Pellegrini et al., 2022). Our dataset indicates that, in the northern hemisphere, 91 Pg C

at 30 cm and 218 Pg C at 100 cm across 1,050 Mha are exposed to fires annually, whereas in the southern hemisphere, 41 Pg C at 30 cm and 128 Pg C at 100 cm are exposed in 1,060 Mha. Despite fires affecting similar land areas in each hemisphere, they expose 121% more carbon at 30 cm and 70% more carbon at 100 cm in the northern hemisphere, due to the region's carbon-dense ecosystems.

      In the southern hemisphere, fires primarily affect grasslands and savannas (Olson biome 7), whereas in the north,

they affect carbon-dense peatlands of boreal forests (Olson biome 6). In boreal peatlands, most emissions come from the combustion of dense organic soil layers, rather than biomass (Walker et al., 2020). Peatlands of black spruce (*Picea mariana*) forests, for instance, are highly flammable (Walker et al., 2020). Figure 6 shows that fires of northern latitudes often occur in carbon-rich soils, with density values exceeding 300 t C/ha at 30 cm. Fire-prone soils hold 121 t C/ha (100 cm) in the south, while they hold 1.7 times more carbon with 208 t C/ha (100 cm) in the north. Consequently, fires in the

northern hemisphere have a much greater impact on carbon fluxes.

      We trained a ML model using MODIS fire frequency data as a covariate layer. We found that the model predicted 5 Pg C more in topsoil when fire data was included (1,044 vs. 1,049 Pg C at 30 cm) and 28 Pg C less in subsoil (2,850 vs.



2,822 Pg C at 100 cm). While topsoils show a slight increase in carbon when fire data is included, our model suggests that fires ultimately reduce carbon storage when we account for 100 cm depths. Therefore, fire effects must be considered
beyond the top 30 cm of soil. Our results suggest that fires can be a significant source of carbon emissions.

Mechanisms by which fires affect soil carbon in the long-term remain largely unknown. In savannas, some suggest that fires may stabilize ecosystems by maintaining fuel access and composition (Staver et al., 2011; Wimberly, 2024). In boreal forests, fires can more than double the concentration of lignin and polyphenols in soil organic matter, improving carbon stability and its sequestration (Pellegrini et al., 2022). This principle supports the use of 'prescribed burns', where
controlled fires are used to manage ecosystem fuel sustainably. However, in savanna grasslands and broadleaf forests, frequent fires can also lead to nitrogen depletion, reducing primary productivity and thereby, limiting C sequestration (Pellegrini et al., 2018). In tropical forests, fires can generate feedback loops as they alter forest understory fuels, making forests more vulnerable to further fire degradation (Dwomoh & Wimberly, 2017; Wimberly, 2024). The impact of fire on long-term carbon stocks remains under investigation. Our model suggests that fires impact carbon sinks globally and that
carbon depletion must be studied beyond the top 30 cm of soil.

### 3.2.2. Agriculture

We find that global agricultural soils hold 140 Pg C in topsoil and 368 Pg C in subsoil across 2,104 Mha (Figure 7). This represents 13% of the global soil C stock at each depth, based on 2015 satellite imagery. While fires significantly
influence the Southern hemisphere, agriculture has a greater impact in the Northern hemisphere. In our study, agricultural soils hold 114 Pg C (30 cm) across 1,606 Mha in the north, and 26 Pg C (30 cm) across 502 Mha in the south. They hold 291 Pg C in the north and 79 Pg C in the south at 100 cm. We find that 57% (80/140 Pg C, 30 cm) and 48% (178/368 Pg C, 100 cm) of agricultural soil C comes from Chernozems, despite only representing 32% of the agricultural span (666/2,104 Mha). This highlights the importance of these soils for global food security. Notably, agricultural Chernozems of Eastern Europe
and European Russia store 49 Pg C at 30 cm and 109 Pg C across 374 Mha (bbox: 23.730469, 46.195042, 105.468750, 60.844911 degrees). In comparison, mid-latitude North American agricultural soils hold 7.5 Pg C at 30 cm and 17.2 Pg C at 100 cm across 78 Mha (bbox: -117.246094, 43.325178, -90.703125, 56.944974 degrees). This highlights important differences in the soil carbon stocks of different agricultural regions.



**Figure 7. Soil organic carbon stocks (t C/ha) of agricultural land**

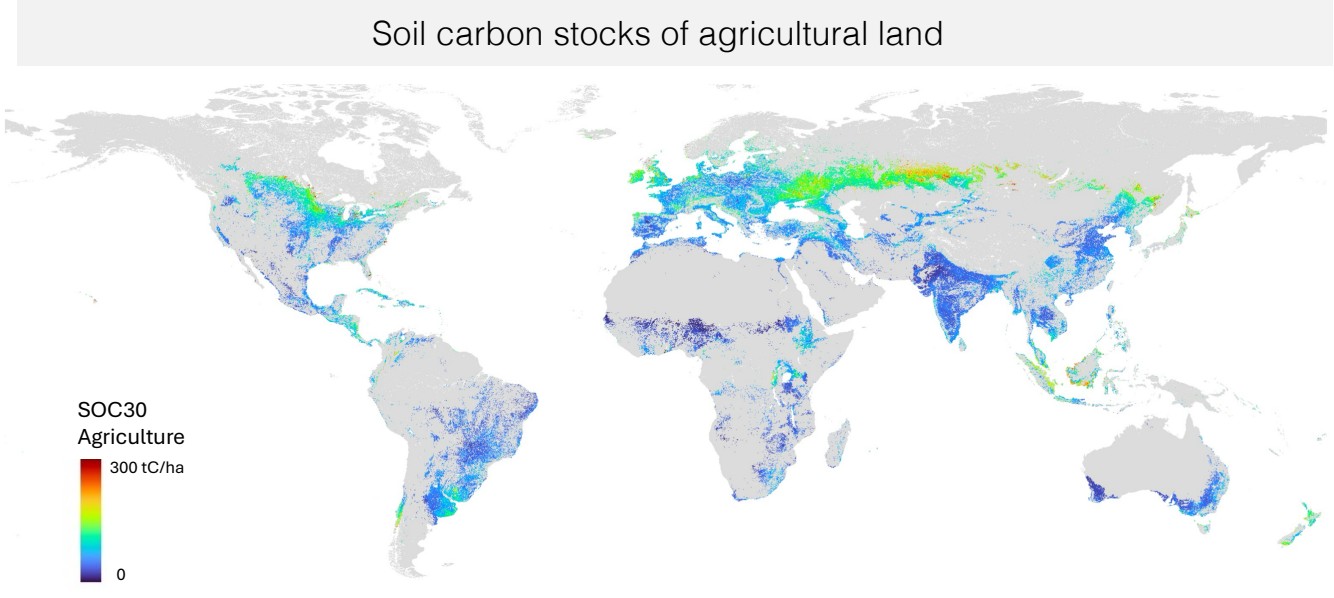


Globally, agricultural soil carbon density ranges from 31 t C/ha to 132 t C/ha at 30 cm depth (10[th] to 90[th] percentile), and from 88 to 308 t C/ha at 100 cm depth. Data indicates large variation in soil carbon density across different agricultural regions. Our map generally aligns with Zomer et al. (2017), but we predict significantly lower average stocks in cultivated peatlands of SE Asia with 167 t C/ha (30 cm) compared to their higher estimates (>400 t C/ha), using the

bounding box: 91.933594, -13.068777, 131.132813, 8.581021 degrees. Productive agricultural land in parts of Borneo and Sumatra holds on average 111 t C/ha (30 cm) and 355 t C/ha (100 cm), totalling 6 Pg C and 19 Pg C respectively across 55 Mha. In contrast, agricultural land of mid-latitude North America averages 37 t C/ha (30 cm) and 219 t C/ha (100 cm). Agricultural regions of India show lower average carbon densities of 39 t C/ha (30 cm) and 129 t C/ha (100 cm), potentially due to higher regional soil bulk density (Figure 8). However, due to its extensive agricultural area (~219 Mha; bbox:

65.214844, 8.059230, 91.054688, 33.724340 degrees), India's agricultural regions store 8.6 Pg C (30 cm) and 28 Pg C (100 cm). These regional contrasts underscore the need for spatially explicit assessments of land use activities to better understand agricultural carbon dynamics.

Small changes in agricultural SOC stocks are expected to significantly impact atmospheric $CO_2$ levels, as described in the '4 per mille' initiative launched in the Paris Climate Agreement (2015). However, the C sequestering potential of

agricultural land globally remains unknown with estimates ranging from 0.90 and 1.85 Pg C/year (Zomer et al., 2017) to ~3.5 Pg C/year (Amelung et al., 2020; Rumpel et al., 2020). These efforts are hindered by the scarcity of ground-truth samples. Although models broadly identify land cover as a key driver of SOC change globally (Stockmann et al., 2015;



Hengl et al., 2017; Heuvelink, 2021), the dynamic nature of cropping systems may limit the accuracy of static optical imagery in predicting carbon status. Better remote sensing and field data are essential to improve SOC estimates for

agricultural lands globally.

**Figure 8. Global soil bulk density (cg/cm²) of agricultural lands based on SoilGrids 2.0 bulk density gridded data (Hengl et al., 2017) and LGRIP v001 agricultural extent data (Teluguntla et al., 2023)**

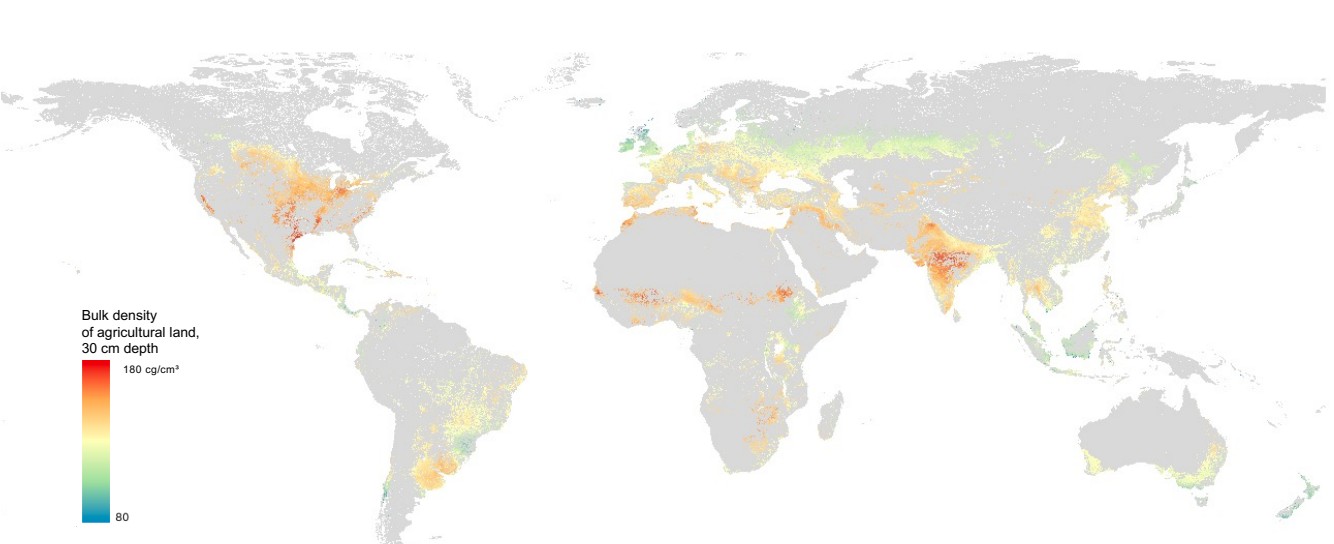

*\*Based on SoilGrids gridded product for bulk density and LGRIP agricultural extent data.*

We find that 17% of fire-prone land (351/2,109 Mha**)** is located in agricultural zones. Our analysis shows that fires on agricultural land expose 26 Pg C at 30 cm and 67 Pg C at 100 cm annually, while fires in natural systems account for 106 Pg C at 30 cm and 278 Pg C at 100 cm. The impacts of agriculture and fire on carbon fluxes are closely linked, making it difficult to isolate their individual effects. Fire and agriculture cover areas of similar scale, with less than a 1% difference in area (2,109 vs. 2,104 Mha). Their associated SOC stocks are also similar (132 vs. 140 Pg C at 30 cm and 345 vs. 368 Pg C at

100 cm). Our analysis cannot delimit transitional zones, wherein natural lands are being cleared and burnt for cropland establishment. However, we identify critical areas, wherein fire and agricultural activity may impact large C stocks, detailed in a later section. Overall, our study suggests that fire management may have greater leverage over global carbon emissions than agricultural management alone.






### 3.3. Critical Land Use Impacts on Carbon Pools across Ecosystems

### 3.3.1. Central Africa

**Figure 9. Soil organic carbon density (t C/ha) of fire-prone lands and the extent of the Congo Basin Peatland Complex**

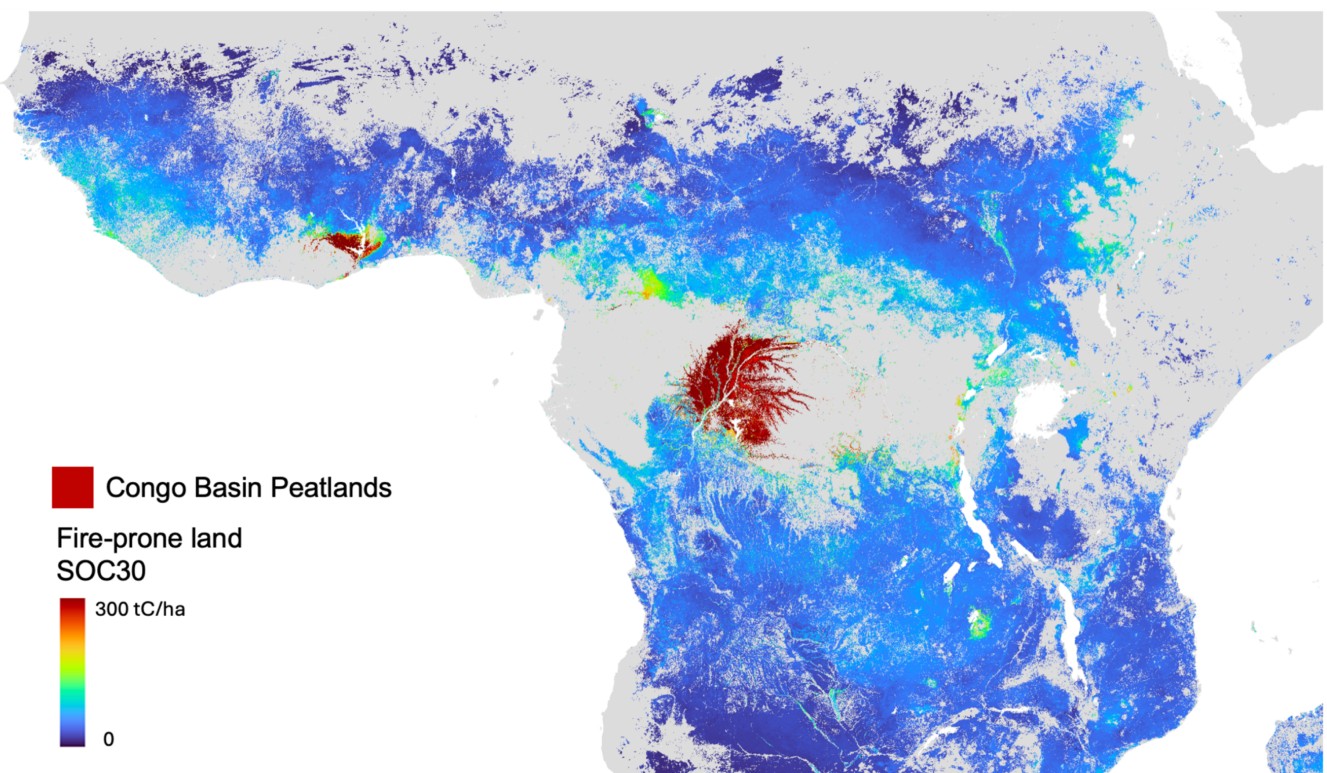

Africa experiences the most extensive biomass burning on the planet. We find that fires in Africa cover ~894 Mha on average annually and this area holds 39 Pg C at 30 cm and 116 Pg C at 100 cm (bbox: -25.136719, -34.597042, 55.722656, 38.822591 degrees). The burned area in Africa is almost equivalent in size to the entire continent of Europe. Our estimates indicate that burned area in Africa represents 42% of the global fire extent and 30% (30 cm) and 34% (100 cm) of fire-prone SOC stocks, which differs from prior estimates. Notably, previous studies have suggested that Africa contains

~2/3 of annual global burned area and that fires accounted for >1/2 of total global C emissions (Van Der Werf et al., 2017; Giglio et al., 2018; Wimberly et al., 2024). Most African wildfires occur in savannas and grasslands (Olson biome 7) (van der Werf et al., 2010; Wimberly et al., 2024). We find that fire-prone land in Africa holds low SOC on average (44 t C/ha at 30 cm and 129 t C/ha at 100 cm). However, fire activity in African also extends into tropical forests, impacting these ecosystems (Wimberly et al., 2024). Our data shows that fires impact carbon-rich soils near the Congo Basin, where carbon

densities exceed 295 t C/ha at 30 cm and 1,021 t C/ha at 100 cm. Historically, fires have been rare in tropical forests, but climate change and agricultural expansion are making these forests increasingly vulnerable to fire (Jiang et al., 2023). Figure 9 illustrates how fire activity is now reaching the edges of the basin, making it susceptible to carbon emissions.



Our findings align with previous studies, suggesting that preserving this critical terrestrial carbon sink will depend on effective management of fire activity (Jiang et al., 2023).

### 3.3.2. SE Asia and Oceania

**Figure 10. (a) Extent of fire and agricultural activity and, (b) Soil carbon density (t C/ha) at 30 cm in peatlands of Sumatra Island**

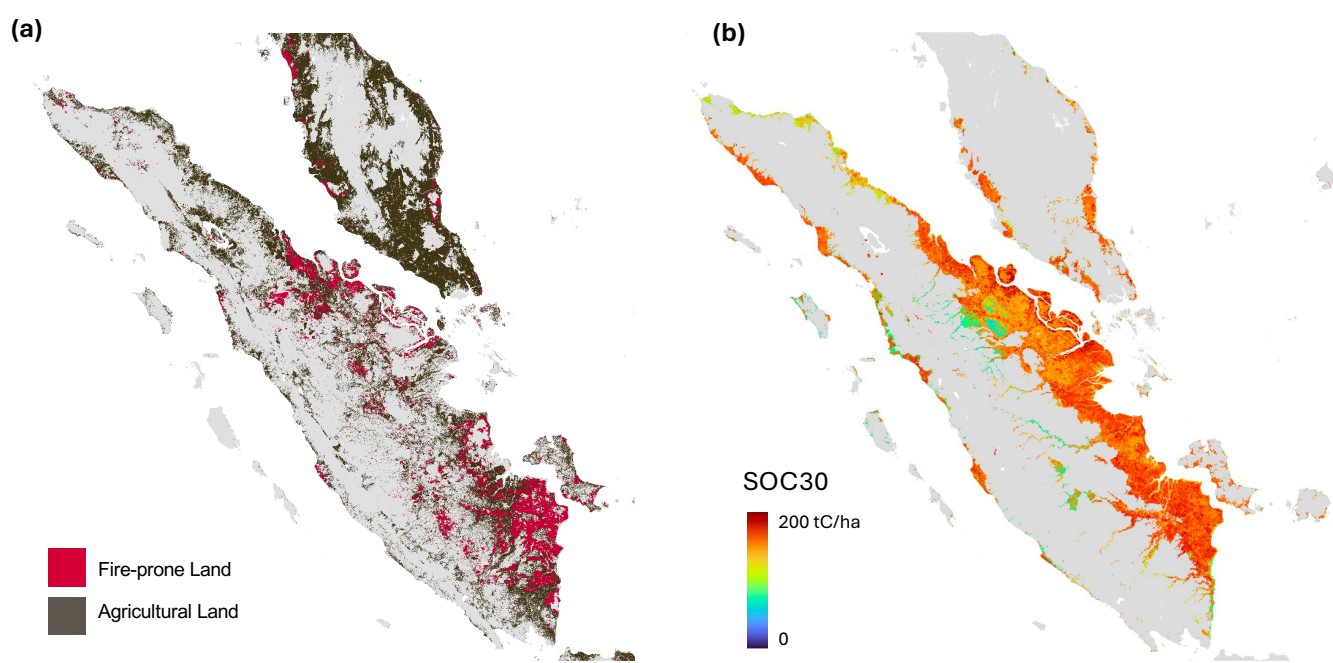

We defined the bounding box (bbox: 91.933594, -13.068777, 131.132813, 8.581021 degrees) to study key peatland regions within SE Asia and Oceania. This region holds approximately 32 Mha of peatlands with 5 Pg C (30 cm) and 17 Pg C (100 cm). Within this area, peatlands store an average of 160 t C/ha at 30 cm and 539 t C/ha at 100 cm, compared to the regional average of 110 t C/ha and 367 t C/ha at the same depths, respectively. Prior research reports higher SOC values under specific conditions; for instance, Warren et al. (2017) found that SE Asia peatland density can exceed 1,000 t C/ha, with values reaching >7,500 t C/ha in thick peat of >12 meters. Similarly, Siregar et al. (2021) measured 621 t C/ha at 100 cm in peat swamp forests. Overall, tropical forested peatlands in SE Asia are estimated to store between 42 and 55 Pg C when accounting for full peat depths (Jaenicke et al., 2008; Hooijer et al., 2010). Although our study focuses on the top 100 cm, it is important to note that peat depths can range from 0.5 to over 20 meters, emphasizing the vast carbon storage potential of these ecosystems beyond the sampled depth.

Within peatlands of our study area, agriculture occupies 16 Mha, and fires affect 7 Mha with an overlap of 5 Mha, meaning that 56% of these peatlands (18/32 Mha) are under pressure by either agricultural expansion or fire (Figure 10). We find that 1.2 Pg C (30 cm) and 3.7 Pg C (100 cm) are prone to fires and 2.7 Pg C (30 cm) and 8.8 Pg C (100 cm) to agricultural activities. Thereby, more than 20% of peatland carbon stocks are exposed to significant land use activities in this





region. This impact is notable, as prior research estimates that 30–50% of SE Asian peatlands have already been lost due to land-use change and drainage (Hooijer et al., 2010). Fires are often used to clear and prepare land for agriculture in this region (Langner et al., 2007; Warren et al., 2017). This process drastically alters peat organic composition, physical
properties, and carbon reserves (Tonks et al., 2017).

Deforested peatlands of SE Asia are often converted to industrial and smallholder plantations, particularly oil palm (*Elaeis guineensis*) and acacia (*Acacia spp.*) productions (Warren et al., 2017). Our findings further confirm that agricultural activities place substantial pressure on the region's natural peatlands. In addition to peatlands, coastal mangrove forests are also increasingly affected by agricultural expansion and land-use change. Figure 11 illustrates that smallholder plantations
create a mosaic of diverse land types, unlike the uniform layout of industrial plantations (Descals et al., 2021). While industrial plantations are relatively well-mapped worldwide, smallholder farms remain largely unmapped due to limited high-resolution data (Descals et al., 2021). Coarse data may largely overlook or underrepresent important landscape changes associated with smallholder farming, emphasizing the need for high-resolution analyses to capture these shifts and the impacts on both peatland and mangrove ecosystems.


**Figure 11. (a) Extent of mangrove forests and agricultural land within mangrove areas; (b) Soil carbon density (t C/ha) at 30 cm depth of mangrove forests in Myanmar**

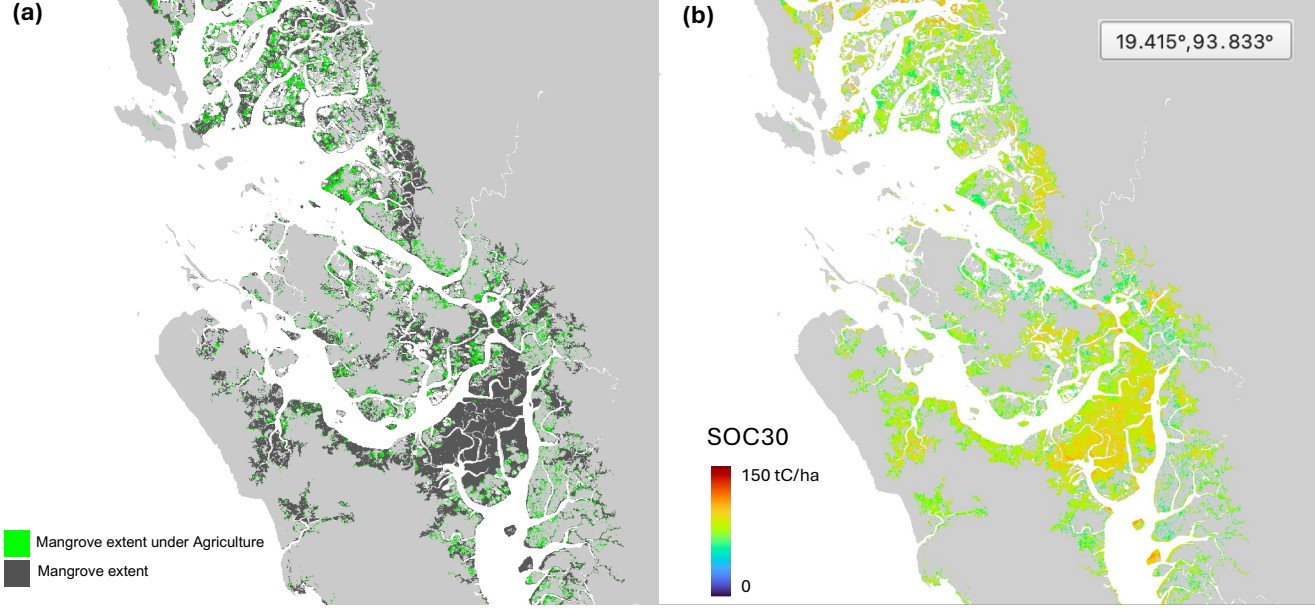





### 680    3.3.3. Cerrado Biome of Brazil

**Figure 12. (a) Extent of fire and agricultural land use in the Cerrado biome of Brazil; (b) Soil carbon density (t C/ha) at 30 cm depth in the Cerrado**

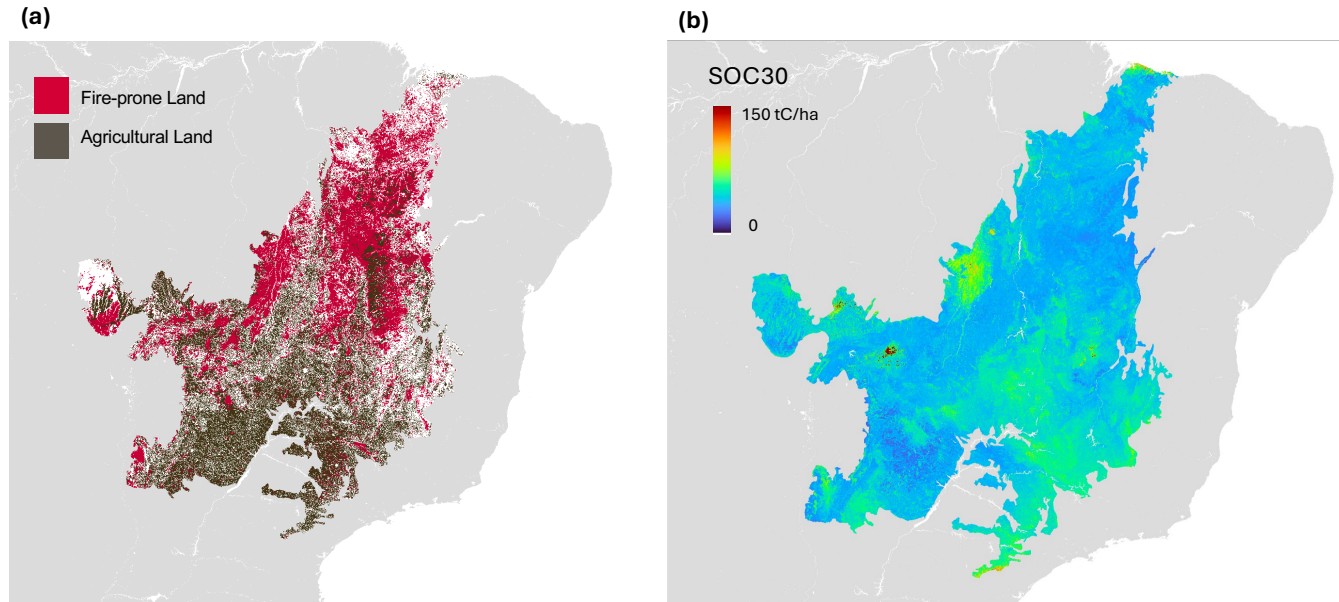

In the Cerrado biome of Brazil, our model predicts 8.4 Pg C at 30 cm and 32 Pg at 100 cm across 204 Mha. Our
data indicates that 35% of the Cerrado is used for agriculture (72/204 Mha), and 38% is affected by fire (77/204 Mha), with
over half of the biome (63%) influenced by either agriculture or fire (Figure 12). These activities overlap across 20 Mha. In
this biome, 3.2 Pg C (30 cm) and 11.5 Pg C (100 cm) are prone to fires and 3 Pg C (30 cm) and 10 Pg C (100 cm) to
agricultural activities. Therefore, 64% (30 cm) and 58% (100 cm) of carbon stocks are exposed to significant land use
activities in this region. Our map highlights high fire activity in the Matopiba region, likely linked to land clearings. Known
as Brazil's "last frontier" for agricultural expansion, Matopiba is experiencing rapid growth in farming activities (Hershaw &
Sauer, 2023). This emphasizes the need to study fire and agricultural impacts together to understand their combined effects
on SOC dynamics. However, current datasets do not differentiate between agricultural and naturally occurring fires, leading
to uncertainties about their ecological impacts in these areas.

At 100 cm depth, undisturbed areas (unaffected by agriculture or fire) store, on average, 18% more carbon per
hectare (174 t C/ha) than disturbed areas (147 t C/ha) in the Cerrado biome. In contrast, this difference is minimal in the
topsoil, where undisturbed areas hold about 2% more carbon density (42 t C/ha compared to 41 t C/ha). While these
differences are relatively small, they suggest an interesting pattern where the effects of fire and land use may become more





apparent deeper in the soil profile. Fire is often assumed to reduce soil carbon mainly by burning organic material in the
topsoil. However, it can also lower soil carbon by decreasing inputs from deeper roots, as tree loss from fire reduces overall
root biomass (Jackson et al., 2002; Pellegrini et al., 2020). Overall, field studies examining fire impacts on soil carbon show
mixed results, with some reporting no significant carbon loss in deeper soils, even under long-term burning (Coetsee et al.,
2010; Pellegrini et al., 2014). Our data, accounting for spatial variation, suggests that fire impacts on soil carbon in the
Cerrado may be mostly noticeable in deeper soils.

**3.3.4. Chernozems**

**Figure 13. Soil organic carbon stocks (t C/ha) of global Chernozems**

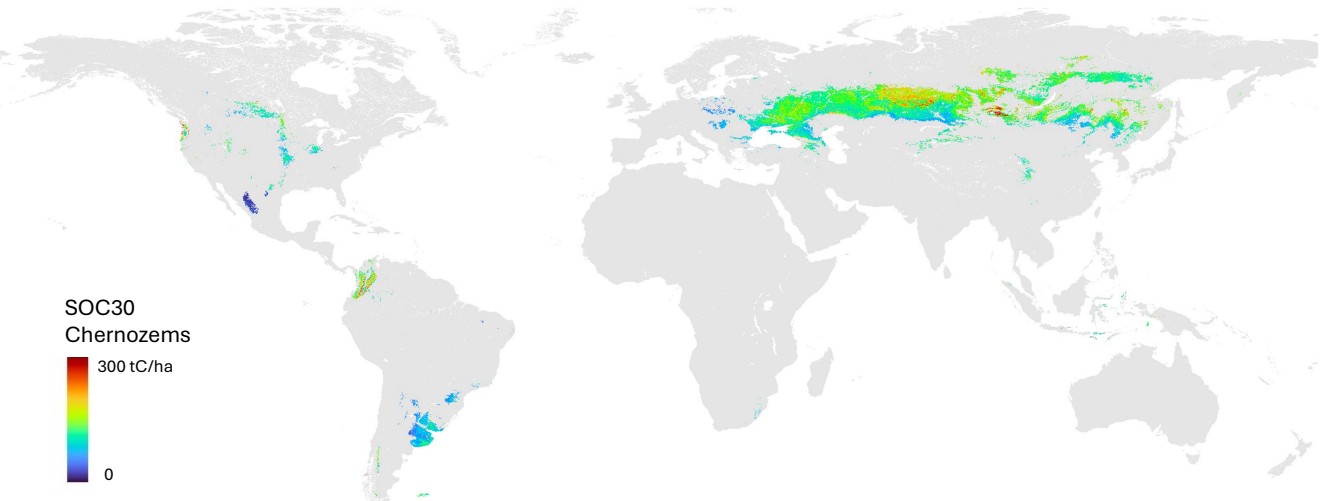

Our study shows that Chernozems (= Mollisols in U.S. Taxonomy, =value 7 in SoilGrids 2.0 WRB classes) store 80
Pg C at 30 cm and 178 Pg C at 100 cm across 666 Mha, which is 7% of the global stock of 1,049 Pg C (30 cm) and 8% of
2,822 Pg C (100 cm) (Figure 13). Our results are similar to the FAO's recent estimate of 8.2% (FAO, 2022), although the
FAO estimate was based on a lower global SOC stock, leading to a smaller total carbon value. Chernozems are important
carbon sinks due to their depth and long-term storage of carbon (Kurganova et al., 2019; Sorokin et al., 2021; Lal, 2021;
Labaz et al., 2024), but they are vulnerable to land use changes. A significant portion of Chernozems is used for agriculture,
and it is estimated that the cultivation of these soils could reduce their SOC by around 30%, with losses extending beyond 50
cm depths (Lal, 2021). In Russia, the practice of cropland burning is well-documented and presents a significant source of
carbon emissions from Chernozems (Soja et al., 2004; McCarty et al., 2012). These fires are also linked to forest fires,
compounding carbon release. Our data shows that nearly half (47%, 315 Mha) of Chernozem soils are cultivated for
agriculture, exposing 37 Pg C at 30 cm and 86 Pg C at 100 cm. Additionally, 181 Mha (27%) of Chernozem soils are
impacted by fires, which represents a carbon stock of 23 Pg C at 30 cm and 51 Pg C at 100 cm. The FAO reports that
Chernozems' carbon sequestration potential accounts for about 10% of the global total (FAO, 2022). Our findings emphasize



that Chernozems remain a critical carbon sink, and their conservation may depend on sustainable fire and agricultural management practices.

### 3.4. Map application and future development

### 3.4.1. Current Status of Global Datasets

We note that there are trade-offs between spatial coverage and temporal accuracy in current soil carbon datasets. We combined data from multiple sources, with the largest datasets, ISCN and WoSIS, contributing 19% and 16% of the total ground truth samples, respectively (Figure S1 and S2). Most global datasets were collected after 1980, coinciding with the use of satellite data. However, the WoSIS-2023 dataset includes SOC samples from several decades, spanning from 1950 to 2018, with varying coverage across countries. For example, the US contributed large datasets from the 1960s, while

European countries like France, Germany, and the UK mostly have data collected after 2010, tied to the EU-LUCAS campaigns. In this study, we used the most recent regional data to enhance the global datasets. Specifically, we included recent data from Crezee et al. (2022) collected in the Congo Basin between 2018 and 2020, while the WoSIS data for this region mostly dates back to the 1990s. We also added the latest MapBiomas dataset for Brazil (MapBiomas, 2023). While Guevara et al. (2020) found that dividing regional datasets by decade led to increased uncertainty in SOC predictions, our

approach combines global samples into a single, more representative dataset.

        The ground truth sample distribution across global and regional datasets is uneven. At 30 cm depth, ~51% of the data points come from North America, ~16% from South America, ~7% from Africa, ~22% from Europe, and only ~0.5% from Southeast Asia. Some regions, like Russia, are poorly represented, with only ~1% of the ground truth samples, and India holds ~550 samples. Hengl et al. (2017) noted that global datasets tend to be biased towards agricultural areas,

with a significant portion of data coming from developed countries like the United States. Our map highlights regional uncertainties, particularly in natural ecosystems. Carbon-rich areas, such as peatlands in Borneo, Alaska, and Florida, show high uncertainty (>20 t C/ha) indicating the need for more field monitoring in these regions (Figure 2).

### 3.4.2. Country-level ground truthing and application

        Over the past decade, increasing involvement in digital soil mapping at the country level has significantly

contributed to ground truthing global datasets and reduced uncertainty (Chen et al., 2022). Across Europe, national-scale efforts in the Netherlands (Knotters et al., 2022), Denmark (Adhikari et al., 2014), France (Meersmans et al., 2012; Mulder et al., 2016), and Scotland (Poggio and Gimona, 2014) have made notable advances, alongside continental-scale efforts for Europe (Orgiazzi et al., 2018). In North America, mapping efforts in Canada (Gonsamo et al., 2022; Sothe et al., 2022), the USA (Soil Survey Staff RaCA, 2013), and Mexico (Guevara et al., 2020) have further improved soil carbon estimates.

There remain critical data gaps in certain parts of the world, notably in parts of South America, Africa and Asia. The emergence of important datasets from Brazil (MapBiomas, 2023), Chile (Padarian et al., 2017; Dinamarca et al., 2023),



Guatemala (Vásquez-Toxcón et al., 2023), Ecuador (Armas et al., 2022), Tanzania (Kempen et al., 2019), Cameroon (Silatsa, F. et al., 2020), South Africa (Venter et al., 2021), Madagascar (Ramifehiarivo et al., 2017), and China (Yang et al., 2023) are deeply valued in addressing these gaps. We included the datasets that were openly accessible in our study. These
new contributions, particularly from regions with limited coverage, are essential for improving global soil maps and further improving our understanding of global carbon storage.

### 3.4.3. Critical data gaps for soil carbon stock measurement

Global estimates of soil carbon stocks are constrained by limited data on soil volume. While global products for
bulk density and soil depth exist, their resolution and accuracy remain low due to sparse ground truth observations (Hengl et al., 2017; Shangguan et al., 2017). Errors in bulk density measurements can impact carbon stock calculations (Hengl et al., 2017; Poeplau et al., 2017), and there are regional differences, with higher bulk densities found in northern latitudes, and in parts of South America and SE Asia (Hengl et al., 2017). Soil depth to bedrock also varies globally. As a result, current carbon stock estimates may underrepresent total stocks in some regions. Better understanding of soil volume and
composition is essential to improve our knowledge of this important carbon pool.

### 3.5. Data availability

The global SOC maps from this study (30 cm and 100 cm depths – both at 100 m resolution), along with the uncertainty maps, are hosted on a Zenodo repository. These data will be made publicly available upon final publication. The repository is accessible through the reviewer link (Creze et al., 2025):
https://zenodo.org/records/15391412?preview=1&token=eyJhbGciOiJIUzUxMiJ9.eyJpZCI6ImNjZjk2YTM5LTQyM2ItND ZiMC1iY2RlLTg0ZTA1ZjU1MDZjNSIsImRhdGEiOnt9LCJyYW5kb20iOiIxMjc1YmI0OTZhOTNiMmQyNTIxYjYyNzRi M2ZlZjBmMyJ9.M5VUSwR4GkeoKV1Kno1v3b3qLUAzErns1Zh6u0om2HhVDrnxcjKJS3WCOVAoJlSyxt-5Kbc809apXwYmAnMqyQ

### 3.6. Conclusion

Our study builds on previous work and integrates global datasets to create the most comprehensive map of global soil carbon stock distribution to date. These maps provide key insights into areas with high SOC stocks that need protection, regions vulnerable to land-use activities, and remote areas with high uncertainty that require further field-based monitoring. Our findings highlight the importance of conserving, restoring, and sustainably managing peatlands. Protecting these carbon-rich ecosystems is critical as climate change and biodiversity loss are increasingly interconnected. Additionally, our results
emphasize the need for high-resolution, multi-season SAR and optical imagery, combined with field data, to improve global carbon estimates and support the conservation of these vital ecosystems.





**Author contributions**

Conceptualization and data collection: C.M.C. and S.S.; Data processing: C.M.C and N.K.; C.M.C., S.S., N.K., Y.Y. and
S.L. contributed to global mapping, uncertainty analysis and validation; C.M.C. wrote the paper with inputs from all of the
authors; Funding acquisition: S.S.

**Competing interests**

The authors declare that they have no conflict of interest.


**Acknowledgements**

        This work was made possible by the major data contributions of ISRIC (International Soil Reference and
Information Center), ESDAC (European Soil Data Center), Food and Agriculture Organization of the United Nations (FAO),
Intergovernmental Technical Panel on Soils (ITPS), United Nations Environment Programme (UNEP), MapBiomas, and the
International Soil Carbon Network (ISCN), along with the valuable input from the broader scientific community whose
research and open data resources supported this study. This work has been supported by grants from Grantham and High
Tide Foundations.

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
