# Peer review of "High-resolution global map (100 m) of soil organic carbon reveals critical ecosystems for carbon storage"

_Earth System Science Data, 2025_

## Author Comment (AC1)

**Reviewer 1:**

This study leveraged a rich collection of global soil organic carbon (SOC) data to create an updated, high-resolution map of the spatial distribution of SOC storage at the top 30 cm and 100 cm. The motivation for developing a spatially resolved, high-resolution soil C map for better land management, C accounting, and global C monitoring is strong. The resulting map can be valuable to the urgent need of accurate monitoring and estimation of soil carbon removal impacts through climate solutions around agricultural and natural ecosystems. Overall, this work is a good contribution to the soil community, but I have a few concerns about the delivery of results and the writing. Therefore, I recommend this manuscript to be published on ESSD if a major revision can fully address the main concerns.

**First, the good things:**

I appreciate that the authors harmonized and cleaned up a massive collection of layered soil data rigorously. For example, they used only samples with bulk density measurements, adjusted measurement values according to sampling protocols, and filtered out redundant points. The authors dealt with SOC stock calculations and layered data at different depths responsibly.

The authors also fitted separate models for peatland and mangroves, which seems reasonable. This approach implies an assumption that relationships between SOC storage and predictor layers are different for these two ecosystems compared to the rest of global ecosystems.

**Author Response:**

Thank you for reviewing our manuscript and sharing your feedback. We hope our revisions address your questions and concerns.

**Major concern 1-1:**

My first major concern is that the entire article showed lots of summarized total SOC storage at biome, regional, and global level, but none of these estimates come with an uncertainty range or confidence interval in the main text or the supplemental material. Although pixel level uncertainties were shown as main figures, the authors should still report uncertainties on the estimated global and regional SOC stocks in the abstract and throughout the result sections. This should be easy given the authors already fitted 20 models from bootstrapped training data. The CIs (and bootstrap distributions) can be derived once the regional and global

estimates were calculated from each of the 20 models. The lack of regional uncertainties makes some of the conclusions unconvincing.

**Author Response:**

Thank you for your comment regarding the need to report uncertainties for global and regional SOC stock estimates. We calculated the mean and total SOC stock for each region from our 20 bootstrapped model outputs, and computed corresponding uncertainty measures, including model variance and residual variance, following established zonal inference methods (Xu et al., 2017; McRoberts et al., 2019; 2022). These regional uncertainties are reported in the revised manuscript.

We also added the following section to the methodology:

**2.10 Uncertainty assessment**

"We assessed uncertainty in our SOC estimates by quantifying errors at both pixel and regional scales, incorporating uncertainties from model development and prediction. At the pixel level, uncertainty reflects both model residuals defined as the differences between observed and predicted values and parameter-estimation variability due to the finite size of the training dataset. To capture these uncertainties and their spatial variability, we implemented a bootstrap resampling framework: in each iteration, the SOC dataset was randomly split into 70% training and 30% testing sets, and 20 independent global Random Forest models (300 trees) were trained to generate separate SOC maps. Pixel-level uncertainty was quantified as the standard deviation across the 20 model outputs, and 95% confidence intervals were computed. This approach captures the inherent variability of the global SOC dataset, enabling evaluation of heteroscedasticity in model performance. For both 30 cm and 100 cm depths, we produced two uncertainty maps: pixel-based standard deviation and 95% confidence-interval maps (Fig. 4 & Fig. S4). At the regional scale, total uncertainty was derived from the 20 bootstrapped SOC prediction maps for each depth by summing model variance and residual variance following zonal inference approaches developed in previous studies (Xu et al., 2017; McRoberts et al., 2019, 2022)."

**References:**

Xu, L., Saatchi, S. S., Shapiro, A., Meyer, V., Ferraz, A., Yang, Y., Bastin, J.-F., Banks, N., Boeckx, P., Verbeeck, H., Lewis, S. L., Muanza, E. T., Bongwele, E., Kayembe, F., Mbenza, D., Kalau, L., Mukendi, F., Ilunga, F., and Ebuta, D.: Spatial Distribution of Carbon Stored in Forests of the Democratic Republic of Congo, Scientific Reports, 7, 15030, https://doi.org/10.1038/s41598-017-15050-z, 2017.

McRoberts, R. E., Næsset, E., Saatchi, S., and Quegan, S.: Statistically rigorous, model-based inferences from maps, Remote Sensing of Environment, 279, 113028, <a href="https://doi.org/10.1016/j.rse.2022.113028">https://doi.org/10.1016/j.rse.2022.113028</a>, 2022.

McRoberts, R. E., Næsset, E., Liknes, G. C., Chen, Q., Walters, B. F., Saatchi, S., and Herold, M.: Using a Finer Resolution Biomass Map to Assess the Accuracy of a Regional, Map-Based Estimate of Forest Biomass, Surveys in Geophysics, 40, 1001–1015, https://doi.org/10.1007/s10712-019-09507-1, 2019.

**Major concern 1-2:**

For example, in section "3.2.1 Global fires", the authors concluded that adding fire as a predictor layer increased global top 30cm SOC but decreased top 1m SOC. However, the difference for SOC at 1m is only 1% of the total estimated SOC (28 out of 2822, line 552-553). I don't know whether the CIs of the estimated 2850 PgC and 2822 PgC overlap with each other. If so, this evidence can't convince me that "fire can be a significant source of carbon emission".

**Author Response:** We thank the reviewer for this comment. Following your suggestion, we have revised the discussion of fire-related effects in the manuscript. Specifically, we have made the following updates:

- We have reorganized Section 4.2 Land-use and disturbance to highlight three main aspects: (i) how fire overlaps with SOC distribution, (ii) the influence of the fire feature on SOC predictions based on partial dependence analysis, and (iii) differences in SOC response across biomes.
- 2. We no longer suggest that fire is a major source of global carbon emissions. Instead, we emphasize how fire overlaps SOC stocks regionally, with peatlands showing the largest fire-related effects, and PDP results indicating changes of 37 t C/ha at 30 cm and -58 t C/ha at 100 cm depth.
- 3. Total uncertainty combining model variance and residual variance has been computed and added to the manuscript.

**Major concern 1-3:**

Another concern about this paragraph (line 551-555) is that, although adding fire as a predictor seems ecologically reasonable, I'd still expect to see the authors show that the model performance increased after adding fire as a predictor, in order to make that final conclusion. Without proving fire as a statistically justifiable predictor, this result might point to some artefacts rather than an improvement in the model due to fire data. Another way to look into fire's effect is to predict the global SOC stock from the fitted model (with the fire layer), in a scenario where no fire happens on Earth, and then compare the no-fire prediction to global SOC stock estimate with real fire frequencies (Perhaps this is what you did. Am I misunderstanding?).

**Author Response:**

We thank the reviewer for this comment. To address this concern, we have revised the **Methods section 2.2**. to clarify the role of the fire layer. The updated manuscript now reads:

"For each biome model and soil depth, we compared the model performance (R²) on the full dataset with and without the fire variable. Including fire as a predictor had a negligible to very small effect of ~0.01 on model performance across biomes and soil depths. The minimal change in R² suggests that adding fire does not introduce artifacts or model instability. Thus, we retain fire as a predictor because it captures an important ecological process influencing soil carbon dynamics."

Regarding the second point, we no longer compare global stock predictions with and without fire as an input layer to the model. Instead, we focus on the results of the fire feature effects on SOC predictions per biome, and find large fire effects in tundra/boreal regions and in peatlands, consistent with known ecological patterns.

**Major concern 2:**

It seems like land cover is not an explicit predictor in the model. It gives me two questions.

(1) It is hard for me to understand that without this predictor, how does the model discern different SOC contents between a forest and a nearby cropland that shares essentially the same environmental condition? Perhaps you expect the Landsat bands to give such information implicitly?

**Author Response:**

We appreciate the reviewer's comment. The Landsat 8 bands we used (Red, NIR, SWIR1, and SWIR2) capture spectral information related to vegetation and land cover, providing an implicit signal that distinguishes forests, croplands, and other land cover types. Even without an explicit land cover layer, these bands allow the model to differentiate areas with similar environmental conditions (e.g., climate, elevation) but differing in vegetation type or management.

(2) The peatland and mangrove models were fitted by points within their remote sensing extents. Is it possible to find ground-truth information on whether a soil sampling belongs to peatland or mangroves? Because small-scale farming can make patchy drained lands within such systems (as you mentioned), it is likely that some points within the peatland/mangrove

extent weren't in fact peatland or mangrove, raising potential bias to the biome-specific models. It would be great if the authors could validate the peatland/mangrove extent with their ground truth land cover—if you have some site-level information from the data source, what proportion of peatland/mangrove sites were and weren't included in the peatland and mangrove masks? And how many sites that weren't peatland/mangrove end up within the masks?

**Author Response:** Thank you. We have added further clarification in **Section 2.5** of the Methods:

"To minimize the inclusion of non-mangrove or non-peatland sites, we relied on the most accurate and spatially explicit global datasets available: the Global Mangrove Watch v3.0 (2020) (Bunting et al., 2022) and the Global Peatland Assessment Database v2 (GPM2.0, 2022) from UNEP, both of which account for small-scale land use changes such as local farming and deforestation."

"While not all SOC ground-truth points had land cover metadata, 58% of points within the mangrove extent at 30 cm (71% at 100 cm) were explicitly identified as mangrove, and 14% of points within the peatland extent at 30 cm (35% at 100 cm) were identified as peatland."

Although some uncertainty remains in the land cover data, using high-quality remote sensing masks for mangroves and peatlands along with the best available ground-based data provides a reliable basis for biome-specific model training.

**Major concern 3:**

The soil data were sampled at different times spanning 1950s to 2020s, but the goal of this work is to represent a static map of global SOC stock distribution. It is common and maybe justifiable to use all historical data for mapping a static global SOC stock distribution, but I wish the authors could extend more discussion on how they expect readers to use this data product. The authors motivated this work with the recent need for soil carbon removal and land management, but how will this map of static SOC stock estimation help with these motivations, building upon the previous global SOC maps? I was excited to read an insightful reasoning of how this work is needed in the context of climate solution, policy making, and land management (which was pitched as the motivation), but I find it lacking in both the introduction and discussion. I hope the authors can elaborate more explicitly and clearly on the real impact and value of this data product, acknowledging that land use management and carbon removal programs would deal with SOC change over a short period of time (5-10 years). Some relevant literature on the perspective of natural climate solutions and carbon

removal include https://www.nature.com/articles/s41558-024-02182-0 and https://www.nature.com/articles/s41467-023-44425-2.

**Author Response:**

Thank you. We appreciate the reviewer for raising this important point. To address it, we have added the following sections to the manuscript.

**4.3.1 Baseline map for research and monitoring**

"We present a high-resolution (100 m) soil organic carbon map at two depths, representing baseline SOC stocks circa 2022. Soil profile data collected from the 1950s to the 2020s were combined with contemporary remote sensing covariates, using Landsat 8 bands to capture landscape conditions around 2022. The use of long-term soil datasets aligns with previous global mapping efforts, such as SoilGrids v1-v2 and GSOCmap v1.5, which integrate multi-decadal data to capture large-scale spatial patterns. The primary goal of this work was to represent SOC variation across diverse landscape gradients with a resolution fine enough to capture local differences while maintaining global coverage. The resulting map can support applications in land management, nature-based climate solutions, and future assessments of SOC dynamics.

SOC data inevitably span decades, while Earth observation covariates reflect more recent conditions. In landscapes without major human disturbances, SOC is assumed to remain relatively stable over time, so historical soil measurements can reliably represent baseline stocks when combined with current remote sensing data. In areas affected by landuse change, degradation, or recovery, such as forests regrowing after wildfire, agricultural rotations, or restored wetlands, SOC may have changed since sampling. For instance, samples from western CONUS encompass forests at different successional stages recovering from events such as wildfires or storms. Remote sensing captures spectral signals related to land cover and vegetation recovery, helping to contextualize historical soil measurements. Consequently, the SOC map reflects not only static soil conditions but also the imprints of temporal changes across landscapes. To address potential temporal discrepancies at the pixel level, we generated an uncertainty map alongside the SOC estimates, allowing users to assess confidence in areas where SOC may have changed substantially since sampling.

Our dataset integrates both natural and human-modified landscapes, including agricultural areas, managed forests, wetlands, and coastal ecosystems such as mangroves. By combining long-term soil measurements with remote sensing covariates and machine learning models, the map captures both spatial variation in SOC and the influence of historical land-use changes and recovery processes. Overall, the high-resolution SOC maps represent a valuable tool for carbon management and conservation. The maps enable fine-scale detection of spatial variability, supports assessments of land-use impacts, and provides a

foundation for monitoring SOC dynamics over time, even in regions with limited historical soil data."

**4.3.2 Policy and management applications**

"The baseline SOC map serves as a benchmark with broad relevance across scientific, management, and policy applications. For carbon stock reporting, it enables estimation of SOC across regions, land cover types, and management units, with particularly high precision for areas larger than 1 hectare where pixel aggregation reduces uncertainty. At the policy level, the map provides national and sub-national jurisdictions with a means to assess soil carbon storage and sequestration potential, accounting for spatial variability across land uses and vegetation successional stages. Such information supports the design and implementation of land-use and restoration policies aimed at improving soil carbon conditions. The map also facilitates participation in carbon markets, where project developers require reliable SOC stock data to estimate emissions reductions or removal factors, including avoided losses and potential accumulation rates in disturbed versus natural areas. In addition, the high-resolution benchmark map at multiple depths can be integrated into biogeochemical models to quantify potential changes in soil carbon. Building on this work, we plan to combine repeated SOC measurements with process-based models to advance understanding of SOC dynamics. By leveraging this baseline, carbon management strategies can be refined and climate policies better informed."

**Major concern 4:**

There are no plots showing the effect of each covariate on SOC storage in the fitted model. I understand if the work doesn't want to emphasize the fitted relationships in the model (which can be messy to interpret), but I think it should be reported in the supplemental material for transparency. A group of partial dependence plots for each biome-specific model would suffice, so that the readers don't wonder if model predictions are driven by one or two strong but non-physical effects.

**Author Response:**

Thank you for the suggestion. We conducted additional calculations using partial dependence analysis to evaluate feature influence on SOC predictions. These analyses were stratified by biome to highlight differences in covariate effects.

Methods are detailed in the new **Section 2.7** ("Partial dependence analysis"), and results are presented in **Section 3.5** ("Feature effects on SOC predictions"), where we discuss how the models' response patterns are ecologically interpretable.

Full results are provided in **Supplementary Figure S3**.

**Minor comments:**

Line 22-24, Abstract. "Our analysis indicates that annual wildfire dynamics and shifts in agricultural land can influence SOC by 132 Pg C and 140 Pg C at 30 cm, and by 345 Pg C and 368 Pg C at 100 cm, representing approximately 13% of the global stocks."

These numbers seem to be calculated from the total estimated SOC stock in global areas that fall within the agricultural and fire-prone extent (line 619). Indeed, the theoretical maximum potential of SOC loss driven by any disturbances is always bounded by the total SOC stored in a given area, so the sentence is technically correct. However, this sentence reads like "we formally analyzed the effect of recent/near-future fire and agricultural dynamics on SOC change, which show that these activity poses a readily threat to this much SOC stock in the next several decades". This does not accurately summarize the relevant results (line 619). In the fire-prone pixels defined by "average annual number of burned days >1" (line 296-297), such fire dynamics very likely do not pose an immediate threat to the entire soil C pool in every fire-prone pixel (the author also pointed out multiple times that fire has mixed effects to SOC change yet to be understood). The same point applies to agricultural activities. To avoid misrepresenting the main result of this work, please adjust the phrasing of this sentence. Maybe something like "estimate shows that XXX PgC of soil carbon sits in fire-prone area and/or area with ongoing agricultural activities..."

**Author Response:**

Thank you for this comment. We have revised the sentence to clarify that the reported values represent SOC stocks located in fire-prone and agricultural areas. The sentence in the manuscript now reads:

"Our estimates indicate  $134 \pm 2$  Pg C and  $340 \pm 5$  Pg C sit in fire prone areas at 30 cm and 100 cm depth, and that  $140 \pm 2$  Pg C and  $384 \pm 8$  Pg C are in areas of ongoing agricultural activity at 30 cm and 100 cm depth, representing about 13% of global SOC stocks."

Line 505. "Our study suggests considerably greater carbon storage in the Amazon Basin."

Is it because your study better predicts the large carbon storage in the Amazon peatland (your next paragraph)? If so, explain explicitly here.

**Author Response:**

Thank you. We have revised this section of the manuscript, which now reads:

"Across the entire Amazon Basin (593 Mha), we estimate  $37 \pm 1$  Pg C at 30 cm and  $162 \pm 7$  Pg C at 100 cm, representing 4 % and 6 % of the global SOC total, respectively. Our results at 100 cm are higher than previous Amazon Basin assessments, including 47 Pg C at 100 cm (Moraes et al., 1995), 46.5 Pg C (Batjes and Dijkshoorn, 1999), and 36.1 Pg C

(Gomes et al., 2019), due in part to our use of the recently updated peatland extent dataset that better captures carbon-rich areas."

Line 551-555.

See major concern 1.

**Author Response:** We have addressed this section in our response to Major Concern 1.

Line 685 "Our data indicates that 35% of the Cerrado is used for agriculture (72/204 Mha)"

And Line 689 "Our map highlights high fire activity in the Matopiba region"

If I understand correctly, the agricultural land extent is cited from another product, and the fire activity extent is indicated by the MODIS fire frequency map. How is that "your map" and your data? (I would've assumed these terms to refer to your SOC data product in this manuscript.) It kind of confuses and distracts readers from focusing on what's truly your valuable map and your data product —the SOC stock maps at two depths.

**Author Response:**

Thank you for this comment. The agricultural land extent and fire activity are derived from external datasets (LGRIP 2015 and MODIS 2000–2023, respectively) rather than generated directly from our SOC mapping effort.

To clarify, we have revised the text to explicitly attribute these datasets to their original sources. The manuscript now reads:

"In Brazil's Cerrado, our model estimates that soils store  $8.7 \pm 0.2$  Pg C at 30 cm and  $28.1 \pm 10.8$  Pg C at 100 cm across 204 Mha. Although the Cerrado contains moderate SOC stocks compared to other biomes, 38% of the region is classified as fire-prone and 35% as agricultural land, with a 63% spatial overlap with either category, based on LGRIP 2015 and MODIS 2000-2023 datasets (Figure 8). Fire-prone areas contain an estimated  $3.3 \pm 0.1$  Pg C (30 cm) and  $10.0 \pm 0.6$  Pg C (100 cm) across 77 Mha, and agricultural areas contain  $3.1 \pm 0.1$  Pg C (30 cm) and  $9.4 \pm 0.6$  Pg C (100 cm) across 72 Mha. Fire-prone areas and agricultural land show larger SOC contrasts at 100 cm than at 30 cm compared to other areas, which may reflect differential SOC dynamics with depth. High fire activity in regions such as Matopiba could be associated with rapid agricultural expansion. These patterns suggest a combined role of fire and land use in shaping SOC dynamics (LGRIP 2015; MODIS, 2000-2023)."

**Line 699-703**

The authors have already discussed the mixed and unknown impact of fire on soil in the global fire section and the grassland section (lines 556-565, lines 424-426). Coming after these sections, these lines read redundant. Perhaps all of these can be consolidated better into the global fire section (the paragraph of lines 556-565)

**Author Response:** Thank you. We have reworked the manuscript outline and consolidated the discussion of fire impacts into a single coherent section. The revised outline now reads:

**4.2 Land-use and disturbance**

- 4.2.1 Global fires and agriculture (now includes a consolidated global fire discussion)
- 4.2.2 Regional SOC stocks relative to land-use and fire (focuses on regional distribution of fire and agriculture)

---

## Author Comment (AC2)

**Reviewer 2:**

The authors utilized 84880 and 44304 field measurements at 30cm depth and 100cm depth and combined with biome-specific machine learning approaches to map SOC at 100m spatial resolution. This novel idea would provide significant spatially explicit information in reducing uncertainties related to SOC estimation. However, there are a few issues that need to be addressed before the paper being accepted. The detailed comments are as follows.

**Author Response:**

Thank you for your careful review and helpful comments. We hope the changes we made address your suggestions and improve the manuscript.

**Line 13**, 1-hectare refers to the area, instead of the spatial resolution of the map, please make this consistent with the "100m" spatial resolution that mentioned in the title.

Author Response: Thank you. We have corrected this.

**Line 18**, What is the "average of prior estimates", which research?

**Author Response:**

Thank you for pointing this out. We have clarified the source of the "average of prior estimates" in the revised manuscript.

**Revised text:**

"We estimate global soil organic carbon stocks of 1,023  $\pm$  20 Pg C at 30 cm and 2,837  $\pm$  57 Pg C at 100 cm, representing increases of 28 % and 46 %, respectively, relative to the average of previous estimates (798 Pg C at 30 cm; 1,947 Pg C at 100 cm) calculated from SoilGrids v1-v2, Sanderman et al. (2017), GSOCmap v1.5, WISE30sec, GSDE, and HWSD v1.21."

**Line 110**, Please add a map to provide the spatial distribution of all the field measurements. Also, please add a table describing the basic information of sampled points in each biome, such as the total number of samples, mean and standard deviation values of soil samples in each biome.

**Author Response:**

Thank you for the suggestion. We have added **Figure S2** to show the spatial distribution of ground-truth SOC observations across biomes. The figure illustrates the global extent of biomes (in orange) and the locations of SOC measurements (in dark grey) at 30 cm and 100 cm depths, and includes summary statistics (number of observations, mean SOC, and standard deviation) for each biome. A subset of representative maps is shown below. **Figure S2** presents the full set of 27 maps in the Supplementary Material.

**Figure S2 (subsection).** Global distribution of biomes and ground-truth soil organic carbon (SOC) data with summary statistics. This figure shows the spatial extent of the biomes in orange and the locations of ground-truth SOC observations for 0-30 cm and 0-100 cm depths, in grey. The number of observations, mean SOC density (t C/ha), and standard deviation are reported for each biome and depth.

Line 153, This approach may introduce mistakes in treating gridded products as ground truth, leading to biased models, especially in data-scarce regions where underlying map quality is uncertain. Additionally, subsampling and binning may not fully capture spatial or ecological variability, and global performance metrics can obscure significant local errors. I suggest the authors provide (1) analysis that does not include samples that are sampled from these maps (2) model metrics and R2 that are not including these samples. As samples collected from maps are not ground truth, which will change the model's performance.

**Author Response:** We thank the reviewer for highlighting this point. We have added detail regarding the inclusion of subsampled data in **Section 2.1.3** of the methodology, which now reads:

"We analysed model performance both with and without subsamples derived from the previously described gridded datasets. The results show that, when subsamples are excluded, model performance remains largely robust, with only minor reductions of 0.01-0.02 in R² values across most biomes and ecosystems (Table S8). Montane grasslands are an exception, showing a larger drop in R² due to the low number of samples in this biome. At 100 cm depth, removing subsamples similarly preserves model performance, with only marginal changes of 0.02-0.03 in R², again with the exception of montane grasslands. These results indicate that, although map-derived samples contribute to model training, they do not fundamentally alter overall model performance or spatial patterns. The model is therefore not overly reliant on these potentially biased samples. We include these samples to improve spatial representativeness, but we have verified that their inclusion does not introduce substantial bias in our predictions. The full biome-level results with and without subsamples are available in Table S8."

|                                                              | with subs  | amples              | without sub | samples             |
|--------------------------------------------------------------|------------|---------------------|-------------|---------------------|
| Biome/Ecosystem soc30                                        | Full Count | Full R 2 | Full Count  | Full R 2 |
| Forests                                                      | i un count | I UII IX            | 1 un count  | I ull IX            |
| Tropical and subtropical moist broadleaf forests             | 9,830      | 0.84                | 9,162       | 0.82                |
| Tropical and subtropical dry broadleaf forests               | 3,438      | 0.74                | 3,424       | 0.74                |
| Tropical and subtropical coniferous forests                  | 1,234      | 0.79                | 1,232       | 0.79                |
| Temperate broadleaf and mixed forests                        | 24,935     | 0.72                | 24,889      | 0.73                |
| Temperate coniferous forest                                  | 8,820      | 0.71                | 8,781       | 0.71                |
| Boreal forests / taiga and tundra                            | 4,595      | 0.60                | No subsa    | imples              |
| Mangroves - based on Global Mangrove Watch (2020) extent     | 1,577      | 0.80                | 1,238       | 0.79                |
| Grasslands and shrublands                                    |            |                     |             |                     |
| Tropical and subtropical grasslands, savannas and shrublands | 4,074      | 0.83                | 3,439       | 0.83                |
| Temperate grasslands, savannas and shrublands                | 8,667      | 0.74                | 8,448       | 0.74                |
| Flooded grasslands and savannas                              | 1,298      | 0.76                | 1,024       | 0.76                |
| Montane grasslands and shrublands                            | 418        | 0.73                | 33          | 0.40                |
| Other biomes                                                 |            |                     |             |                     |
| Mediterranean forests, woodlands and scrub                   | 7,042      | 0.79                | 6,962       | 0.79                |
| Deserts and xeric shrublands                                 | 9,041      | 0.70                | 8,493       | 0.70                |
| Peatlands - based on UNEP classification (2022)              | 3,372      | 0.67                | 2,805       | 0.67                |

|                                                              | with subs  | amples              | without sub | samples             |
|--------------------------------------------------------------|------------|---------------------|-------------|---------------------|
| Biome/Ecosystem soc100
Forests                            | Full Count | Full R 2 | Full Count  | Full R 2 |
| Tropical and subtropical moist broadleaf forests             | 2,155      | 0.90                | 1,573       | 0.87                |
| Tropical and subtropical dry broadleaf forests               | 1,322      | 0.71                | 1,295       | 0.71                |
| Temperate broadleaf and mixed forests                        | 14,516     | 0.72                | 14,503      | 0.72                |
| Temperate coniferous forest                                  | 6,919      | 0.70                | 6,873       | 0.70                |
| Boreal forests / taiga and tundra                            | 1,716      | 0.68                | No subsa    | imples              |
| Mangroves - based on Global Mangrove Watch (2020) extent     | 1,453      | 0.80                | 1,097       | 0.78                |
| Grasslands and shrublands                                    |            |                     |             |                     |
| Tropical and subtropical grasslands, savannas and shrublands | 1,406      | 0.90                | 1,360       | 0.90                |
| Temperate grasslands, savannas and shrublands                | 8,236      | 0.62                | 8,219       | 0.62                |
| Flooded grasslands and savannas                              | 1,034      | 0.77                | 790         | 0.75                |
| Montane grasslands and shrublands                            | 385        | 0.56                | 21          | 0.28                |
| Other biomes                                                 |            |                     |             |                     |
| Mediterranean forests, woodlands and scrub                   | 1,298      | 0.83                | 1,297       | 0.83                |
| Deserts and xeric shrublands                                 | 4,431      | 0.66                | 4,370       | 0.66                |
| Peatlands - based on UNEP classification (2022)              | 1,936      | 0.77                | 1,286       | 0.75                |

**Table S8.** Model performance (R²) and sample count per biome and ecosystem for soil organic carbon (SOC) predictions for 0-30 cm and 0-100 cm depths. Results are shown for biome-specific models trained with the full dataset and with or without the inclusion of subsampled gridded datasets.

**Line 177**, The differences in SOC lab methods (e.g., dry combustion vs. Walkley-Black) are not accounted for. Could this introduce systematic regional bias in SOC predictions?

**Author Response:**

Thank you for this comment. To address it, we have added the following clarification to **Methods Section 2.1.4** of the manuscript:

"Soil organic carbon was assessed using different methods (e.g., dry combustion or Walkley-Black) across datasets, reflecting the extended timeline over which data were collected. Most datasets compiled in this study had already been internally harmonized using conversion factors, and we did not apply further adjustments for potential method-based discrepancies. Most datasets reported final soil carbon stocks standardized in t C/ha. Older campaigns relied more on loss on ignition (LOI) and Walkley-Black methods, potentially introducing regional differences despite conversions. In contrast, more recent datasets, which we integrated to enhance spatial coverage, were predominantly analysed using dry combustion. Thus, methodological differences remain a potential source of bias in the compiled SOC estimates; however, remaining method-related effects are expected to be relatively minor compared with the overall spatial variability captured by the model."

Line 197, Please replace "Environment" with "environment".

**Author Response:** Thank you. We corrected this.

**Line 213**, Please justify why ET was included for SOC estimation.

**Author Response:**

Thank you. We have updated the manuscript to include a clear justification for including ET. The revised text now reads:

"To account for climate-driven factors influencing soil carbon, we integrated MODIS Evapotranspiration (ET) data (kg/m²/8-day; MODIS/061/MOD16A2GF). ET serves as an indicator of both water availability and ecosystem productivity, which directly affect plant growth, litter input, and soil organic carbon accumulation. By including ET, we capture spatial variations in climate that help explain SOC distribution globally."

Line 219, Is fire frequency also derived from MODIS based products?

**Author Response:**

Thank you for the comment. In the Discussion section of the manuscript, we clarified the use of MODIS products for fire frequency. Further, we expanded **Methods Section 2.2**. The fire raster used as an input layer was derived from the MODIS/061/MCD64A1 dataset for 2000–2023 and represents the average number of burned days per pixel per year. This raster was then converted into a binary fire mask, with pixels having an average annual burned days ≥1

assigned a value of 1 (fire-prone) and all others assigned 0 (not fire-prone). Zonal statistics were computed using this binary mask.

Line 222, Please specify which years' ALOS2 data have been utilized.

**Author Response:**

Thank you. This information is provided in the manuscript, which reads:

"We used ALOS-2 PALSAR-2 data for synthetic aperture radar (SAR) backscatter (HH and HV bands) to account for the effects of soil moisture and structure on soil carbon. We calculated the median for the years 2019-2020 for each band."

Line 225, Please replace NiR with "NIR" and justify why these bands in Landsat8 were selected?

**Author Response:**

Thank you. We have corrected "NiR" to "NIR" and updated the manuscript to justify the selection of these bands. The revised text now reads:

"We processed Landsat 8 data (bands red, NIR, SWIR1, and SWIR2), taking the median for 2022 and gap-filling with data from prior years. These four bands capture vegetation and surface reflectance, providing insight into land cover, vegetation type, and land-use patterns, which are factors closely associated with soil organic carbon distribution at the landscape-scale. Furthermore, these bands capture spectral signals related to vegetation recovery and land cover dynamics, reflecting temporal changes across landscapes. This information helps to contextualize historical soil measurements. Of all Landsat 8 bands, these four are the most informative and commonly used for SOC estimation."

Line 229, The spatial resolution of all the remote sensing data or remote sensing-based products are not the same, how the mismatch in spatial resolution was handled? Please provide more details.

**Author Response:**

We thank the reviewer for this important comment. We have revised **Methods Section 2.4** to further detail our approach. As described in the manuscript:

"All geospatial input raster layers were reprojected to a standardized spatial reference system and a uniform 100 m spatial resolution prior to model training and inference. The input datasets, described in previous sections, included: Copernicus Global Land Cover (100 m), Landsat 8 bands (30 m), MODIS Land Surface Temperature (1 km), MODIS Evapotranspiration (500 m), ALOS PALSAR (25 m), SoilGrids soil properties (250 m), GPM2.0 global peatland map (1 km), Global Mangrove Watch forest extent (25 m), NCSCDv2

permafrost map (1 km), WRB soil classification (250 m), Congo Basin peatland map (50 m), Peruvian peatland map (50 m), and MarSOC tidal marsh map (10 m). Differences in native spatial resolution among datasets were addressed through interpolation-based resampling. The harmonized 100 m layers were then used as predictor variables in the RF modeling framework to generate the final global 100 m resolution product."

Also, how did the temporal mismatch between different Earth Observation sources and the temporal mismatch between SOC data and EO were handled? SOC maps often combine soil profile data collected over decades with environmental covariates reflecting more recent conditions. This ignores soil property changes over time, particularly in areas affected by land-use change or degradation.

**Author Response:**

Thank you. To address it, we have added the following section to the manuscript.

**4.3.1 Baseline map for research and monitoring**

"We present a high-resolution (100 m) soil organic carbon map at two depths, representing baseline SOC stocks circa 2022. Soil profile data collected from the 1950s to the 2020s were combined with contemporary remote sensing covariates, using Landsat 8 bands to capture landscape conditions around 2022. The use of long-term soil datasets aligns with previous global mapping efforts, such as SoilGrids v1-v2 and GSOCmap v1.5, which integrate multi-decadal data to capture large-scale spatial patterns. The primary goal of this work was to represent SOC variation across diverse landscape gradients with a resolution fine enough to capture local differences while maintaining global coverage. The resulting map can support applications in land management, nature-based climate solutions, and future assessments of SOC dynamics.

SOC data inevitably span decades, while Earth observation covariates reflect more recent conditions. In landscapes without major human disturbances, SOC is assumed to remain relatively stable over time, so historical soil measurements can reliably represent baseline stocks when combined with current remote sensing data. In areas affected by landuse change, degradation, or recovery, such as forests regrowing after wildfire, agricultural rotations, or restored wetlands, SOC may have changed since sampling. For instance, samples from western CONUS encompass forests at different successional stages recovering from events such as wildfires or storms. Remote sensing captures spectral signals related to land cover and vegetation recovery, helping to contextualize historical soil measurements. Consequently, the SOC map reflects not only static soil conditions but also the imprints of temporal changes across landscapes. To address potential temporal discrepancies at the pixel level, we generated an uncertainty map alongside the SOC estimates, allowing users to assess confidence in areas where SOC may have changed substantially since sampling.

Our dataset integrates both natural and human-modified landscapes, including agricultural areas, managed forests, wetlands, and coastal ecosystems such as mangroves. By combining long-term soil measurements with remote sensing covariates and machine learning models, the map captures both spatial variation in SOC and the influence of historical land-use changes and recovery processes. Overall, the high-resolution SOC maps represent a valuable tool for carbon management and conservation. The maps enable fine-scale detection of spatial variability, supports assessments of land-use impacts, and provides a foundation for monitoring SOC dynamics over time, even in regions with limited historical soil data."

**Line 250**, The abstract mentioned a bio-specific machine learning approach for mapping SOC, more details needed to describe the bio-specific approach.

**Author Response:**

Thank you. We have added the following clarification to **Methods Section 2.5** to clarify the biome-specific approach:

"We then explored spatially stratified models based on different levels of ecological classification, using the WWF biomes classification, which delineates the following biomes globally (Olson et al., 2001): TMB-Tropical and subtropical moist broadleaf forests; TDB-Tropical and subtropical dry broadleaf forests; TCF-Tropical and subtropical coniferous forests; TeBF-Temperate broadleaf and mixed forests; TeCF-Temperate coniferous forest; BoF-Boreal forests/taiga; TUN-Tundra; TrG-Tropical and subtropical grasslands, savannas, and shrublands; TeG-Temperate grasslands, savannas, and shrublands; FGr-Flooded grasslands and savannas; MtG-Montane grasslands and shrublands; MeF-Mediterranean forests, woodlands, and scrub; DES-Deserts and xeric shrublands.

For each biome, we tested model performance when biomes were subdivided by continent or region (e.g., TMB in South America vs. Africa vs. Asia). While this biome-by-continent approach often produced high  $R^2$  values in certain regions, it also introduced considerable variability and data imbalance. Some biome-continent combinations had strong performance (e.g.,  $R^2 > 0.7$ ), while others suffered from limited sample size or overfitting. For example, TMB reached  $R^2$  of 0.53 in South America, 0.77 in Africa, and 0.49 in Asia and TeBF achieved  $R^2$  of 0.31 in the Americas, 0.85 in Asia, and 0.47 in Europe. Based on these findings, we selected a consistent set of 12 biome models that maintained ecological specificity while preserving sufficient data volume and geographic breadth for each model.

Peatland (P) and mangrove (MG) ecosystems were modeled separately at both 30 cm and 100 cm depths due to their distinct biophysical characteristics and the importance of their soil carbon stocks. To minimize the inclusion of non-mangrove or non-peatland sites, we relied on the most accurate and spatially explicit global datasets available: the Global Mangrove Watch v3.0 (2020) (Bunting et al., 2022) and the Global Peatland Assessment Database v2 (GPM2.0, 2022) from UNEP, both of which account for small-scale land use

changes such as local farming and deforestation. The Global Peatland Assessment Database 2022 v.2 (GPM2.0) from UNEP categorizes peatlands into two primary classes: 'peat dominated', where peat is the dominant land cover, and 'peat-in-soil mosaic', where peat occurs in smaller patches within other land cover types. We combined both classes (GPM2.0 classes 1 and 2) into a single mask to capture the full extent of peat-dominated and peat-in-soil mosaics. To improve regional accuracy, we integrated high-resolution peatland datasets for the Congo Basin and Peruvian Amazon. These higher-resolution regional datasets were prioritized in overlapping areas, such that where both global and regional datasets were available, the regional data replaced the global data, while areas outside these regions retained the GPM2.0 coverage. While not all SOC ground-truth points had land cover metadata, 58% of points within the mangrove extent at 30 cm (71% at 100 cm) were explicitly identified as mangrove, and 14% of points within the peatland extent at 30 cm (35% at 100 cm) were identified as peatland.

To evaluate predictive performance of biome- and ecosystem-models, we conducted k-fold cross-validations with 50 simulations. In each simulation, the dataset was partitioned into training and test subsets. A separate model was trained on the training subset and evaluated on both subsets. Model performance (R²) was computed for each simulation, and the mean and standard deviation for the 50 simulations are reported (Table 1). In addition, a weighted R², which combines training and test performance, is reported in Table 1. Overall, the stratified biome-model approach provided a balance between model complexity and generalizability."

Line 251, How the parameter of "mtry" has been trained and specified in the random forest model?

**Author Response:**

Thank you. We have clarified this point in **Methods section 2.6** as follows:

"The number of candidate features considered at each split in the Random Forest (mtry) was not explicitly tuned (tuner\_num\_trials = 0); the TensorFlow Decision Forests (TF-DF) RandomForestModel used default regression settings, automatically setting mtry to one-third of the available features."

**Line 325**, Does the average represent all the aforementioned SOC products? Please consider adding the average SOC of each SOC map.

**Author Response:**

Thank you for the comment. Yes, the average represents the SOC products mentioned in the text. We have added the SOC value of each map in **Figure 1**, in the table attached below:

| Global Soil Organic Carbon Stocks |            |                 |           |                   |            |           |            |           |  |
|-----------------------------------|------------|-----------------|-----------|-------------------|------------|-----------|------------|-----------|--|
| Digital soil mapping              |            |                 |           |                   |            |           |            |           |  |
|                                   |            | calculate-first |           | interpolate-first | t _ | Taxot     | ransfer me | ethod     |  |
|                                   | this study | SoilGridsv2     | Sanderman | SoilGridsv1       | GSOCv1.5   | WISE30sec | GSDE       | HWSDv1.21 |  |
| Global (Pg C                      | )          | 2021            | 2017      | 2017              | 2018       | 2016      | 2014       | 2012      |  |
| soc30                             | 1023 ± 20  | 594             | 869       | 1,176             | 684        | 883       | 794        | 588       |  |
| soc100                            | 2837 ± 57  | -               | 1,960     | 2,769             | -          | 1,969     | 1,907      | 1,130     |  |

**Line 267**, Please provide other model evaluations metrics for mapping SOC such as RMSE instead of only R2.

**Author Response:**

Thank you for the suggestion. We have included additional model evaluation metrics to complement R2. These metrics are presented in the **Supplementary Information** as follows:

| Biome/Ecosystem soc30
Forests                             |         | Train
Count | Train R² | Test
Count | Test R² | Full
Count | Full R 2 |
|--------------------------------------------------------------|---------|----------------|----------|---------------|---------|---------------|---------------------|
| Tropical and subtropical moist broadleaf forests             | TMB     | 7,864          | 0.84     | 1,966         | 0.62    | 9,830         | 0.84                |
| Tropical and subtropical dry broadleaf forests               | TDB     | 2,750          | 0.73     | 688           | 0.52    | 3,438         | 0.74                |
| Tropical and subtropical coniferous forests                  | TCF     | 987            | 0.78     | 247           | 0.43    | 1,234         | 0.79                |
| Temperate broadleaf and mixed forests                        | TeBF    | 19,947         | 0.73     | 4,988         | 0.33    | 24,935        | 0.72                |
| Temperate coniferous forest                                  | TeCF    | 7,055          | 0.68     | 1,765         | 0.43    | 8,820         | 0.71                |
| Boreal forests / taiga and tundra                            | BoF-TUN | 3,675          | 0.59     | 920           | 0.12    | 4,595         | 0.60                |
| Mangroves - based on Global Mangrove Watch (2020) extent     | MG      | 1,261          | 0.79     | 316           | 0.57    | 1,577         | 0.80                |
| Grasslands and shrublands                                    |         |                |          |               |         |               |                     |
| Tropical and subtropical grasslands, savannas and shrublands | TrG     | 3,259          | 0.82     | 815           | 0.65    | 4,074         | 0.83                |
| Temperate grasslands, savannas and shrublands                | TeG     | 6,932          | 0.73     | 1,735         | 0.49    | 8,667         | 0.74                |
| Flooded grasslands and savannas                              | FGr     | 1,038          | 0.77     | 260           | 0.42    | 1,298         | 0.76                |
| Montane grasslands and shrublands                            | MtG     | 334            | 0.93     | 84            | 0.12    | 418           | 0.73                |
| Other biomes                                                 |         |                |          |               |         |               |                     |
| Mediterranean forests, woodlands and scrub                   | MeF     | 5,632          | 0.79     | 1,410         | 0.43    | 7,042         | 0.79                |
| Deserts and xeric shrublands                                 | DES     | 7,232          | 0.70     | 1,809         | 0.33    | 9,041         | 0.70                |
| Peatlands - based on UNEP classification (2022)              | Р       | 2,697          | 0.66     | 675           | 0.22    | 3,372         | 0.67                |

| Biome/Ecosystem soc30                                        |         | Train  | Train | Test  | Test | Full   | Full |
|--------------------------------------------------------------|---------|--------|-------|-------|------|--------|------|
| Forests                                                      |         | Count  | RMSE  | Count | RMSE | Count  | RMSE |
| Tropical and subtropical moist broadleaf forests             | TMB     | 7,864  | 38    | 1,966 | 67   | 9,830  | 39   |
| Tropical and subtropical dry broadleaf forests               | TDB     | 2,750  | 42    | 688   | 44   | 3,438  | 40   |
| Tropical and subtropical coniferous forests                  | TCF     | 987    | 20    | 247   | 33   | 1,234  | 20   |
| Temperate broadleaf and mixed forests                        | TeBF    | 19,947 | 60    | 4,988 | 100  | 24,935 | 62   |
| Temperate coniferous forest                                  | TeCF    | 7,055  | 84    | 1,765 | 111  | 8,820  | 80   |
| Boreal forests / taiga and tundra                            | BoF-TUN | 3,675  | 143   | 920   | 229  | 4,595  | 144  |
| Mangroves - based on Global Mangrove Watch (2020) extent     | MG      | 1,261  | 25    | 316   | 34   | 1,577  | 24   |
| Grasslands and shrublands                                    |         |        |       |       |      |        |      |
| Tropical and subtropical grasslands, savannas and shrublands | TrG     | 3,259  | 31    | 815   | 47   | 4,074  | 31   |
| Temperate grasslands, savannas and shrublands                | TeG     | 6,932  | 35    | 1,735 | 40   | 8,667  | 34   |
| Flooded grasslands and savannas                              | FGr     | 1,038  | 45    | 260   | 44   | 1,298  | 43   |
| Montane grasslands and shrublands                            | MtG     | 334    | 9     | 84    | 54   | 418    | 21   |
| Other biomes                                                 |         |        |       |       |      |        |      |
| Mediterranean forests, woodlands and scrub                   | MeF     | 5,632  | 24    | 1,410 | 35   | 7,042  | 24   |
| Deserts and xeric shrublands                                 | DES     | 7,232  | 28    | 1,809 | 35   | 9,041  | 27   |
| Peatlands - based on UNEP classification (2022)              | Р       | 2,697  | 127   | 675   | 216  | 3,372  | 129  |

| Biome/Ecosystem soc30                                        |         | Train  | Train | Test  | Test | Full   | Full |
|--------------------------------------------------------------|---------|--------|-------|-------|------|--------|------|
| Forests                                                      |         | Count  | MAE   | Count | MAE  | Count  | MAE  |
| Tropical and subtropical moist broadleaf forests             | TMB     | 7,864  | 15    | 1,966 | 27   | 9,830  | 15   |
| Tropical and subtropical dry broadleaf forests               | TDB     | 2,750  | 16    | 688   | 26   | 3,438  | 16   |
| Tropical and subtropical coniferous forests                  | TCF     | 987    | 13    | 247   | 24   | 1,234  | 13   |
| Temperate broadleaf and mixed forests                        | TeBF    | 19,947 | 23    | 4,988 | 39   | 24,935 | 23   |
| Temperate coniferous forest                                  | TeCF    | 7,055  | 35    | 1,765 | 59   | 8,820  | 35   |
| Boreal forests / taiga and tundra                            | BoF-TUN | 3,675  | 67    | 920   | 108  | 4,595  | 68   |
| Mangroves - based on Global Mangrove Watch (2020) extent     | MG      | 1,261  | 14    | 316   | 25   | 1,577  | 14   |
| Grasslands and shrublands                                    |         |        |       |       |      |        |      |
| Tropical and subtropical grasslands, savannas and shrublands | TrG     | 3,259  | 11    | 815   | 18   | 4,074  | 10   |
| Temperate grasslands, savannas and shrublands                | TeG     | 6,932  | 14    | 1,735 | 22   | 8,667  | 13   |
| Flooded grasslands and savannas                              | FGr     | 1,038  | 19    | 260   | 28   | 1,298  | 18   |
| Montane grasslands and shrublands                            | MtG     | 334    | 6     | 84    | 15   | 418    | 8    |
| Other biomes                                                 |         |        |       |       |      |        |      |
| Mediterranean forests, woodlands and scrub                   | MeF     | 5,632  | 13    | 1,410 | 22   | 7,042  | 13   |
| Deserts and xeric shrublands                                 | DES     | 7,232  | 11    | 1,809 | 17   | 9,041  | 10   |
| Peatlands - based on UNEP classification (2022)              | Р       | 2,697  | 58    | 675   | 103  | 3,372  | 59   |

**Table S9.** Biome-specific model performance metrics for 0-30 cm soil organic carbon (SOC30).

| Biome/Ecosystem soc100 Forests                               |         | Train
Count | Train R² | Test
Count | Test R² | Full
Count | Full R 2 |
|--------------------------------------------------------------|---------|----------------|----------|---------------|---------|---------------|---------------------|
| Tropical and subtropical moist broadleaf forests             | TMB     | 1.724          | 0.89     | 431           | 0.79    | 2.155         | 0.90                |
|                                                              |         | ,              |          |               |         | ,             |                     |
| Tropical & subtropical dry broadleaf and coniferous forests  | TDB-TCF | 1,057          | 0.73     | 265           | 0.27    | 1,322         | 0.71                |
| Temperate broadleaf and mixed forests                        | TeBF    | 11,612         | 0.70     | 2,904         | 0.36    | 14,516        | 0.72                |
| Temperate coniferous forest                                  | TeCF    | 5,535          | 0.70     | 1,384         | 0.26    | 6,919         | 0.70                |
| Boreal forests / taiga and tundra                            | BoF-TUN | 1,372          | 0.71     | 344           | 0.28    | 1,716         | 0.68                |
| Mangroves - based on Global Mangrove Watch (2020) extent     | MG      | 1,162          | 0.78     | 291           | 0.52    | 1,453         | 0.80                |
| Grasslands and shrublands                                    |         |                |          |               |         |               |                     |
| Tropical and subtropical grasslands, savannas and shrublands | TrG     | 1,124          | 0.89     | 282           | 0.91    | 1,406         | 0.90                |
| Temperate grasslands, savannas and shrublands                | TeG     | 6,588          | 0.58     | 1,648         | 0.40    | 8,236         | 0.62                |
| Flooded grasslands and savannas                              | FGr     | 827            | 0.76     | 207           | 0.45    | 1,034         | 0.77                |
| Montane grasslands and shrublands                            | MtG     | 308            | 0.52     | 77            | 0.60    | 385           | 0.56                |
| Other biomes                                                 |         |                |          |               |         |               |                     |
| Mediterranean forests, woodlands and scrub                   | MeF     | 1,038          | 0.81     | 260           | 0.64    | 1,298         | 0.83                |
| Deserts and xeric shrublands                                 | DES     | 3,544          | 0.64     | 887           | 0.24    | 4,431         | 0.66                |
| Peatlands - based on UNEP classification (2022)              | Р       | 1,548          | 0.74     | 388           | 0.41    | 1,936         | 0.77                |

| Biome/Ecosystem soc100                                       |         | Train  | Train | Test  | Test | Full   | Full |
|--------------------------------------------------------------|---------|--------|-------|-------|------|--------|------|
| Forests                                                      |         | Count  | RMSE  | Count | RMSE | Count  | RMSE |
| Tropical and subtropical moist broadleaf forests             | TMB     | 1,724  | 155   | 431   | 196  | 2,155  | 145  |
| Tropical & subtropical dry broadleaf and coniferous forests  | TDB-TCF | 1,057  | 125   | 265   | 243  | 1,322  | 135  |
| Temperate broadleaf and mixed forests                        | TeBF    | 11,612 | 193   | 2,904 | 286  | 14,516 | 189  |
| Temperate coniferous forest                                  | TeCF    | 5,535  | 201   | 1,384 | 258  | 6,919  | 193  |
| Boreal forests / taiga and tundra                            | BoF-TUN | 1,372  | 270   | 344   | 478  | 1,716  | 288  |
| Mangroves - based on Global Mangrove Watch (2020) extent     | MG      | 1,162  | 91    | 291   | 123  | 1,453  | 87   |
| Grasslands and shrublands                                    |         |        |       |       |      |        |      |
| Tropical and subtropical grasslands, savannas and shrublands | TrG     | 1,124  | 113   | 282   | 82   | 1,406  | 105  |
| Temperate grasslands, savannas and shrublands                | TeG     | 6,588  | 129   | 1,648 | 144  | 8,236  | 121  |
| Flooded grasslands and savannas                              | FGr     | 827    | 131   | 207   | 277  | 1,034  | 140  |
| Montane grasslands and shrublands                            | MtG     | 308    | 68    | 77    | 37   | 385    | 61   |
| Other biomes                                                 |         |        |       |       |      |        |      |
| Mediterranean forests, woodlands and scrub                   | MeF     | 1,038  | 91    | 260   | 124  | 1,298  | 87   |
| Deserts and xeric shrublands                                 | DES     | 3,544  | 94    | 887   | 134  | 4,431  | 91   |
| Peatlands - based on UNEP classification (2022)              | Р       | 1,548  | 283   | 388   | 502  | 1,936  | 278  |

| Biome/Ecosystem soc100                                       |         | Train  | Train | Test  | Test | Full   | Full |
|--------------------------------------------------------------|---------|--------|-------|-------|------|--------|------|
| Forests                                                      |         | Count  | MAE   | Count | MAE  | Count  | MAE  |
| Tropical and subtropical moist broadleaf forests             | TMB     | 1,724  | 74    | 431   | 108  | 2,155  | 70   |
| Tropical & subtropical dry broadleaf and coniferous forests  | TDB-TCF | 1,057  | 47    | 265   | 98   | 1,322  | 49   |
| Temperate broadleaf and mixed forests                        | TeBF    | 11,612 | 63    | 2,904 | 100  | 14,516 | 62   |
| Temperate coniferous forest                                  | TeCF    | 5,535  | 83    | 1,384 | 128  | 6,919  | 80   |
| Boreal forests / taiga and tundra                            | BoF-TUN | 1,372  | 129   | 344   | 232  | 1,716  | 135  |
| Mangroves - based on Global Mangrove Watch (2020) extent     | MG      | 1,162  | 50    | 291   | 82   | 1,453  | 47   |
| Grasslands and shrublands                                    |         |        |       |       |      |        |      |
| Tropical and subtropical grasslands, savannas and shrublands | TrG     | 1,124  | 38    | 282   | 45   | 1,406  | 35   |
| Temperate grasslands, savannas and shrublands                | TeG     | 6,588  | 42    | 1,648 | 67   | 8,236  | 41   |
| Flooded grasslands and savannas                              | FGr     | 827    | 60    | 207   | 108  | 1,034  | 59   |
| Montane grasslands and shrublands                            | MtG     | 308    | 18    | 77    | 27   | 385    | 16   |
| Other biomes                                                 |         |        |       |       |      |        |      |
| Mediterranean forests, woodlands and scrub                   | MeF     | 1,038  | 42    | 260   | 68   | 1,298  | 41   |
| Deserts and xeric shrublands                                 | DES     | 3,544  | 41    | 887   | 67   | 4,431  | 41   |
| Peatlands - based on UNEP classification (2022)              | Р       | 1,548  | 126   | 388   | 212  | 1,936  | 123  |

**Table S10.** Biome-specific model performance metrics for 0-100 cm soil organic carbon (SOC100).

**Line 356**, Please position the captions below the figure, same comment for all the figures.

**Author Response:** Captions have been placed below the figures.

**Line 202**, The authors mentioned that they used SoilGrids2 data as the inputs of the models. However, SoilGrids2 itself is a modeled product with known spatial biases and uncertainties. Could the authors clarify what steps were taken to mitigate the risk of error propagation from SoilGrids2 into the final SOC predictions?

**Author Response:** We used a set of soil characteristics from SoilGrids2 to support global SOC modeling. We recognize that SoilGrids2 is itself a predictive product derived from ground observations and ancillary data, and therefore may contain spatial biases and uncertainties in magnitude. Nevertheless, we chose to include these datasets for three main reasons:

- Comparative modeling tests: We evaluated SOC models with and without SoilGrids2 layers, using only remote-sensing, vegetation, and climate data in the latter case. Treating SOC measurements as independent targets, our results showed that including SoilGrids2 layers substantially improved mapping accuracy at both global and biome scales.
- Feature space enrichment: In principle, any dataset, even if noisy or weakly correlated with SOC, can serve as part of the feature space for SOC prediction. Features that do not contribute meaningfully can simply be excluded during model optimization.
- 3. Error characterization: While SoilGrids2 does not provide pixel-level uncertainty estimates that could propagate through modeling frameworks, our approach relies on cross-validation and pixel-level SOC uncertainty estimates. Consequently, the errors we report represent true prediction uncertainties at both pixel and regional levels. The main limitation is that fine-scale spatial variability, which may reflect SoilGrids2's inherent spatial errors, cannot be fully quantified. However, at larger spatial scales and biome levels, these uncertainties are unlikely to exert a significant effect.

---

## Author Comment (AC3)

**Reviewer 3:**

**General comments:**

This manuscript mapped global 100-m resolution SOC through compiling 84,880 topsoil, 44,304 subsoil SOC samples and covariates to multi-sources remote sensing and other data products as extra layers for training biome-specific random forest models. This high spatial resolution product is an important resource for future studies on soil carbon management, thus the major outcome from this work is useful and timely needed for soil biogeochemistry and carbon cycle modeling communities. The comprehensive data inputs authors used, random forest based geospatial predictive mapping are popular and robust methods, thus the quality of the produced SOC map is partially justified. However, the writing of the discussion section in this manuscript, justification of bias correction and missing attribution of different uncertainty sources, have not been addressed properly. In addition, I checked the data product and found lots of missing points in the uploaded geotiff file. Overall, this is a solid and interesting work, but I would only recommend it for publication at ESSD after major revisions to address my following concerns.

**Author Response:**

Thank you for taking the time to review our manuscript and provide detailed feedback. We appreciate your comments. Regarding the missing points in the uploaded GeoTIFF, we checked that the dataset is complete. The file is in COG format with overviews, and some overviews may take slightly longer to load than others.

**Specific comments:**

1. Histogram-based bias correction can be tricky. Authors claim the better match is achieved after bias-correction, but the correlation of SOC stock to environmental covariates and categorical datasets for training are forced to be changed. Since Soilgrid's product did not show biased results after training, I wonder if any of these covariates and categorical maps added to layers in your training model are the reason? Could you explain the reason causing this bias and justify that the overall uncertainty of the data product is not largely affected by your bias-correction?

**Author Response:**

In the previous version of the soil carbon map, a global histogram matching bias correction was applied to the combined outputs of the three broad models (global, mangrove, peatland) to reduce systematic over- or underestimation relative to observed values. In the current version, we modeled soil carbon separately for 14 biomes and ecosystems, with each model trained on a more homogeneous subset of data. This biome-specific approach inherently reduces systematic bias, as each model is better able to capture local ecological

patterns. Therefore, no additional bias correction was applied, and model outputs are reported directly from the biome-specific models.

This clarification has been added to **Methods Section 2.6**, as follows:

"The biome-specific approach inherently reduces systematic bias, allowing each model to better capture local patterns. Consequently, no additional bias correction (i.e. histogram-based bias correction) was applied, and the reported values reflect the direct outputs of the biome-specific models."

2. The highlight of this work is the ultra fine 100-m resolution product, which is unique product. But in your manuscript, I cannot find any discussion on how you get a meaningful high resolution data product. By just using soil profiles to evaluate your model, I entrust your model to mapping the non-linear correlation of SOC stock to other covariates. But without high resolution input, your 100m resolution is less persuasive, such as Soilgrids 250m product using some 250m input covariates like MODIS products. In your study, I did not see any very high resolution products for model training. I would suggest explaining what high resolution products you're used to increase the credibility of your high resolution product.

**Author Response:**

We thank the reviewer for this important comment. We have revised **Methods Section 2.4** to detail our approach. As described in the manuscript:

"All geospatial input raster layers were reprojected to a standardized spatial reference system and a uniform 100 m spatial resolution prior to model training and inference. The input datasets, described in previous sections, included: Copernicus Global Land Cover (100 m), Landsat 8 bands (30 m), MODIS Land Surface Temperature (1 km), MODIS Evapotranspiration (500 m), ALOS PALSAR (25 m), SoilGrids soil properties (250 m), GPM2.0 global peatland map (1 km), Global Mangrove Watch forest extent (25 m), NCSCDv2 permafrost map (1 km), WRB soil classification (250 m), Congo Basin peatland map (50 m), Peruvian peatland map (50 m), and MarSOC tidal marsh map (10 m). Differences in native spatial resolution among datasets were addressed through interpolation-based resampling. The harmonized 100 m layers were then used as predictor variables in the RF modeling framework to generate the final global 100 m resolution product."

3. Authors described SOC stock results over specific regions/categories, but also wrote lengthy discussion with lots of statements only weakly related to the SOC stock product itself. I feel like reading a review paper and lots of results are just descriptions of categorical or environmental covariates input. I recommend authors to simplify your writings in the results section and focus on discussing SOC stock. I have one example in additional points, but expect authors to double check the whole manuscript.

**Author Response:**

We appreciate this helpful comment. We consolidated the Results and Discussion to focus on the SOC stock dataset. The manuscript structure was reorganized (Results 3.1–3.6; Discussion 4.1–4.3) to emphasize model performance, spatial patterns, feature effects, and uncertainty, in alignment with the data product and its interpretation. The revised Discussion highlights global patterns of SOC in comparison with existing global maps and briefly addresses land use and disturbance, providing context for map applications in carbon management, monitoring, and research.

**Revised Manuscript Outline:**

**Results:** 3.1 Global distribution of soil organic carbon, 3.2 Ground truth data across biomes and depths, 3.3 Model generalization across biomes, 3.4 Final model performance across biomes, 3.5 Feature effects on SOC predictions, 3.6 Global and regional uncertainty.

**Discussion:** 4.1 Global patterns of SOC, 4.1.1 Comparison with existing global map, 4.1.2 Critical ecosystems and patterns of SOC across biomes, 4.2 Land-use and disturbance, 4.2.1 Global fires and agriculture, 4.2.2 Regional SOC stocks relative to land-use and fire, 4.3 Implications for carbon management and conservation, 4.3.1 Baseline map for research and monitoring, 4.3.2 Policy and management applications.

4. This concern is related to my point #3. I found the discussion on uncertainty is missing in the results section. I appreciate authors separately discussing their results under different categories like fire-prone region, agriculture land and peatland, but a word or two to summarize the uncertainty from your data products over these regions can be valuable evaluation and probably can help you find the source of bias when training your model with raw data?

**Author Response:**

We appreciate this constructive comment. We have added a separate section (Results 3.6 Global and regional uncertainty) presenting global maps of uncertainty. We also computed regional uncertainties, including model variance and residual variance, using established zonal inference methods (Xu et al., 2017; McRoberts et al., 2019; 2022). These regional uncertainties are reported throughout the revised manuscript, providing a clearer evaluation of potential sources of bias.

**References:**

Xu, L., Saatchi, S. S., Shapiro, A., Meyer, V., Ferraz, A., Yang, Y., Bastin, J.-F., Banks, N., Boeckx, P., Verbeeck, H., Lewis, S. L., Muanza, E. T., Bongwele, E., Kayembe, F., Mbenza, D., Kalau, L., Mukendi, F., Ilunga, F., and Ebuta, D.: Spatial Distribution of Carbon Stored in

Forests of the Democratic Republic of Congo, Scientific Reports, 7, 15030, https://doi.org/10.1038/s41598-017-15050-z, 2017.

McRoberts, R. E., Næsset, E., Saatchi, S., and Quegan, S.: Statistically rigorous, model-based inferences from maps, Remote Sensing of Environment, 279, 113028, https://doi.org/10.1016/j.rse.2022.113028, 2022.

McRoberts, R. E., Næsset, E., Liknes, G. C., Chen, Q., Walters, B. F., Saatchi, S., and Herold, M.: Using a Finer Resolution Biomass Map to Assess the Accuracy of a Regional, Map-Based Estimate of Forest Biomass, Surveys in Geophysics, 40, 1001–1015, https://doi.org/10.1007/s10712-019-09507-1, 2019.

5. It is not reasonable to use bbox for calculating averaged or total SOC stock for specific geopolitical regions. Please use maps for masking geopolitical regions. Also, since some of the categorical maps are overlapping each other, for example, agriculture and fire-prone areas, I have an extra suggestion that you can prepare a global map to depict different categories you discussed with different colors, so readers will have a better idea.

**Author Response:**

Thank you for the suggestions. We have replaced all bounding boxes with GeoJSON masks and recalculated the statistics for geopolitical regions.

To address overlapping categories, we prepared a reprojected and combined figure showing fire-prone and agricultural areas for clarity and improved readability:

**Figure 6.** Soil organic carbon stocks (t C/ha) at 0-30 cm (SOC30) for fire-prone and agricultural regions, showing fire-prone areas (≥1 burned day per year; MODIS/061/MDC64A1, 2000-2023) (a) and agricultural land extent (LGRIP v001; Teluguntla et al., 2023) (b).

6. The data product contains lots of missing points for no reason. Shall double check your uploaded geotiff files or explain the reason why you have these missing values.

**Author Response:**

Thank you. We reviewed the dataset and can confirm that all points are present. The apparent gaps may be due to the COG format; some areas may appear empty at certain zoom levels, but all data points are visible when zoomed in.

**Additional points:**

Line 158: Explain how you bin the samples, by SOC stock?

**Author Response:**

Thank you for your comment. We have added further clarification in the revised manuscript, which now reads:

"We applied a randomized subsampling approach to national and regional maps based on SOC values. SOC pixels were stratified into 10 quantile-based bins, from lowest to highest, with each bin containing approximately the same number of data points. A fixed number of samples per bin (10 bins; 50 samples per bin) were then randomly drawn, to get an equal representation across the full SOC distribution."

**Line 173:** Shall explain: D is the soil layer depth (cm), and equals to 30cm for topsoil and 70cm for subsoil in this study?

**Author Response:**

Thank you. We have updated section 2.1.4 of the manuscript to clarify this. It now reads:

"We produced maps for two standard depth intervals commonly used in global SOC mapping: 0-30 cm and 0-100 cm."

**Line 331:** R square is a metric to show how large a fraction of the variance of a dependent variable explained by the independent variable. Since it is a fraction, the relative magnitude of SOC from dependent and independent variables cancels out each other, and does not affect the R square. Please revise or remove your conclusion.

**Author Response:**

Thank you. As we have refined the modeling approach by stratifying across distinct biomes and training separate machine learning models for each, this statement has been removed from the manuscript.

**Line 355:** What are these underrepresented regions?

**Author Response:**

Thank you. We have updated the manuscript, which now reads:

"Critical peatland complexes have been largely overlooked in prior global maps, most notably those of the Congo Basin, now recognized as the world's largest tropical peatland (Dargie et al., 2017; UNEP, 2022). The extensive Brazilian and Peruvian Amazon peatlands are also underrepresented in previous datasets despite growing evidence of their significant extent and carbon densities (Draper et al., 2014; Hastie et al., 2022, 2024)."

Line 386: Delete one "and"

Author Response: Done.

Line 423: Which data you used for global fire?

**Author Response:**

Thank you for the comment. In the Discussion section, we clarified the use of MODIS products for fire frequency. Additionally, we expanded **Methods Section 2.2**. The fire raster used as an input layer was derived from the MODIS/061/MCD64A1 dataset for 2000–2023 and represents the average number of burned days per pixel per year. This raster was then converted into a binary fire mask, with pixels having an average annual burned days ≥1 assigned a value of 1 (fire-prone) and all others assigned 0 (not fire-prone). Zonal statistics were subsequently computed using this binary mask.

**Line 442:** "Our value at 100 cm is similar to the carbon density of 361 t C/ha found by Sanderman et al. (2018)." But authors mention that total SOC stock to be 45% less from Sanderman's study compared to this work, with similar mangrove extent from both studies. Need to clarify.

**Author Response:**

Thank you for this helpful comment. We have clarified this section of the manuscript. In our study, we report the following at 100 cm depth:

- SOC stock = 5.07 ± 0.27 Pg C
- SOC density = 346 ± 19 t C/ha
- Mangrove extent = 15 Mha

**Sanderman et al. (2018) reported:**

- SOC stock = 6.4 Pg C
- SOC density = 361 t C/ ha
- Mangrove extent = 13.8 Mha

**We have revised the manuscript to read:**

"Our SOC estimates at 100 cm (5.07 Pg C) are slightly lower than Sanderman et al. (2018) (6.4 Pg C), despite using a larger mapped extent. This difference arises from lower carbon densities in our dataset (346 t C/ha vs. 361 t C/ha) and may also result from differences in spatial resolution, aggregation methods and methodological factors, such as ground-truth datasets and the application of depth functions."

**Line 464:** I'm more interested in the peatland SOC stock between northern peat and tropical peat, not SOC stock from Northern-Hemisphere, since you discuss these two separated regions later.

**Author Response:**

Thank you for the comment. The section has been updated accordingly. The revised text now reads in **Section 4.1.2.**:

"Northern peatlands store the bulk of peatland carbon, with 140 Pg C (85 % of peatland SOC) at 30 cm and 327 Pg C (80 %) at 100 cm across 759 Mha, highlighting the importance of high-latitude ecosystems in carbon storage. Tropical peatlands, despite their smaller extent (134 Mha), contribute 23 Pg C (14 % of peatland SOC) at 30 cm and 81 Pg C (20 %) at 100 cm, reflecting their high carbon density."

**Line 470:** How did you combine these datasets? By calculating the maximal extent that any dataset shows peatland coverage? Also, what threshold did you use (e.g., 1% of the grid cell)?

**Author Response:**

Thank you for the comment. We combined the peatland datasets by prioritizing the higher-resolution regional datasets (30 m) for the Congo Basin and Amazon/Peru regions over the coarser-resolution global peatland dataset (GPM2.0). We updated **Section 2.5** of the manuscript, which now reads:

"To improve regional accuracy, we integrated high-resolution peatland datasets for the Congo Basin and Peruvian Amazon. These higher-resolution regional datasets were prioritized in overlapping areas, such that where both global and regional datasets

were available, the regional data replaced the global data, while areas outside these regions retained the GPM2.0 coverage."

**Line 481:** Did you calculate the peatland extent or you used a dataset? I got the feeling that you obtained the peatland extent through a dataset but you claimed this as "finding", which means the peatland extent is the output of your trained model? Please correct.

**Author Response:** The peatland extent is not an output of our work. We have updated all sections to clarify the source of the dataset: Global Peatland Assessment (GPM2.0) (UNEP, 2022).

**Line 512:** "Mapping peatlands across the Amazon Basin has largely depended on modelling approaches". I'm still curious about how you model the peatland extent. Is it just constrained by several datasets?

**Author Response:**

Thank you. To clarify, peatland extent is not an output of our model. We use the Global Peatland Assessment (GPM2.0) (UNEP, 2022) to define peatland areas. The sentence in Line 512 refers to previous studies that modelled peatland extent; our work does not model peatland extents but relies on the GPM2.0 dataset. We have clarified this in the revised manuscript.

**Figure 5.** It may be better to have another subplot to the right showing relative uncertainty (% of uncertainty to the SOC stock)

**Author Response:** Thank you. We have added regional uncertainties throughout the revised manuscript, as well as in **Figure 1** and **Table 3**.

**Figure 6.** What is the mask map you used for fire-prone land?

**Author Response:**

Thank you for the comment. We have added a reference to the MODIS product used to identify fire-prone land in the caption of **Figure 6**. The fire mask was derived from the MODIS/061/MCD64A1 dataset (2000–2023), where pixels with an average annual burned days ≥1 were classified as fire-prone and all others as not fire-prone.

**Line 537:** "This represents 13% and 12% of the global SOC total". This is interesting to show a large difference from Pellegrini's work on Nature Geosciences. My understanding is still that your fire-prone area mask is much smaller compared to the previous one. Please explain.

**Author Response:**

Thank you. In consolidating the discussion section of the revised manuscript, this section has been removed. We also acknowledge that the estimated SOC stocks largely depend on the definition of fire-prone areas, which varies between studies. The revised **Section 4.2.1** of the manuscript now reads:

"Fire-prone areas were mapped from the MODIS/061/MDC64A1 dataset (2000-2023), with pixels averaging  $\geq$ 1 burned day per year classified as fire-prone. [...] From 2000 to 2023, fires affected an average of ~2,107 Mha of land per year, encompassing an estimated 134  $\pm$  2 Pg C at 30 cm and 340  $\pm$  5 Pg C at 100 cm, equivalent to 13% and 12% of global SOC, respectively."

Line 562: I have a feeling that lots of discussion in this manuscript is not closely related to the data product itself. For example, "In tropical forests, fires can generate feedback loops as they alter forest understory fuels, making forests more vulnerable to further fire degradation (Dwomoh & Wimberly, 2017; Wimberly, 2024). The impact of fire on long-term carbon stocks remains under investigation." stated the importance of understanding how aboveground carbon stock and residence time respond to wildfire, which is not closely relevant to your fine resolution SOC stock dataset. I would suggest either shortening or removing unnecessary discussions. Also see my major concern #3.

**Author Response:** Thank you for this note. As outlined in our response to Comment #3, we have revised the Results and Discussion to maintain focus on the SOC stock dataset.

**Line 615:** "We find that 17% of fire-prone land (351/2,109 Mha) is located in agricultural zones.". Here I found evidence that you have overlaid analysis from different categories. This may cause potential confusions so I would suggest clearly defining the boundary of each category in a map. See my major concern #5.

**Author Response:**

Thank you. We have provided further clarification in the manuscript and added the combined **Figure 6**. The line now reads:

"Based on MODIS fire data (2000-2023) and LGRIP v001 (circa 2015), fires on agricultural land account for 17% of global fire-prone areas (351/2,107 Mha), containing 26  $\pm$  1 Pg C at 30 cm and 72  $\pm$  2 Pg C at 100 cm."

**Line 631:** "bbox: -25.136719, -34.597042, 55.722656, 38.822591 degrees" Does this mean your calculated statistics for Africa based on this bbox extent? Or you accounted for the whole geographical Africa? Since Africa is a geopolitical region with a defined extent, you shall use a certain regional map as a mask.

**Author Response:**

Thank you. We have recalculated the statistics using a continent-level GeoJSON mask rather than the bounding box.

**Line 667:** "Our findings further confirm that agricultural activities place substantial pressure on the region's natural peatlands." I doubt a one time snapshot of SOC product itself can show how agricultural activities place substantial pressure on peatland. Here is another example that discussions are not closely related to your data product and not enough reasoning in your statement. Either simplify/remove or present your complete logical reasoning on how you "confirmed" your conclusions.

**Author Response:**

Thank you. We have corrected our statement. The manuscript now reads:

"Agricultural land (derived from LGRIP 2015) occupies 16 Mha, and fire-prone land (derived from MODIS, 2000-2023) covers 8 Mha, with a 5 Mha overlap, suggesting that 41% of peatlands coincide spatially with agricultural or fire-prone zones."

**Line 689:** "Our map highlights high fire activity in the Matopiba region, likely linked to land clearings." Another example of weak relevance to your SOC stock data product. I doubt you can find evidence of land clearings from your SOC stock data product. Please revise.

**Author Response:**

Thank you for the comment. The revised section now reads:

"In Brazil's Cerrado, our model estimates that soils store  $8.7 \pm 0.2$  Pg C at 30 cm and  $28.1 \pm 10.8$  Pg C at 100 cm across 204 Mha. Although the Cerrado contains moderate SOC stocks compared to other biomes, 38% of the region is classified as fire-prone and 35% as agricultural land, with a 63% spatial overlap with either category, based on LGRIP 2015 and MODIS 2000-2023 datasets (Figure 8). Fire-prone areas contain an estimated  $3.3 \pm 0.1$  Pg C (30 cm) and  $10.0 \pm 0.6$  Pg C (100 cm) across 77 Mha, and agricultural areas contain  $3.1 \pm 0.1$  Pg C (30 cm) and  $9.4 \pm 0.6$  Pg C (100 cm) across 72 Mha. Fire-prone areas and agricultural land show larger SOC contrasts at 100 cm than at 30 cm compared to other areas, which may reflect differential SOC dynamics with depth. High fire activity in regions such as Matopiba could be associated with rapid agricultural expansion. These patterns suggest a combined role of fire and land use in shaping SOC dynamics (LGRIP 2015; MODIS, 2000-2023)."

**Line 703:** "Our data, accounting for spatial variation, suggests that fire impacts on soil carbon in the Cerrado may be mostly noticeable in deeper soils." Authors should also add "fire and agriculture impacts" here.

**Author Response:** We have updated the sentence to include both fire and agriculture impacts. The revised text now reads:

"Fire-prone areas and agricultural land show larger SOC contrasts at 100 cm than at 30 cm compared to other areas, which may reflect differential SOC dynamics with depth."

Fig S3. Missing colorbar.

**Author Response:**

Thank you. The figure has been updated and is now **Figure 2**, which includes a colorbar and shows the performance of the revised biome-specific models.

**Figure 2.** Predicted vs. observed SOC density (t C/ha; full dataset) of 14 biome- and ecosystem-models at 30 cm: (TMB) Tropical and subtropical moist broadleaf forest; (TDB) Tropical and subtropical dry broadleaf forests; (TCF) Tropical and subtropical coniferous forests; (TeBF) Temperate broadleaf and mixed forests; (TeCF) Temperate coniferous forest; (BoF-TUN) Boreal forests/taiga and tundra; (MG) Mangroves (Global Mangrove Watch, 2020); (TrG) Tropical and subtropical grasslands, savannas, and shrublands; (TeG) Temperate grasslands, savannas, and shrublands; (FGr) Flooded grasslands and savannas; (MtG) Montane grasslands and shrublands; (MeF) Mediterranean forests, woodlands, and scrub; (DES) Deserts and xeric shrublands; (P) Peatlands (UNEP, 2022). Extreme values outside axis limits are omitted for comparability.

---

## Author Comment (AC4)

**Reviewer 4:**

**General Evaluation:**

This manuscript presents a 100-meter resolution global soil organic carbon (SOC) map at 30 cm and 100 cm depths, built using a harmonized collection of over 120,000 point measurements and remote sensing covariates. The authors use random forest (RF) models, with special treatment of peatlands and mangroves through ecosystem-specific models, and provide uncertainty maps based on bootstrap ensembles.

The development of a globally consistent, fine-resolution SOC map is highly relevant and potentially valuable for applications in carbon accounting, land restoration planning, and natural climate solutions. However, while the dataset is promising in scope and spatial granularity, the manuscript currently lacks structural completeness, spatial transparency, and modeling rigor necessary for scientific reproducibility and policy relevance.

I recommend major revision before the manuscript is considered for publication. Below are my detailed comments.

**Author Response:**

We thank the reviewer for their constructive feedback. In response, we have revised our manuscript and refined the modeling approach stratifying across 14 distinct biomes and ecosystems. We trained separate machine learning models for each, allowing for more localized SOC predictions. Updated raster data for the revised SOC predictions have been uploaded to the Zenodo repository.

**Major Comments**

**1. Biome- and Region-specific SOC Estimates, Uncertainty, and Validation Are Missing**

While the authors emphasize the use of biome-specific models and regionally tuned data inputs (e.g., for mangroves, peatlands), they do not provide SOC estimates, confidence intervals (CI), or model performance metrics at these spatial units. Readers cannot evaluate whether modeling SOC separately by biome or region has indeed improved performance.

Additionally, there is no validation of these outputs against either prior SOC products (e.g., GSOCmap, SoilGrids, WISE30sec) or independent in-situ observations within these biomes or regions.

**Recommendation:**

- Present SOC maps, uncertainty maps, and model metrics stratified by biome and/or key geographic regions (e.g., Amazon, SE Asia, Congo Basin).
- Compare SOC estimates for each biome with existing datasets and, if available, ground truth values.
- Consider including these outputs in main figures or supplementary materials.

**Author Response:**

Thank you for these important points. We have addressed them as follows:

- Biome-level performance metrics: Table 2 presents the performance of the biome-specific RF models at 30 cm and 100 cm. Figures 2 and 3 show predicted vs. observed SOC density per biomes. Results are discussed in Section 3.4 of the Results section (Final model performance across biomes).
- Biome-level SOC estimates and associated total uncertainty: Table 3 reports biome area, biome-level average SOC density (t C/ha) and SOC stocks (Pg C), and the associated total uncertainty at 30 cm and 100 cm depths. Total uncertainty combines model variance and residual variance, as described in Section 2.10. Results are discussed in Section 4.1.2.
- Comparison of biome-level SOC estimates with existing datasets: Figure 1 compares our global SOC output with seven widely used global SOC maps (WISE30secv.3, HWSDv1.21, GSDE, GSOCmap v1.5, Sanderman et al. (2017) and SoilGrids250m v.1 and v.2). Table 3 provides biome-level comparisons of SOC estimates with existing datasets. Results are discussed in Section 4.1.1. (Comparison with existing global maps).
- **Uncertainty: Figure 1** and **Table 3** summarize total propagated uncertainty in biomelevel SOC stocks.

**2. Model Accuracy is Low and Methodological Alternatives Are Not Explored**

The global model shows modest predictive accuracy ( $R^2 = 0.35-0.38$ ) and suffers further degradation post bias-correction. Yet, only a single modeling method - random forest - is used, with no evaluation of alternative algorithms or ensemble strategies.

This is concerning given the complexity of SOC drivers and the goal of high-accuracy mapping at fine resolution.

**Recommendation:**

- Compare multiple modeling approaches (e.g., XGBoost, LightGBM, Cubist) and report their relative performance.
- Explore ensemble or stacked models to increase generalization and reduce bias.

• Report R2, RMSE, and MAE for each biome or region and for each model tested.

**Author Response:**

Thank you for this important comment. We added the following methods section to the manuscript. It details our model selection approach, the evaluation of alternative algorithms and their relative performance.

**2.5 Model selection and biome-level framework**

"We initially developed a single global model, with test  $R^2$  values of 0.35 at 30 cm and 0.38 at 100 cm depth (before histogram-based bias correction; model trained using an 80/20 split and evaluated on the held-out test set). We tested alternative global model configurations. The initial global Random Forest model with 300 trees and no hyperparameter tuning achieved  $R^2$  = 0.35. Adding latitude and longitude as predictors slightly increased  $R^2$  to 0.37. Increasing the number of trees to 500 without latitude and longitude yielded the same  $R^2$  of 0.37. A Gradient Boosted Trees (GBT) model performed worse ( $R^2$  = 0.29), and hyperparameter tuning (i.e. two trials) increased performance only marginally to  $R^2$  = 0.34.

We then explored spatially stratified models based on different levels of ecological classification, using the WWF biomes classification, which delineates the following biomes globally (Olson et al., 2001): TMB-Tropical and subtropical moist broadleaf forests; TDB-Tropical and subtropical dry broadleaf forests; TCF-Tropical and subtropical coniferous forests; TeBF-Temperate broadleaf and mixed forests; TeCF-Temperate coniferous forest; BoF-Boreal forests/taiga; TUN-Tundra; TrG-Tropical and subtropical grasslands, savannas, and shrublands; TeG-Temperate grasslands, savannas, and shrublands; FGr-Flooded grasslands and savannas; MtG-Montane grasslands and shrublands; MeF-Mediterranean forests, woodlands, and scrub; DES-Deserts and xeric shrublands.

For each biome, we tested model performance when biomes were subdivided by continent or region (e.g., TMB in South America vs. Africa vs. Asia). While this biome-by-continent approach often produced high  $R^2$  values in certain regions, it also introduced considerable variability and data imbalance. Some biome-continent combinations had strong performance (e.g.,  $R^2 > 0.7$ ), while others suffered from limited sample size or overfitting. For example, TMB reached  $R^2$  of 0.53 in South America, 0.77 in Africa, and 0.49 in Asia and TeBF achieved  $R^2$  of 0.31 in the Americas, 0.85 in Asia, and 0.47 in Europe. Based on these findings, we selected a consistent set of 12 biome models that maintained ecological specificity while preserving sufficient data volume and geographic breadth for each model."

Regarding histogram-based bias correction, we clarified the following in **Section 2.6:**

"The biome-specific approach inherently reduces systematic bias, allowing each model to better capture local patterns. Consequently, no additional bias correction (i.e. histogram-based bias correction) was applied, and the reported values reflect the direct outputs of the biome-specific models."

Further, we included model performance metrics (R2, RMSE, and MAE) for each biome model in **Tables S9** and **S10**. These metrics are reported in the main text and in the Supplementary Information.

| Biome/Ecosystem soc30
Forests                             |         | Train
Count | Train R 2 | Test
Count | Test R² | Full
Count | Full R 2 |
|--------------------------------------------------------------|---------|----------------|----------------------|---------------|---------|---------------|---------------------|
| Tropical and subtropical moist broadleaf forests             | TMB     | 7,864          | 0.84                 | 1,966         | 0.62    | 9,830         | 0.84                |
| Tropical and subtropical dry broadleaf forests               | TDB     | 2,750          | 0.73                 | 688           | 0.52    | 3,438         | 0.74                |
| Tropical and subtropical coniferous forests                  | TCF     | 987            | 0.78                 | 247           | 0.43    | 1,234         | 0.79                |
| Temperate broadleaf and mixed forests                        | TeBF    | 19,947         | 0.73                 | 4,988         | 0.33    | 24,935        | 0.72                |
| Temperate coniferous forest                                  | TeCF    | 7,055          | 0.68                 | 1,765         | 0.43    | 8,820         | 0.71                |
| Boreal forests / taiga and tundra                            | BoF-TUN | 3,675          | 0.59                 | 920           | 0.12    | 4,595         | 0.60                |
| Mangroves - based on Global Mangrove Watch (2020) extent     | MG      | 1,261          | 0.79                 | 316           | 0.57    | 1,577         | 0.80                |
| Grasslands and shrublands                                    |         |                |                      |               |         |               |                     |
| Tropical and subtropical grasslands, savannas and shrublands | TrG     | 3,259          | 0.82                 | 815           | 0.65    | 4,074         | 0.83                |
| Temperate grasslands, savannas and shrublands                | TeG     | 6,932          | 0.73                 | 1,735         | 0.49    | 8,667         | 0.74                |
| Flooded grasslands and savannas                              | FGr     | 1,038          | 0.77                 | 260           | 0.42    | 1,298         | 0.76                |
| Montane grasslands and shrublands                            | MtG     | 334            | 0.93                 | 84            | 0.12    | 418           | 0.73                |
| Other biomes                                                 |         |                |                      |               |         |               |                     |
| Mediterranean forests, woodlands and scrub                   | MeF     | 5,632          | 0.79                 | 1,410         | 0.43    | 7,042         | 0.79                |
| Deserts and xeric shrublands                                 | DES     | 7,232          | 0.70                 | 1,809         | 0.33    | 9,041         | 0.70                |
| Peatlands - based on UNEP classification (2022)              | Р       | 2,697          | 0.66                 | 675           | 0.22    | 3,372         | 0.67                |

| Biome/Ecosystem soc30                                        |         | Train  | Train | Test  | Test | Full   | Full |
|--------------------------------------------------------------|---------|--------|-------|-------|------|--------|------|
| Forests                                                      |         | Count  | RMSE  | Count | RMSE | Count  | RMSE |
| Tropical and subtropical moist broadleaf forests             | TMB     | 7,864  | 38    | 1,966 | 67   | 9,830  | 39   |
| Tropical and subtropical dry broadleaf forests               | TDB     | 2,750  | 42    | 688   | 44   | 3,438  | 40   |
| Tropical and subtropical coniferous forests                  | TCF     | 987    | 20    | 247   | 33   | 1,234  | 20   |
| Temperate broadleaf and mixed forests                        | TeBF    | 19,947 | 60    | 4,988 | 100  | 24,935 | 62   |
| Temperate coniferous forest                                  | TeCF    | 7,055  | 84    | 1,765 | 111  | 8,820  | 80   |
| Boreal forests / taiga and tundra                            | BoF-TUN | 3,675  | 143   | 920   | 229  | 4,595  | 144  |
| Mangroves - based on Global Mangrove Watch (2020) extent     | MG      | 1,261  | 25    | 316   | 34   | 1,577  | 24   |
| Grasslands and shrublands                                    |         |        |       |       |      |        |      |
| Tropical and subtropical grasslands, savannas and shrublands | TrG     | 3,259  | 31    | 815   | 47   | 4,074  | 31   |
| Temperate grasslands, savannas and shrublands                | TeG     | 6,932  | 35    | 1,735 | 40   | 8,667  | 34   |
| Flooded grasslands and savannas                              | FGr     | 1,038  | 45    | 260   | 44   | 1,298  | 43   |
| Montane grasslands and shrublands                            | MtG     | 334    | 9     | 84    | 54   | 418    | 21   |
| Other biomes                                                 |         |        |       |       |      |        |      |
| Mediterranean forests, woodlands and scrub                   | MeF     | 5,632  | 24    | 1,410 | 35   | 7,042  | 24   |
| Deserts and xeric shrublands                                 | DES     | 7,232  | 28    | 1,809 | 35   | 9,041  | 27   |
| Peatlands - based on UNEP classification (2022)              | Р       | 2,697  | 127   | 675   | 216  | 3,372  | 129  |

| Biome/Ecosystem soc30                                        |         | Train  | Train | Test  | Test | Full   | Full |
|--------------------------------------------------------------|---------|--------|-------|-------|------|--------|------|
| Forests                                                      |         | Count  | MAE   | Count | MAE  | Count  | MAE  |
| Tropical and subtropical moist broadleaf forests             | TMB     | 7,864  | 15    | 1,966 | 27   | 9,830  | 15   |
| Tropical and subtropical dry broadleaf forests               | TDB     | 2,750  | 16    | 688   | 26   | 3,438  | 16   |
| Tropical and subtropical coniferous forests                  | TCF     | 987    | 13    | 247   | 24   | 1,234  | 13   |
| Temperate broadleaf and mixed forests                        | TeBF    | 19,947 | 23    | 4,988 | 39   | 24,935 | 23   |
| Temperate coniferous forest                                  | TeCF    | 7,055  | 35    | 1,765 | 59   | 8,820  | 35   |
| Boreal forests / taiga and tundra                            | BoF-TUN | 3,675  | 67    | 920   | 108  | 4,595  | 68   |
| Mangroves - based on Global Mangrove Watch (2020) extent     | MG      | 1,261  | 14    | 316   | 25   | 1,577  | 14   |
| Grasslands and shrublands                                    |         |        |       |       |      |        |      |
| Tropical and subtropical grasslands, savannas and shrublands | TrG     | 3,259  | 11    | 815   | 18   | 4,074  | 10   |
| Temperate grasslands, savannas and shrublands                | TeG     | 6,932  | 14    | 1,735 | 22   | 8,667  | 13   |
| Flooded grasslands and savannas                              | FGr     | 1,038  | 19    | 260   | 28   | 1,298  | 18   |
| Montane grasslands and shrublands                            | MtG     | 334    | 6     | 84    | 15   | 418    | 8    |
| Other biomes                                                 |         |        |       |       |      |        |      |
| Mediterranean forests, woodlands and scrub                   | MeF     | 5,632  | 13    | 1,410 | 22   | 7,042  | 13   |
| Deserts and xeric shrublands                                 | DES     | 7,232  | 11    | 1,809 | 17   | 9,041  | 10   |
| Peatlands - based on UNEP classification (2022)              | Р       | 2,697  | 58    | 675   | 103  | 3,372  | 59   |

**Table S9.** Biome-specific model performance metrics for 0-30 cm soil organic carbon (SOC30).

**3. Data Sparsity in Key Regions Remains Unresolved**

Although the manuscript compiles an impressive collection of point data, it is unclear whether it significantly improves training sample density in previously under-sampled areas (e.g., SE Asia, Amazon peatlands, Congo Basin, boreal permafrost zones).

**Recommendation:**

- Provide maps or histograms of sample density by region or biome.
- Quantify the number and proportion of new samples added to underrepresented areas.
- Compare data coverage with existing global SOC datasets to clarify how this product advances spatial completeness.

**Author Response:** Thank you. We have taken several steps to address this.

To show sample density by biome, we have added **Figure S2** to show the distribution of ground-truth SOC observations across biomes. The figure illustrates the global extent of biomes (in orange) and the locations of SOC measurements (in dark grey) at 30 cm and 100 cm depths, and includes summary statistics (number of observations, mean SOC, and standard deviation) for each biome. A subset of representative maps is shown below. **Figure S2** presents the full set of 27 maps in the Supplementary Material.

**Tropical & subtropical moist broadleaf forests (TMB)**

9830 points; Mean SOC: 71 tC/ha; Std: 99 tC/ha

2155 points; Mean SOC: 436 tC/ha; Std: 458 tC/ha

**Tropical and subtropical grasslands, savannas, and shrublands (TrG)**

4074 points; Mean SOC: 47 tC/ha; Std: 75 tC/ha

1406 points; Mean SOC: 160 tC/ha; Std: 327 tC/ha

**Figure S2 (subsection).** Global distribution of biomes and ground-truth soil organic carbon (SOC) data with summary statistics. This figure shows the spatial extent of the biomes in orange and the locations of ground-truth SOC observations for 0-30 cm and 0-100 cm depths, in grey. The number of observations, mean SOC density (t C/ha), and standard deviation are reported for each biome and depth.

Further, we added the following clarification to **Methods Section 2.8**:

"In this study, we integrated multiple global and large regional soil datasets, including Europe-LUCAS 2018, ISCN, ISRIC-WISEv3, RaCA, and the WoSIS snapshot 2023. While previous global SOC maps such as SoilGrids v1/v2, Sanderman et al. (2017), GSOCmap v1.5, WISE30sec, GSDE and HWSD v1.21 relied on earlier versions of these datasets, our study incorporates the latest WoSIS release and additional national and ecosystem-specific datasets (Table S2), many published after 2017. By integrating these diverse inventories, we improve spatial completeness and enhance representation of SOC across biomes and regions (Figure S1)."

We added **Figure S1** to show the global spatial coverage of the main large datasets (Europe-LUCAS 2018, ISCN, ISRIC-WISEv3, RaCA, and WoSIS) and the coverage of regional datasets.

**Figure S1.** Locations of ground-truth data points for (a) soil depth 0–30 cm (SOC30) and (b) soil depth 0–100 cm (SOC100). Grey points show large regional and global datasets (SOC30: Europe-LUCAS 2018, ISCN, ISRIC-WISEv3, RaCA, WoSIS 2023; SOC100: ISCN, ISRIC-WISEv3, RaCA, WoSIS 2023), while red points indicate additional ground-truth datasets used in this study, excluding sampled gridded datasets.

**4. No Connection Between Static SOC Map and Dynamic Carbon Policy Use**

The authors claim their product supports natural climate solutions and carbon removal strategies. However, the map reflects a static snapshot of SOC conditions and does not account for management interventions or disturbance effects.

**Recommendation:**

- Clearly position this dataset as a static SOC baseline and discuss its potential role in MRV (Monitoring, Reporting, Verification) systems.
- Provide examples or scenarios where SOC stocks are compared under different management or fire regimes using model predictions.
- Alternatively, simulate SOC distributions under no-fire or no-agriculture scenarios using the trained models to demonstrate potential use in change detection or policy design.

**Author Response:**

We greatly appreciate the reviewer's comments and concerns. It is clear that our map serves as a one-time baseline or benchmark for soil organic carbon (SOC) stock at two depths. To clarify the use cases of the SOC maps, we have added the following sections to the manuscript.

**4.3.1 Baseline map for research and monitoring**

"We present a high-resolution (100 m) soil organic carbon map at two depths, representing baseline SOC stocks circa 2022. Soil profile data collected from the 1950s to the 2020s were combined with contemporary remote sensing covariates, using Landsat 8 bands to capture landscape conditions around 2022. The use of long-term soil datasets aligns with previous global mapping efforts, such as SoilGrids v1-v2 and GSOCmap v1.5, which integrate multi-decadal data to capture large-scale spatial patterns. The primary goal of this work was to represent SOC variation across diverse landscape gradients with a resolution fine enough to capture local differences while maintaining global coverage. The resulting map can support applications in land management, nature-based climate solutions, and future assessments of SOC dynamics.

SOC data inevitably span decades, while Earth observation covariates reflect more recent conditions. In landscapes without major human disturbances, SOC is assumed to remain relatively stable over time, so historical soil measurements can reliably represent baseline stocks when combined with current remote sensing data. In areas affected by landuse change, degradation, or recovery, such as forests regrowing after wildfire, agricultural rotations, or restored wetlands, SOC may have changed since sampling. For instance, samples from western CONUS encompass forests at different successional stages recovering from events such as wildfires or storms. Remote sensing captures spectral signals related to land cover and vegetation recovery, helping to contextualize historical soil measurements. Consequently, the SOC map reflects not only static soil conditions but also

the imprints of temporal changes across landscapes. To address potential temporal discrepancies at the pixel level, we generated an uncertainty map alongside the SOC estimates, allowing users to assess confidence in areas where SOC may have changed substantially since sampling.

Our dataset integrates both natural and human-modified landscapes, including agricultural areas, managed forests, wetlands, and coastal ecosystems such as mangroves. By combining long-term soil measurements with remote sensing covariates and machine learning models, the map captures both spatial variation in SOC and the influence of historical land-use changes and recovery processes. Overall, the high-resolution SOC maps represent a valuable tool for carbon management and conservation. The maps enable fine-scale detection of spatial variability, supports assessments of land-use impacts, and provides a foundation for monitoring SOC dynamics over time, even in regions with limited historical soil data."

**4.3.2 Policy and management applications**

"The baseline SOC map serves as a benchmark with broad relevance across scientific, management, and policy applications. For carbon stock reporting, it enables estimation of SOC across regions, land cover types, and management units, with particularly high precision for areas larger than 1 hectare where pixel aggregation reduces uncertainty. At the policy level, the map provides national and sub-national jurisdictions with a means to assess soil carbon storage and sequestration potential, accounting for spatial variability across land uses and vegetation successional stages. Such information supports the design and implementation of land-use and restoration policies aimed at improving soil carbon conditions. The map also facilitates participation in carbon markets, where project developers require reliable SOC stock data to estimate emissions reductions or removal factors, including avoided losses and potential accumulation rates in disturbed versus natural areas. In addition, the high-resolution benchmark map at multiple depths can be integrated into biogeochemical models to quantify potential changes in soil carbon. Building on this work, we plan to combine repeated SOC measurements with process-based models to advance understanding of SOC dynamics. By leveraging this baseline, carbon management strategies can be refined and climate policies better informed."

**5. No Analysis of Predictor Effects Across Biomes or Regions**

The manuscript includes no interpretation of model behavior through partial dependence plots (PDPs), SHAP values, or marginal effect visualizations. This is a major weakness given the low model R2 and the ecological complexity of SOC formation.

Understanding how each covariate influences SOC predictions is critical to evaluate whether the model captures realistic biogeochemical relationships or is driven by spurious correlations.

**Recommendation:**

- Provide variable importance rankings and partial dependence plots for key predictors (e.g., pH, CEC, clay, NDVI, temperature).
- Stratify these analyses by biome or region to highlight differences in covariate effects.
- Comment on whether the model's response patterns are ecologically interpretable and consistent with known SOC mechanisms.

**Author Response:**

Thank you. We conducted additional calculations using partial dependence analysis to evaluate feature influence on SOC predictions. These analyses were stratified by biome to highlight differences in covariate effects.

Methods are detailed in the new **Section 2.7** ("Partial dependence analysis"), and results are presented in **Section 3.5** ("Feature effects on SOC predictions"), where we discuss how the models' response patterns are ecologically interpretable.

Full results are provided in **Supplementary Figure S3**.

**Minor Comments**

**1. Abstract (Lines 22–24):** Rephrase "can influence SOC by 132 Pg C…" to avoid misinterpretation as quantified changes. Suggest: An estimated 132 Pg C of SOC is located in areas affected by wildfire and 140 Pg C in agricultural areas.

**Author Response:**

Thank you for this comment. We have revised the sentence to clarify that the reported values represent SOC stocks located in fire-prone and agricultural areas. The sentence in the manuscript now reads:

"Our estimates indicate  $134 \pm 2$  Pg C and  $340 \pm 5$  Pg C sit in fire prone areas at 30 cm and 100 cm depth, and that  $140 \pm 2$  Pg C and  $384 \pm 8$  Pg C are in areas of ongoing agricultural activity at 30 cm and 100 cm depth, representing about 13% of global SOC stocks."

**2. Line 505:** If the increased Amazon SOC estimate is driven by better mapping of peatlands, please state that explicitly.

**Author Response:**

Thank you. We have revised this section of the manuscript, which now reads:

"Across the entire Amazon Basin (593 Mha), we estimate  $37 \pm 1$  Pg C at 30 cm and  $162 \pm 7$  Pg C at 100 cm, representing 4 % and 6 % of the global SOC total, respectively. Our results at 100 cm are higher than previous Amazon Basin assessments, including 47 Pg C at 100 cm (Moraes et al., 1995), 46.5 Pg C (Batjes and Dijkshoorn, 1999), and 36.1 Pg C (Gomes et al., 2019), due in part to our use of the recently updated peatland extent dataset that better captures carbon-rich areas."

**3. Line 551–555:** The difference between SOC estimates with and without fire (28 Pg C at 100 cm) is small (~1%). Confidence intervals should be reported to assess significance. Also clarify whether adding fire improved model performance.

**Author Response:**

We thank the reviewer for this comment. Regarding the first point, we no longer compare global stock predictions with and without fire as an input layer to the model. Instead, we focus on the results of the fire feature effects on SOC predictions per biome, and find large fire effects in tundra/boreal regions and in peatlands, consistent with known ecological patterns.

Regarding the second point, total uncertainty combining model variance and residual variance has been computed and added to the manuscript.

Regarding the third point, for each biome model and soil depth, we compared the model performance ( $R^2$ ) on the full dataset with and without the fire variable. Including fire as a predictor had a negligible to very small effect of ~0.01 on model performance across biomes and soil depths. The minimal change in  $R^2$  suggests that adding fire does not introduce artifacts or model instability. Thus, we retain fire as a predictor because it captures an important ecological process influencing soil carbon dynamics. This has been added to the manuscript **Section 2.2**.

4. Figures 1–5: Add consistent color bars, units (e.g., t C/ha), and labels for clarity.

**Author response:** We have revised **Figures 1–5** to include consistent color bars and clearly labeled units (t C/ha).

**5. Line 685 & 689:** The phrase "our map shows high fire/agriculture activity" may confuse readers. Clarify that these are input datasets, not part of the new SOC product.

**Author Response:**

Thank you for this comment. The agricultural land extent and fire activity are derived from external datasets (LGRIP 2015 and MODIS 2000–2023, respectively) rather than generated directly from our SOC mapping effort.

To clarify, we have revised the text to explicitly attribute these datasets to their original sources. The manuscript now reads:

"In Brazil's Cerrado, our model estimates that soils store  $8.7 \pm 0.2$  Pg C at 30 cm and  $28.1 \pm 10.8$  Pg C at 100 cm across 204 Mha. Although the Cerrado contains moderate SOC stocks compared to other biomes, 38% of the region is classified as fire-prone and 35% as agricultural land, with a 63% spatial overlap with either category, based on LGRIP 2015 and MODIS 2000-2023 datasets (Figure 8). Fire-prone areas contain an estimated  $3.3 \pm 0.1$  Pg C (30 cm) and  $10.0 \pm 0.6$  Pg C (100 cm) across 77 Mha, and agricultural areas contain  $3.1 \pm 0.1$  Pg C (30 cm) and  $9.4 \pm 0.6$  Pg C (100 cm) across 72 Mha. Fire-prone areas and agricultural land show larger SOC contrasts at 100 cm than at 30 cm compared to other areas, which may reflect differential SOC dynamics with depth. High fire activity in regions such as Matopiba could be associated with rapid agricultural expansion. These patterns suggest a combined role of fire and land use in shaping SOC dynamics (LGRIP 2015; MODIS, 2000-2023)."

**6. Line 699–703:** Discussion of fire impacts here is redundant with earlier sections. Consider consolidating to avoid repetition.

**Author Response:** Thank you. We have reworked the manuscript outline and consolidated the discussion of fire impacts into a single coherent section. The revised outline now reads:

**4.2 Land-use and disturbance**

- 4.2.1 Global fires and agriculture (now includes a consolidated global fire discussion)
- 4.2.2 Regional SOC stocks relative to land-use and fire (focuses on regional distribution of fire and agriculture)

---

## Author Comment (AC5)

**Reviewer 5:**

The production of global maps of SOC stocks is valuable research and the dataset is of high interest for the global community. This manuscript is therefore timely. The current methodology and datasets used to build the maps, however, have several very fundamental issues that need to be resolved before any publication is made. For several of these issues, well-accepted solutions exist in the literature, and it is not clear if the authors are proposing something new (in which case, it should first be tested), or if they simply decided not to account for past developments. The manuscript suggests an important lack of familiarity with the current state of the art in soil science and digital soil mapping. All of the above is common knowledge and not prone to any specific discussion in the field.

I focused on the very major comments because I see that the other reviewers mentioned several minor issues.

The key issues are (explained in more details below):

- 1. Predictions should be made for depth intervals, not exact depths.
- 2. Depth harmonization (e.g. mass-preserving splines) is required to address differing sample supports.
- 3. Input datasets (e.g. RaCA, Australia) require quality checks and better sources.
- 4. Clarification needed on coarse fragment data and SOC measurement methods.
- 5. Modeled data should not be mixed with measured data.
- 6. Pseudo-observations should be added for areas with no SOC (e.g. deserts).
- 7. The choice of 100 m resolution should be justified.
- 8. Cross-validation, not single data splits, should be used.
- 9. Bias correction is unusual and needs justification.
- 10. Uncertainty assessment should use prediction intervals (e.g. quantile regression forest).
- 11. Check for residual autocorrelation and consider kriging residuals if needed.

**Author Response:**

Thank you for your review. We appreciate the time that you took to provide feedback on our work.

Depth intervals. In soil science, predictions are made for depth intervals, not specific depths as the authors did here for 30 and 100 cm. The concept of calculating a stock at exactly 30 cm or 100 cm depth does not make scientific sense. A stock is based on volume, i.e., a depth interval. The authors justify this in Section 2.1 by stating "... continuous soil depth functions

(Malone et al., 2009), which have shown significant variability in results." Using depth functions is the standard in soil science because soil samples are collected at different depth intervals (e.g., 0–10 cm or 5–15 cm) and need to be harmonized before use. This is a necessary pre-processing step. It becomes very clear in the next sentence of the same section why it is needed: "At a depth of 30 cm, we included 84,880 ground truth data points." What does this mean exactly? Did you merge together samples collected for any interval between 0 and 30 cm? What did you do for samples that were collected, for example, on the interval 0–40 cm? This shows that using depth harmonization is a prerequisite as much as calculating the stock for a depth interval. The justification given in Section 2.1.4 for depth harmonization does not make sense. I invite the authors to read the papers on depth harmonization in the soil science literature, where the approach given here seems outdated and not correct.

Further on this matter of the depth interval, the authors are currently mixing samples with different supports. It is not the same to estimate the SOC stocks for a sample obtained on a 0–5 cm interval compared to a sample with a 0–30 cm depth support. These are very different, yet the authors are putting everything together and considering it the same measurement. This again supports the need for using a mass-preserving spline as a basic pre-processing step for SOC stock data.

**Author Response:**

Thank you. To address these concerns, we have added further clarification to the methods **Section 2.1.4**, as follows:

"Many of the datasets were already standardized to the 0-30 cm or 0-100 cm depth intervals. For profiles with multiple layers, we only considered those with both upper and lower depth boundaries reported and combined consecutive nested layers to calculate total SOC. For the 0-30 cm interval, we included profiles that adequately covered the topsoil layer, allowing a small tolerance to maximize dataset inclusion (20-45 cm actual soil depth) while maintaining depth consistency. Profiles that were too shallow or had gaps in measurement were excluded. For deeper layers (0-100 cm), we similarly retained only profiles with complete depth coverage, again applying the small tolerance for uniformity (≥80 cm actual soil depth). Linear interpolation was applied where minor adjustments were necessary. Peatland datasets were handled separately to retain deeper measurements due to limited data availability. Samples missing depth information were excluded, and duplicate entries with identical coordinates and SOC values were removed.

To evaluate the quality of the harmonized datasets, we tested their predictive performance using biome-specific models, training each model only on samples from the corresponding biome. We calculated R2 values for biome-dataset combinations with more than 50 samples. Globally distributed datasets such as ISCN, WoSIS, and ISRIC-WISE were

the most widely represented and consistently showed high predictive reliability. At 0-30 cm, WoSIS performed best in tropical biomes (average test  $R^2$  = 0.62), while ISCN showed broad coverage with good performance in temperate biomes (average test  $R^2$  = 0.45). Similar trends were observed for the 0-100 cm depth interval, though performance variability increased with depth and region, with some datasets providing stronger signals in specific biomes. This assessment indicates that the harmonized datasets reliably capture SOC patterns at the biome level, supporting their use for biome-specific modeling and prediction."

Problems with input point datasets. There are several known problems with the datasets the authors used. Most of the US data are based on the Rapid Carbon Assessment (RaCA) dataset. This dataset is known to have several issues. First, most values in this dataset are not measured, but predicted with infrared spectroscopy, which introduces an additional source of error. Second, there is a high likelihood of issues with bulk density measurements. The authors should contact the dataset maintainers to get more information. This dataset should at least be checked carefully and harmonized to retain the highest-quality bulk density measurements. As an example, the best mapping paper so far on soil properties in the US did not incorporate the RaCA dataset as input

(https://acsess.onlinelibrary.wiley.com/doi/10.1002/saj2.20769).

**Author Response:**

Thank you. The RaCA dataset represented a relatively small portion of the training data for the biome models. At 30 cm depth, RaCA contributed roughly 15% of the data in temperate grasslands, savannas and shrublands, as well as in temperate coniferous forests, about 10% in temperate broadleaf and mixed forests, and in deserts and xeric shrublands, and minimally (1-4%) in four other biomes and absent in all remaining biomes. At 100 cm depth, it contributed ~20% in temperate coniferous forests and in deserts and xeric shrublands, ~15% in Mediterranean forests, woodlands and scrub, in temperate broadleaf and mixed forests, and in temperate grasslands, savannas and shrublands, and minimally in two other biomes.

We found that the RaCA dataset provided moderate yet positive predictive performance, reflecting meaningful spatial structure. Across the relevant biomes, the average performance (test R2) for RaCA was approximately 0.25 at 30 cm depth and 0.31 at 100 cm depth. Based on this performance, we decided to include the RaCA dataset in the models.

We have added this clarification to **Methods section 2.1.1**.

Another concern is the lack of real data for Australia. The authors randomly sampled an old map to generate data (note that this 2014 map has been updated in 2022 with entirely new predictions). This is problematic. There is a wealth of publicly available datasets in Australia

that the authors could use, available through the SoilDataFederator. Alternatively, the authors could contact the authors of the recent paper on mapping SOC stocks in Australia to obtain their pre-processed data.

**Author Response:**

Thank you for your comment. SOC in Australia is derived from a combination of ground-truth points, biome-specific models, and stratified map-based sampling. We have provided further clarification in **Methods Section 2.1.3**., which now reads:

"Soil organic carbon in Australia was derived from a combination of ground-truth points, stratified map-based sampling, and eight globally applied biome-specific models. For Australia, our dataset included recent ground-truth SOC data from WISE30sec v.3 (Batjes, 2016), WoSIS (Dec 2023), ISCN v.3 (Nave et al., 2022), MarSOC (Maxwell et al., 2023), and WHRC-TNC (Sanderman et al., 2018). Using the quantile-based bin approach previously described, we additionally sampled 500 points evenly across the full SOC range of the Australian SOC map developed by Viscarra Rossel et al. (2014). These 500 samples accounted for no more than 3% of the eight biome datasets covering Australia, providing regional detail without dominating the broader analysis."

Two major attention points are raised with the SOC stock calculation:

Coarse fragments are notoriously difficult to obtain and predict. How did the authors obtain these data? It is not mentioned anywhere except for a brief statement that "it is not available everywhere." This needs to be made very clear because it will substantially affect the SOC stock calculation.

**Author Response:**

Thank you for this comment. To address it, we have added the following clarification to the **Methods Section 2.1.4** of the manuscript:

"Across datasets, coarse fragments were inconsistently reported, but in most cases they were either explicitly provided or implicitly accounted for during SOC stock calculations. Many harmonized datasets that reported final SOC stocks did not include CF values as a separate variable, but indicated that coarse fragments were handled internally. Datasets that did not provide final SOC stocks often included CF measurements, although these were only partially available (i.e., not all data points). We adjusted SOC stocks to account for the presence of CF when possible. Overall, coarse fragments were generally considered in stock estimation, but differences in their treatment remain a source of uncertainty in global SOC datasets (Hengl et al., 2017; Poeplau et al., 2017), and ongoing efforts aim to improve consistency."

The authors ignored the difference between SOC obtained by dry combustion and the Walkley–Black method. They cite one paper to justify this, but the difference between methods can be significant. It is standard to apply a correction or otherwise account for this difference.

**Author Response:**

Thank you for this comment. To address it, we have added the following clarification to **Methods Section 2.1.4** of the manuscript:

"Soil organic carbon was assessed using different methods (e.g., dry combustion or Walkley-Black) across datasets, reflecting the extended timeline over which data were collected. Most datasets compiled in this study had already been internally harmonized using conversion factors, and we did not apply further adjustments for potential method-based discrepancies. Most datasets reported final soil carbon stocks standardized in t C/ha. Older campaigns relied more on loss on ignition (LOI) and Walkley-Black methods, potentially introducing regional differences despite conversions. In contrast, more recent datasets, which we integrated to enhance spatial coverage, were predominantly analysed using dry combustion. Thus, methodological differences remain a potential source of bias in the compiled SOC estimates; however, remaining method-related effects are expected to be relatively minor compared with the overall spatial variability captured by the model."

Mixing measured and modeled data. The use of map sampling to generate observations for model fitting is a serious problem. The authors have combined observations from measured SOC stocks (derived using various calculation methods) with values obtained from SOC stock maps. This is problematic, as the map values are already modeled predictions, not direct measurements. They are smoothed and often carry significant uncertainties. The authors should avoid this practice.

**Author Response:** We thank the reviewer for highlighting this point. We have added detail regarding the inclusion of subsampled data in **Section 2.1.3** of the methodology, which now reads:

"We analysed model performance both with and without subsamples derived from the previously described gridded datasets. The results show that, when subsamples are excluded, model performance remains largely robust, with only minor reductions of 0.01-0.02 in R² values across most biomes and ecosystems (Table S8). Montane grasslands are an exception, showing a larger drop in R² due to the low number of samples in this biome. At 100 cm depth, removing subsamples similarly preserves model performance, with only marginal changes of 0.02-0.03 in R², again with the exception of montane grasslands. These

results indicate that, although map-derived samples contribute to model training, they do not fundamentally alter overall model performance or spatial patterns. The model is therefore not overly reliant on these potentially biased samples. We include these samples to improve spatial representativeness, but we have verified that their inclusion does not introduce substantial bias in our predictions. The full biome-level results with and without subsamples are available in Table S8."

|                                                              | with subsamples |                     | without subsamples |                     |
|--------------------------------------------------------------|-----------------|---------------------|--------------------|---------------------|
| Biome/Ecosystem soc30                                        | Full Count      | Full R 2 | Full Count         | Full R 2 |
| Forests                                                      | I all Count     | ı un ix             | i un oount         | - unix              |
| Tropical and subtropical moist broadleaf forests             | 9,830           | 0.84                | 9,162              | 0.82                |
| Tropical and subtropical dry broadleaf forests               | 3,438           | 0.74                | 3,424              | 0.74                |
| Tropical and subtropical coniferous forests                  | 1,234           | 0.79                | 1,232              | 0.79                |
| Temperate broadleaf and mixed forests                        | 24,935          | 0.72                | 24,889             | 0.73                |
| Temperate coniferous forest                                  | 8,820           | 0.71                | 8,781              | 0.71                |
| Boreal forests / taiga and tundra                            | 4,595           | 0.60                | No subsamples      |                     |
| Mangroves - based on Global Mangrove Watch (2020) extent     | 1,577           | 0.80                | 1,238              | 0.79                |
| Grasslands and shrublands                                    |                 |                     |                    |                     |
| Tropical and subtropical grasslands, savannas and shrublands | 4,074           | 0.83                | 3,439              | 0.83                |
| Temperate grasslands, savannas and shrublands                | 8,667           | 0.74                | 8,448              | 0.74                |
| Flooded grasslands and savannas                              | 1,298           | 0.76                | 1,024              | 0.76                |
| Montane grasslands and shrublands                            | 418             | 0.73                | 33                 | 0.40                |
| Other biomes                                                 |                 |                     |                    |                     |
| Mediterranean forests, woodlands and scrub                   | 7,042           | 0.79                | 6,962              | 0.79                |
| Deserts and xeric shrublands                                 | 9,041           | 0.70                | 8,493              | 0.70                |
| Peatlands - based on UNEP classification (2022)              | 3,372           | 0.67                | 2,805              | 0.67                |

|                                                              | with subsamples |                     | without subsamples |                     |
|--------------------------------------------------------------|-----------------|---------------------|--------------------|---------------------|
| Biome/Ecosystem soc100
Forests                            | Full Count      | Full R 2 | Full Count         | Full R 2 |
| Tropical and subtropical moist broadleaf forests             | 2,155           | 0.90                | 1,573              | 0.87                |
| Tropical and subtropical dry broadleaf forests               | 1,322           | 0.71                | 1,295              | 0.71                |
| Temperate broadleaf and mixed forests                        | 14,516          | 0.72                | 14,503             | 0.72                |
| Temperate coniferous forest                                  | 6,919           | 0.70                | 6,873              | 0.70                |
| Boreal forests / taiga and tundra                            | 1,716           | 0.68                | No subsamples      |                     |
| Mangroves - based on Global Mangrove Watch (2020) extent     | 1,453           | 0.80                | 1,097              | 0.78                |
| Grasslands and shrublands                                    |                 |                     |                    |                     |
| Tropical and subtropical grasslands, savannas and shrublands | 1,406           | 0.90                | 1,360              | 0.90                |
| Temperate grasslands, savannas and shrublands                | 8,236           | 0.62                | 8,219              | 0.62                |
| Flooded grasslands and savannas                              | 1,034           | 0.77                | 790                | 0.75                |
| Montane grasslands and shrublands                            | 385             | 0.56                | 21                 | 0.28                |
| Other biomes                                                 |                 |                     |                    |                     |
| Mediterranean forests, woodlands and scrub                   | 1,298           | 0.83                | 1,297              | 0.83                |
| Deserts and xeric shrublands                                 | 4,431           | 0.66                | 4,370              | 0.66                |
| Peatlands - based on UNEP classification (2022)              | 1,936           | 0.77                | 1,286              | 0.75                |

**Table S8.** Model performance  $(R^2)$  and sample count per biome and ecosystem for soil organic carbon (SOC) predictions for 0-30 cm and 0-100 cm depths. Results are shown for biome-specific models trained with the full dataset and with or without the inclusion of subsampled gridded datasets.

Ignoring zero-SOC areas. The authors ignored areas that contain no SOC in soils, such as deserts. It is common procedure to add pseudo-observations in such areas to prevent the model from predicting SOC stocks where there should be none (because of smoothing effects) and to reduce uncertainty in total SOC stock estimates.

**Author Response:** We confirm that we did not include value-zero pseudo-observations in our dataset. Many desert regions are already represented in global and regional datasets with very low or value-zero SOC. We modeled deserts and xeric shrublands separately, which further limits the risk of over-prediction. The combination of these observed data and Landsat surface reflectance inputs allows these regions to be accurately represented in our global SOC map.

The specific resolution chosen (100 m) is not justified. We know that it is simply a matter of computing power and that predictions could be made at 10 m if desired. However, many researchers refrain from using such fine resolutions for global products because these products tend to perform poorly locally and can mislead soil management decisions. The resolution choice should be justified.

**Author Response:**

While satellite data are now publicly available at spatial resolutions as fine as 10 m, studying forests, land use, and soil properties at that level of detail is often problematic, unnecessary, and inefficient. A 100 m resolution (1 ha) is a more appropriate choice for several reasons:

**1. Ecological and land-use processes operate at coarser scales.**

Many ecological and land-use processes, including forest composition, land cover transitions, and management zones, typically manifest at coarser spatial scales, generally either at landscape scale or disturbance scales. When considering both natural and anthropogenic disturbance patterns, a minimum mapping unit of 1 ha (100 m) is a reasonable scale for representing vegetation and soil parameters.

**2. Soil properties exhibit meaningful variation at the hectare scale.**

Soil heterogeneity is often structured around ~1 ha units, whereas finer resolutions tend to capture noise rather than ecologically relevant patterns (McBratney et al., 2003; Delbari et al., 2011).

**3. Aggregating pixels improves prediction accuracy.**

The predictive power of remote sensing data improves significantly when aggregating pixels. At 10 or 30 m resolution, reflectance measurements are affected by calibration issues, measurement noise, satellite geometry, atmospheric conditions, georeferencing errors of a pixel and more, and small-scale heterogeneity unrelated to surface parameters. Machine learning or analytical models perform better at

aggregated scales with less reflectance errors than at the level of individual noisy pixels. Consequently, most high-resolution maps at 10 or 30 m do not achieve true pixel-level accuracy and are often mistaken for being more precise than they actually are.

It is important to distinguish between map resolution and pixel spacing; posting data at 10 or 30 m does not imply that parameters are accurately resolved at that scale, as pixel spacing for gridding and the true prediction resolution are separate concepts. The spacing of the pixels does not automatically define the scale at which the model's predictions are reliable. Our 100 m maps could be applied to finer pixel grids (10 m or 30 m) using the same model, with the only difference being computational cost. However, a 100 meter resolution provides a more ecologically appropriate and computationally efficient scale for landscape-level analyses.

**Reference:**

Delbari, M., Afrasiab, P., and Loiskandl, W.: Geostatistical analysis of soil texture fractions on the field scale, Soil and Water Research, 6, 173–189, <a href="https://doi.org/10.17221/9/2010-SWR">https://doi.org/10.17221/9/2010-SWR</a>, 2011.

McBratney, A.B., Mendonça Santos, M.L., and Minasny, B.: On digital soil mapping. Geoderma, 117(1–2), 3–52, https://doi.org/10.1016/S0016-7061(03)00223-4, 2003.

Model validation. The authors used model validation based on data splitting. Repeated cross-validation should be used instead. Data splitting can lead to accidentally high or low validation statistics depending on the split, and it might also be susceptible to fraud if one were to select a split that yields better results. It should be replaced.

**Author Response:**

Thank you. During model development, we conducted K-fold cross-validation with 50 simulations. We added further detail to **Methods section 2.6**, **Results section 3.3** and provide the results in **Table 1** of the revised manuscript.

We note that all machine learning and regression models can suffer from overfitting, particularly at the tails of predicted distributions, which is amplified when working with noisy data. Climate and remote sensing data, when used to assess SOC, are considered noisy due to the weak statistical correlation between spectral information and SOC values. This can lead to overestimation of low values and underestimation of high values, distorting the mean SOC distribution and introducing systematic errors at the extremes. To mitigate this, we use a stratified approach, training biome-specific models. This allows for more localized SOC

estimation, substantially reducing global overfitting and minimizing biases in the tails of SOC distributions.

Previously, we applied a bias-correction approach (Xu et al., 2017, 2021) to improve accuracy in the distribution tails, analogous to histogram matching. While this bias-correction algorithm significantly improves distribution bias, it does introduce a slight increase in random error at the pixel level. We removed this bias correction, accepting small residual biases (<1% in most regions), as biome-specific modeling already mitigates bias, and the impact of these residual biases on large-scale SOC estimates is minimal.

Bias correction. Applying a post-processing bias correction is, to my knowledge, never seen in digital soil mapping. This is because most models we use (geostatistics, random forest) have little to no bias. Are there even biases in the predictions? This is not common for random forest, and the explanation given by the authors on this aspect does not seem relevant to the work (particularly for users applying the map for soil carbon accounting).

**Author Response:**

In the previous version of the soil carbon map, a global histogram matching bias correction was applied to the combined outputs of the three broad models (global, mangrove, peatland) to reduce systematic over- or underestimation relative to observed values. In the current version, we modeled soil carbon separately for 14 biomes and ecosystems, with each model trained on a more homogeneous subset of data. This biome-specific approach inherently reduces systematic bias, as each model is better able to capture local ecological patterns. Therefore, no additional bias correction was applied, and model outputs are reported directly from the biome-specific models.

This clarification has been added to **Methods Section 2.6**, as follows:

"The biome-specific approach inherently reduces systematic bias, allowing each model to better capture local patterns. Consequently, no additional bias correction (i.e. histogram-based bias correction) was applied, and the reported values reflect the direct outputs of the biome-specific models."

Uncertainty assessment. The uncertainty assessment needs to be completely redone. In digital soil mapping, one is interested in a prediction interval, not a confidence interval. This is common knowledge and can be found in standard DSM textbooks and papers. A confidence interval is of limited interest, as it does not inform us about the uncertainty of new observations. The authors should therefore obtain a prediction interval. Second, the idea of bootstrapping a random forest model is not meaningful. Random forest already uses bootstrapping internally, so the authors are effectively bootstrapping twice. A much simpler approach, and one that is widely implemented, is to use quantile regression forest, which directly reports a prediction interval. This is probably the most common procedure in DSM to

generate uncertainty intervals with machine learning. Third, the approach based on Z-scores relies on the assumption of normally distributed errors, which is generally not valid for machine learning.

**Author Response:** Regarding error reporting, we have enhanced our uncertainty reporting to provide detailed information about model performance. Total propagated uncertainty including model variance and residual variance are now included to support inferences over large areas, although they are not applied to spatial information. Additionally, as the reviewer suggested, we used random forest tools to calculate all associated errors. However, these tools do not provide pixel-based uncertainty; instead, pixel-level uncertainty maps must be calculated using a bootstrapping approach, which involves varying the training and testing data pools. We have not done any double bootstrapping for pixel level confidence interval or prediction intervals.

Our mapping pipeline employs this bootstrapping method for all types of machine learning models. While the random forest (RF) model includes additional tools for using Quantile Regression Forests (QRF) to predict conditional quantiles of the response variable, not just the conditional mean, the end results are often similar, as demonstrated in numerous publications. For this study, we provided 95% confidence intervals for the pixel values of mean SOC.

**Residual autocorrelation.** Once the above issues are addressed, the authors should check for residual autocorrelation and report the fitted variograms. It may be that kriging of the residuals is needed if autocorrelation remains.

The many major comments above suggest that the maps may present a misleading representation of the spatial patterns and average or total stocks. These points need to be addressed very seriously, and the authors should ensure familiarity with the state of the art in digital soil mapping, as most of these are standard procedures in the field.

**Author Response:**

We also evaluated residual autocorrelation, an issue that has been addressed in several of our group's previous publications (Weisbin et al., 2014; Saatchi et al., 2011; Xu et al., 2016, 2017, 2021; McRoberts et al., 2021; Cushman et al., 2023), and found that the impact of residual autocorrelation on regional-scale inferences is minimal. Its influence becomes relevant primarily when estimating the uncertainty of SOC means and totals for small regions. At larger spatial scales, however, semi-variograms indicate a rapid exponential decline in autocorrelation with distance (Weisbin et al., 2014; Xu et al., 2017).

**References:**

Cushman, K. C., Albert, L. P., Norby, R. J., and Saatchi, S.: Innovations in plant science from integrative remote sensing research: an introduction to a Virtual Issue, New Phytologist, 240, 1707–1711, <a href="https://doi.org/10.1111/nph.19237">https://doi.org/10.1111/nph.19237</a>, 2023.

McRoberts, R. E., Næsset, E., Saatchi, S., and Quegan, S.: Statistically rigorous, model-based inferences from maps, Remote Sensing of Environment, 279, 113028, https://doi.org/10.1016/j.rse.2022.113028, 2022.

Saatchi, S. S., Harris, N. L., Brown, S., Lefsky, M., Mitchard, E. T. A., Salas, W., Zutta, B. R., Buermann, W., Lewis, S. L., Hagen, S., Petrova, S., White, L., Silman, M., and Morel, A.: Benchmark map of forest carbon stocks in tropical regions across three continents, Proceedings of the National Academy of Sciences, 108, 9899–9904, https://doi.org/10.1073/pnas.1019576108, 2011.

Weisbin, C. R., Lincoln, W., and Saatchi, S.: A Systems Engineering Approach to Estimating Uncertainty in Above-Ground Biomass (AGB) Derived from Remote-Sensing Data, Systems Engineering, 17, 361–373, <a href="https://doi.org/10.1002/sys.21275">https://doi.org/10.1002/sys.21275</a>, 2014.

Xu, L., Saatchi, S. S., Yang, Y., Yu, Y., and White, L.: Performance of non-parametric algorithms for spatial mapping of tropical forest structure, Carbon Balance and Management, 11, 18, https://doi.org/10.1186/s13021-016-0062-9, 2016.

L. Xu, S. S. Saatchi, A. Shapiro, V. Meyer, A. Ferraz, Y. Yang, J.-F. Bastin, N. Banks, P. Boeckx, H. Verbeeck, S. L. Lewis, E. T. Muanza, E. Bongwele, F. Kayembe, D. Mbenza, L. Kalau, F. Mukendi, F. Ilunga, D. Ebuta, Spatial distribution of carbon stored in forests of the Democratic Republic of Congo. *Sci. Rep.* **7**, 15030. 2017.

Xu, L., Saatchi, S. S., Yang, Y., Yu, Y., Pongratz, J., Bloom, A. A., Bowman, K., Worden, J., Liu, J., Yin, Y., Domke, G., McRoberts, R. E., Woodall, C., Nabuurs, G.-J., de-Miguel, S., Keller, M., Harris, N., Maxwell, S., and Schimel, D.: Changes in global terrestrial live biomass over the 21st century, Science Advances, 7, eabe9829, <a href="https://doi.org/10.1126/sciadv.abe9829">https://doi.org/10.1126/sciadv.abe9829</a>, 2021.